# THE INDUCTIVE BIAS OF IN-CONTEXT LEARNING: RETHINKING PRETRAINING EXAMPLE DESIGN

**Yoav Levine, Noam Wies, Daniel Jannai, Dan Navon, Yedid Hoshen & Amnon Shashua**
The Hebrew University of Jerusalem
`{yoav.levine, noam.wies, daniel.jannai, dan.nav}@mail.huji.ac.il`

## ABSTRACT

Pretraining Neural Language Models (NLMs) over a large corpus involves chunking the text into training examples, which are contiguous text segments of sizes processable by the neural architecture. We highlight a bias introduced by this common practice: we prove that the pretrained NLM can model much stronger dependencies between text segments that appeared in the same training example, than it can between text segments that appeared in different training examples. This intuitive result has a twofold role. First, it formalizes the motivation behind a broad line of recent successful NLM training heuristics, proposed for the pretraining and fine-tuning stages, which do not necessarily appear related at first glance. Second, our result clearly indicates further improvements to be made in NLM pretraining for the benefit of Natural Language Understanding tasks. As an example, we propose "kNN-Pretraining": we show that including semantically related non-neighboring sentences in the same pretraining example yields improved sentence representations and open domain question answering abilities. This theoretically motivated degree of freedom for *pretraining example design* indicates new training schemes for self-improving representations.

## 1 INTRODUCTION

Beyond excelling in their core task of pure language modeling, modern Neural Language Models (NLMs) show impressive zero- and few-shot abilities in more general Natural Language Understanding (NLU) tasks (Brown et al., 2020). This implies that the training corpus contains the information required for performing such tasks, and moreover it implies that the common pretraining process grants the trained NLM some access to these higher level capabilities. In this paper, we highlight a connection between the quality of the emergent NLU capabilities and a basic component in the NLM training scheme: the process of segmenting the corpus into training examples.

Specifically, NLMs self-train over huge training corpora (typically, billions to trillions of words). A basic, automatic, operation in the training pipeline is to segment these corpora into training examples: contiguous text chunks of sizes processable by the neural architecture (typically, up to thousands of words). We formalize an expressivity bias that this segmentation process introduces, to be referred to as the *in-context bias*, which directly affects the NLM's ability to integrate cross-corpus information. We show that the NLM can model much stronger dependencies between sentences that were shown together at least once in-context, *i.e.*, in the same training example, than between sentences that were never shown together in the same input. This inductive bias may be good for language modeling, but it implies that NLU capabilities that involve integrating information from different examples across the corpus (see, *e.g.*, figure 1), are under-favored by design in the current setting. Thus, if one sentence in the corpus can elucidate the meaning of another sentence (e.g., defines a hard concept or provides auxiliary information), our result implies that a model that saw them in different training examples will enjoy this elucidation less than a model that saw them in the same training example.

While standard approximation results examine the expressivity of an architecture over a single input, our theoretical approach pertains to the entire training process, and examines the expressive capacity of the resultant NLM with respect to the training set. Therefore, our approximation result ties an optimization parameter (the learning-rate) to the regular NLM architecture expressivity parameters (depth, width). Intuitively, sentences that were never shown in the same input can only access each other via the weights of the network during training. The mechanism for "storing" information in

the network involves a very small learning-rate term $\eta$; our analysis formalizes and quantifies an "expressivity toll" that the model pays when making use of such harder-to-access stored information.

We employ the tool of a function's separation rank with respect to subsets of its variables, which quantifies its ability to model input dependencies between these subsets. The separation rank was employed for analyzing the dependencies modeled by convolutional (Cohen & Shashua, 2017), recurrent (Levine et al., 2018), and self-attention (Levine et al., 2020) networks with respect to a single input example. In order to analyze an NLM's ability to model dependencies between *different* training examples, we refine the usage of this measure in two manners: (1) we introduce the $\varepsilon$-separation rank, which measures the effective ability of a function to model dependencies in a finite precision setting, and (2) we modify the separation rank such that it can account for the more intricate mechanism of mixing between variables that occurs in the sequential case.

Specifically, we upper bound the log of the separation rank of a depth $L$ width $d_x$ self-attention based NLM, with respect to two sentences that are shown in its input, by $\tilde{O}(d_x L)$, and prove that this bound is tight. On the other hand, we upper bound this measure with respect to two sentences that were never shown in the same input by $\tilde{O}(d_x[L - 0.5\log_3(\eta^{-1})])$. Given common learning-rate values of $\eta \in [10^{-6}, 10^{-4}]$, this implies a guaranteed "depth deficit" of $\sim 6$ layers for modeling dependencies between sentences that are not seen in the same training example. After the presentation of our results, we point at empirical evidence that imply that this depth deficit is more significant, and may behave like a fraction of $L$. We leave attempts to tighten the depth deficit estimates to future work.

### 1.1 THE IN-CONTEXT BIAS DRIVES A VARIETY OF EXISTING APPROACHES

Several recent works intuitively rely on the above formalized in-context expressivity bias in different manners, and significantly improve both task-specific training and pretraining of NLMs. Gao et al. (2020) advance the frontier in $k$-shot learning via finetuning. They show that by concatenating several related training examples per input, instead of using standard fine-tuning practice of one example per input, the $k$-shot performance on sentence similarity tasks is considerably boosted. Another example was pointed out in Humeau et al. (2020); Thakur et al. (2020): when training for sentence similarity tasks, including both sentences in the same input leads to a performance gain of around 10 points relative to separately encoding each sentence. In the challenging setting of open-domain question answering, Izacard & Grave (2020) jointly attend to all documents that may contain the answer, and show large gains relative to prior methods that consider these documents in separate forward passes.

Turning our focus to methods that leverage the in-context bias for improved pretraining, the most straightforward effort is a body of work aimed at reducing the quadratic dependence of the Transformer computation on input sequence length (Tay et al., 2020). While allowing for more text in-context during training, this does not improve the model's ability to integrate text across different documents in the corpus. The following approaches take a further step and enable direct cross-corpus connections during pretraining. Lewis et al. (2020) attend to related documents when maximizing the likelihood of a target document. The scope of related documents is restricted by meta-data: taken from the same Wikipedia entry as the input, or published on the same date. Guu et al. (2020) expand the scope of the related documents, by training a Knowledge-Retrieval model that has access to the entire Wikipedia corpus. They retrieve several related documents per target document, but condition on each related document independently. Outside of the natural language domain, Rao et al. (2021) train a Transformer based protein-LM that receives multiple related protein sequences in-context. Their protein-LM surpasses previous methods which process one sequence per input by a wide margin, with significant parameter efficiency.

### 1.2 LEVERAGING THE IN-CONTEXT BIAS FOR NLU ORIENTED TRAINING

Though the in-context bias is intuitive, the above subsection surveys recent advances that leverage it in non-trivial manners. Having formalized the theoretical advantage for in-context integration of related text, the roots of the above successes can be unified, and importantly, new methods for tilting the pretraining bias towards NLU tasks are indicated. Following the presentation of our theoretical results in section 2, we detail in section 3 two controlled setting exemplifications of new methods that directly leverage the in-context bias.

Our first experiment augments the Task Adaptive PreTraining (TAPT) setting of Gururangan et al. (2020), in which an NLM that was pretrained on a general corpus continues pretraining (with its original objective) on the training set of an NLU task. We perform TAPT on the SentEval sentence similarity benchmark (Conneau & Kiela, 2018), and during TAPT introduce the following

augmentation: along with SentEval sentences, we simultaneously pretrain on related sentences from Wikipedia, the general pretraining corpus. The related sentences are found via k-Nearest Neighbors (kNN) search between the embeddings of SentEval examples and all Wikipedia sentences; we thus dub this approach *kNN-TAPT*. Importantly, during kNN-TAPT, each input includes a training example from the task, appended *in-context* by its Wikipedia neighbors. We demonstrate significant gains of the kNN-TAPT over regular TAPT on SentEval sentence similarity tasks. A dedicated ablation study shows the significance of adding the general corpus neighbors in-context, versus in separate training examples, during kNN-TAPT.

Our second experiment introduces a task-independent pretraining phase, dubbed kNN-Pretraining. As in kNN-TAPT, we group together sentences with similar sentence representations in the same training example, but in kNN-Pretraining we use only sentences from the general pretraining corpus. This can be viewed as a sentence-focused variation of the above surveyed pretraining schemes in Lewis et al. (2020) and Guu et al. (2020), who operate on full documents (up to 512 each), and is very similar to RETRO by Borgeaud et al. (2021) (DeepMind), who show the benefits of this

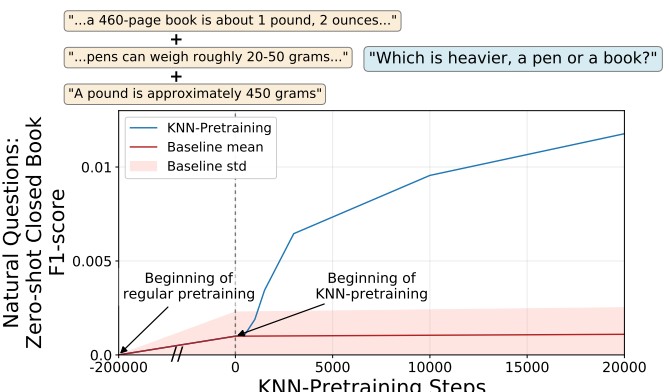

Figure 1: A 10% addition of kNN-Pretraining boosts zero-shot closed book QA score by∼ 5X (evaluation set size is 20,000).

approach given much larger resources. Figure 1 shows that after regular pretraining for 200K steps on Wikipedia, the zero-shot closed book performance of 3 different randomly initialized GPT2-medium models (345M parameters) on open domain questions from Wikipedia (Kwiatkowski et al., 2019) is very low (correct on less than 50 questions out of 20,000 in the evaluation set). Adding kNN-Pretraining for 20K steps raises performance significantly (correct on roughly 250 questions in the evaluation set), reflecting the enhanced ability to integrate knowledge from related sentences, acquired via the in-context bias.

In summary, our main contributions are:

- We formally establish the *in-context bias*: information within pretraining examples is better represented than information integrated across pretraining examples.
- We ask and answer a new type of network expressivity question: *how expressive is a network with respect to examples seen during its training process?*
- We demonstrate that in-context bias motivated "pretraining example design" elicits better representations from the same data: *kNN-Pretraining* improves on several NLU tasks.

## 2   THEORETICAL ANALYSIS: THE IN-CONTEXT BIAS OF SELF-ATTENTION

In this section, we consider the entire NLM training procedure as a functional that receives an unlabeled training corpus and outputs a trained NLM. Our analysis focuses on the corpus segmentation into training examples as a hyper-parameter of this functional. We reduce the high-level notion of representing "cross-corpus correlations" to a quintessential case study: we quantify the NLM's ability to model dependencies between two sentences that appear in the same training example (the *in-context* representation) and in different training examples (the *sequential* representation).

We believe that the in-context bias can be shown to exist in a broad range of architectures, but we focus on self-attention since almost all modern NLMs are based on the Transformer architecture of Vaswani et al. (2017). Our theoretical framework is based on that of Levine et al. (2020); Wies et al. (2021), who analyze a simplified, theoretically accessible, self-attention network. They study the expressivity of this self-attention architecture with respect to its input, and use a measure of a multivariate function's ability to correlate two subsets of its variable set, referred to as the separation rank. The analyzed framework captures the connectivity of self-attention but omits its softmax and ReLU non-linearities (see eq. 1 below). We refer the reader to Levine et al. (2020); Wies et al. (2021) for a discussion on the impact of these relaxations. Essentially, they are shown to weaken the overall network power but still allow a meaningful comparison of the self-attention integration abilities.

Importantly, both works derive unforeseen theoretical conclusions from analyses of the separation rank measure for this architecture class, and then provide extensive empirical corroboration for their manifestation in common Transformer architectures, reinforcing the relevance of this setting. In the following, we describe in section 2.1 the analyzed *in-context* and *sequential* self-attention representations of two sentences. Then, in section 2.2, we present the separation rank, which we use in section 2.3 for quantifying the advantage of in-context representations versus sequential ones.

## 2.1 THE ANALYZED IN-CONTEXT AND SEQUENTIAL REPRESENTATIONS

For an input sequence of $N$ embedding vectors $\{\mathbf{x}^j \in \mathbb{R}^{d_x}\}_{j=1}^N$, denote the function realized by the analyzed $H$-headed depth-$L$ width-$d_x$ Transformer architecture at output location $i \in [N]$ by: $\mathbf{g}_{\mathcal{W}}^{i,L,d_x}\left(\mathbf{x}^1, ..., \mathbf{x}^N\right) \in \mathbb{R}^{d_x}$, where $\mathcal{W}$ stands for learned parameters, recursively defined:

$$\mathbf{g}_{\mathcal{W}}^{i,l+1,d_x}\left(\mathbf{g}_{\mathcal{W}}^{1,l,d_x}, ..., \mathbf{g}_{\mathcal{W}}^{N,l,d_x}\right) = \sum_{h=1}^H W^{\mathrm{O},l,h} \sum_{j=1}^N a_{hj}^i W^{\mathrm{V},l,h} \mathbf{g}_{\mathcal{W}}^{j,l,d_x} \tag{1}$$

$$a_{hj}^i := \left\langle W^{\mathrm{Q},l,h} \mathbf{g}_{\mathcal{W}}^{i,l,d_x}, W^{\mathrm{K},l,h} \mathbf{g}_{\mathcal{W}}^{j,l,d_x} \right\rangle \quad ; \quad \mathbf{g}_{\mathcal{W}}^{i,0,d_x} = \mathbf{x}^i$$

where $\mathcal{W}$ is composed of Key, Query, Value and Output matrices: $\forall l \in [L], h \in [H]$, $W^{\mathrm{K},l,h}, W^{\mathrm{Q},l,h}, W^{\mathrm{V},l,h}, (W^{\mathrm{O},l,h})^\top \in \mathbb{R}^{d_a \times d_x}$, where we assume the standard choice $d_a = d_x/H$. For a word from vocabulary of size $V$, $w \in [V]$, the translation into the Transformer dimension is done via a mapping $E_{M^{\mathrm{V}}} : [V] \to \mathbb{R}^{d_x}$:

$$E_{M^{\mathrm{V}}}(w) = \left(M^{\mathrm{V}}\right)_w \tag{2}$$

where $M^{\mathrm{V}} \in \mathbb{R}^{d_x \times V}$ is the learned vocabulary embedding matrix, and $\left(M^{\mathrm{V}}\right)_w$ is its $w$th column, also referred to as the learned word embedding for $w$. Overall, the function of the analyzed Transformer over a sequence of $N$ words $\{w^j \in [V]\}_{j=1}^N$ can be written by composing eqs. 1 and 2:

$$\mathbf{y}_{\mathcal{W},M^{\mathrm{V}}}^{i,L,d_x}\left(w^1, ..., w^N\right) = \mathbf{g}_{\mathcal{W}}^{i,L,d_x}\left(E_{M^{\mathrm{V}}}\left(w^1\right), ..., E_{M^{\mathrm{V}}}\left(w^N\right)\right) \tag{3}$$

For simplicity of presentation, we examine two sentences $S_1$ and $S_2$ of equal length $N$: $S_1 = \{w_1^j\}_{j=1}^N$ and $S_2 = \{w_2^j\}_{j=1}^N$. The in-context representation simply concatenates both in the input:

$$\mathbf{y}_{\text{in-context}}^{i,L,d_x}(S_1, S_2) := \mathbf{y}_{\mathcal{W},M^{\mathrm{V}}}^{i,L,d_x}\left(w_1^1, ..., w_1^N, w_2^1, ..., w_2^N\right). \tag{4}$$

For the sequential approach, we consider a setup in which sentence $S_1$ is inserted into the network at training step $t$ and sentence $S_2$ is inserted into the network at training step $t + 1$. The output of the network at training step $t$ is therefore: $\mathbf{y}_{\mathcal{W}_t,M_t^{\mathrm{V}}}^{i,L,d_x}(S_1)$, where $\mathcal{W}_t, M_t^{\mathrm{V}}$ stand for all the learned weights before training step $t$. Focusing on autoregressive NLMs for simplicity of presentation (the analysis holds for bidirectional NLMs as well), the log-likelihood loss is given by $\mathcal{L}(S_1) = -\sum_{j=1}^N \log\left[\left(softmax\left\{\left(M_t^{\mathrm{V}}\right)^\top \mathbf{y}_{\mathcal{W}_t,M_t^{\mathrm{V}}}^{j,L,d_x}(S_1)\right\}\right)_{w_1^{j+1}}\right]$, and the gradient update for any learned weight $\theta \in \{\mathcal{W}_t, M_t^{\mathrm{V}}\}$ is: $\theta_{t+1}(S_1; \eta) = \theta_t - \eta \cdot \partial\mathcal{L}(S_1)/\partial\theta_t$, where $\eta$ is the learning rate. Accordingly, the analyzed sequential representation is the network output after training step $t + 1$:

$$\mathbf{y}_{\text{sequential}}^{i,L,d_x,\eta}(S_1, S_2) := \mathbf{y}_{\mathcal{W}_{t+1}(S_1;\eta),M_{t+1}^{\mathrm{V}}(S_1;\eta)}^{i,L,d_x}(S_2). \tag{5}$$

In practice, two relevant non-neighboring sentences are not necessarily shown in consecutive pre-training steps. In comparison to the realistic scenario of $S_1$ and $S_2$ appearing at *any* training step, this simplifications tilts the representation in favor of modeling high correlations between $S_1$ and $S_2$. Thus, by upper bounding the ability to correlate $S_1$ and $S_2$ in the setting of eq. 5 (as we do in section 2.3), we establish an inherent limitation of the network to access information that was stored in its weights via the gradient update mechanism. In the next subsection, we present our approach for measuring a network's ability to correlate two sentences seen during training, which we will use in order to separate between the in-context and sequential settings.

## 2.2 A MEASURE FOR MODELING IN-CONTEXT AND SEQUENTIAL DEPENDENCIES

In this section, we refine the separation rank, used in prior work in order to analyze the dependencies between two sentences appended in-context. In section 2.2.1 we present the separation rank and introduce a finite precision refinement of it, referred to as *the effective separation rank*, which helps to elucidate the degradation in integration ability caused by the gradient update mechanism. In section 2.2.2 we point at a structural problem in employing the separation rank in the same manner in which it was employed in prior work that analyzed only architecture expressivity, and introduce the *the sequential separation rank*, meaningful for both the in-context and sequential cases.

### 2.2.1 THE EFFECTIVE SEPARATION RANK

The separation rank has been established as a measure of dependencies modeled by deep convolutional, recurrent, and self-attention networks (Cohen & Shashua, 2017; Levine et al., 2018; 2020). For a function $y(A, B)$ over variables $A = \{\mathbf{a}^j \in \mathcal{X}\}_{j=1}^M$ and $B = \{\mathbf{b}^j \in \mathcal{X}\}_{j=1}^M$, the separation rank w.r.t. $(A, B)$ is the minimal number of summands that together sum up to equal $y(A, B)$, where each summand is *multiplicatively separable w.r.t.* $(A, B)$, *i.e.*, is equal to a product of two functions – one that intakes only $A$ variables and another that intakes only $B$ variables. Formally, the *separation rank* of $y : \mathcal{X}^{2M} \to \mathbb{R}$ w.r.t. $(A, B)$ is defined as follows:

$$\operatorname{sep}_{(A,B)}(y) := \min \left\{ R \in \mathbb{N} : \exists g_1 \ldots g_R, g_1' \ldots g_R' : \mathcal{X}^M \to \mathbb{R} \; s.t. \; y(A, B) = \sum_{r=1}^R g_r(A) g_r'(B) \right\}$$
(6)

If the separation rank of a function w.r.t. $(A, B)$ is 1, it is multiplicatively separable w.r.t. $(A, B)$, meaning it cannot take into account consistency between $A$ and $B$. The higher $\operatorname{sep}_{(A,B)}(y)$ is, the farther $y$ is from this situation, *i.e.*, the more it models dependency between $A$ and $B$. We will further make use of the *effective separation rank*:

$$\varepsilon\text{-}\operatorname{sep}_{(A,B)}(y) := \min \left\{ R' \leq \operatorname{sep}_{(A,B)}(y) \in \mathbb{N} : \exists \tilde{y} : \mathcal{X}^{2M} \to \mathbb{R} \; s.t. \right.$$
(7)

$$\left. \left[ \operatorname{sep}_{(A,B)}(\tilde{y}) = R' \right] \wedge \left[ \forall x \in \mathcal{X}^{2M} : |\tilde{y}(x) - y(x)| \leq \varepsilon \right] \right\}$$

In words, if a function has a high separation rank, but it can be approximated up to error $\varepsilon$ by a function with a low separation rank, then it has a low $\varepsilon$-separation rank.

Prior works compare two functions by establishing the differences between their *separation ranks*. In principle, these differences could manifest only in irrelevant magnitudes (if many of the summands in the separation rank definition are negligibly small for the function with the higher separation rank, for example). The *effective separation rank* is key to our analysis because we rely on the fact that information on past examples is stored in the network weights in a small magnitude (due to a small learning-rate). We show in section 2.3 that much of the integration between text segments from different training examples occurs in very small magnitudes due to high powers of the learning rate, limiting the effective integration, as measured by the $\varepsilon$-separation rank. Our techniques for bounding the $\varepsilon$-separation rank are extendable to prior works, and while these did not examine the gradient update mechanism, their results can be reinforced due to the guarantees of this introduced measure.

### 2.2.2 THE SEQUENTIAL SEPARATION RANK

Levine et al. (2020), who were the first to apply the separation rank to functions realized by Transformer architectures, studied classical architecture expressivity questions which apply only to the in-context representation. Accordingly, they analyzed only the separation rank of $\mathbf{g}^{i,L,d_x}$, defined in eq. 1, and the input variables considered for calculating the separation rank were the word embedding vectors. A fundamental difficulty arises when attempting to directly apply this method to the sequential representation: the word embedding vectors are learned parameters of the architecture. In the sequential case, when the second sentence $S_2$ is introduced after the calculation at time-step $t$, the vectors used to describe it, if we were to follow prior practice, would already have depended on $S_1$.

In order to meaningfully measure the integration ability of two sentences across examples in the presence of the gradient update mechanism, we introduce an auxiliary *sentence association layer* with new variables: $\mathbf{a}, \mathbf{b} \in \mathbb{R}^{d_x}$, which explicitly associates each employed vocabulary embedding vector with the sentence index $s \in \{1, 2\}$ that invoked its usage:

$$Z^s \left( E_{M^{\mathrm{v}}}(w_s^j) \right) := \begin{cases} E_{M^{\mathrm{v}}}(w_s^j) \odot \mathbf{a} & \text{if } s = 1 \\ E_{M^{\mathrm{v}}}(w_s^j) \odot \mathbf{b} & \text{if } s = 2 \end{cases}$$
(8)

where $\odot$ denotes element-wise multiplication. We define the sentence association operation over the analyzed representations, denoted $\mathcal{Z}_y(\mathbf{a}, \mathbf{b})$ with $y \in \{\mathbf{y}_{\text{in-context}}^{i,L,d_x}(S_1, S_2), \mathbf{y}_{\text{sequential}}^{i,L,d_x,\eta}(S_1, S_2)\}$ (eqs. 4 or 5), to be the application of the sentence association layer of eq. 8 to all uses of the input embedding layer during the computation of $y$. Meaning, for both mechanisms, that chosen word embeddings are marked with the identity of the sentence that invoked them. Finally, we define the following specialization of the separation rank measure to our setting, referred to as the *sequential separation rank* of $y \in \{\mathbf{y}_{\text{in-context}}^{i,L,d_x}(S_1, S_2), \mathbf{y}_{\text{sequential}}^{i,L,d_x,\eta}(S_1, S_2)\}$:

$$\operatorname{seq-sep}(y) := \operatorname{sep}_{(\mathbf{a},\mathbf{b})}(\mathcal{Z}_y)$$
(9)

$$\varepsilon\text{-}\operatorname{seq-sep}(y) := \varepsilon\text{-}\operatorname{sep}_{(\mathbf{a},\mathbf{b})}(\mathcal{Z}_y)$$

Clearly, when the introduced variables are vectors of $\mathbf{1}$, the auxiliary layer in eq. 8 is the identity operation and so $\mathcal{Z}_y(\mathbf{1}, \mathbf{1}) = y$ for both representations. More deeply, our expressivity questions query the ability of the in-context and sequential mechanisms to integrate two sets of variables, and $\mathcal{Z}_y$ captures the essence of this ability by explicating where each set enters the computation.

In the next subsection, we show that for the in-context case, analyzed in prior work, the introduced measure of the sequential separation rank is asymptotically equal to the previously employed measure of separation w.r.t. a partition of the input word embeddings (Levine et al., 2020). Thus, the properties of the existing framework are unchanged under the new definition. At the same time, for the sequential case brought forth in this paper, the sequential separation rank considers both the effect of $S_1$ on the gradient-updated word embedding and the introduction of $S_2$ into the computation.[1] In the following section, we make use of both extensions to the separation rank in eqs. 7 and 9 in order to establish the in-context bias.

## 2.3 THE EXPRESSIVE ADVANTAGE OF IN-CONTEXT LEARNING

We show below that the function computed by a self-attention based NLM when inserting sentences $S_1$ and $S_2$ together in its input (the in-context representation) can model more elaborate dependencies between $S_1$ and $S_2$ than the function attained when showing $S_1$ in the input, modifying the network's weights according to its loss, and then showing $S_2$ in a subsequent input (the sequential representation). We begin by stating the following corollary, following from theorem 2 in Levine et al. (2020) and proposition 1 in appendix A, which upper bounds the sequential separation rank of the *in-context* representation:

**Corollary 1.** *Let $y_{\text{in-context}}^{(p,i),L,d_x}$ be the $p \in [d_x]$ entry of the analyzed in-context representation defined in eq. 4. Assume that $L > \log_3 d_x$. Then ($\tilde{O}$ notation omits log terms: $\log d_x, \log L, \log H$):*

$$\log\left[\text{seq-sep}\left(y_{\text{in-context}}^{(p,i),L,d_x}\right)\right] = \tilde{O}\left(L \cdot d_x\right) \tag{10}$$

However, the $\varepsilon$-separation rank of the *sequential* representation is upper bounded by a lower term:

**Theorem 1.** *(See proof in appendix B). Let $y_{\text{sequential}}^{(p,i),L,d_x,\eta}$ be the $p \in [d_x]$ entry of the analyzed sequential representation defined in eq. 5. Assume that all learned parameters and all gradients are bounded: $\forall \theta \in \{\mathcal{W}, M^V\} : 0 < \Lambda_{\min} \leq |\theta|, |\partial\mathcal{L}(S_1)/\partial\theta| \leq \Lambda_{\max},$[2] $N < d_x$, and that $L > \log_3 d_x$. Then, $\forall \varepsilon > 0$:*

$$\log\left[\varepsilon\text{-seq-sep}\left(y_{\text{sequential}}^{(p,i),L,d_x,\eta}\right)\right] = \tilde{O}\left([L + 0.5\log_3(\eta)] \cdot d_x\right) \tag{11}$$

Therefore, a gap between upper bounds on the ability to model dependencies between $S_1$ and $S_2$ is indicated. Since the learning rate $\eta$ is a small term, its log is negative and the gap is in favor of the in-context representation. The following theorem guarantees that this gap is meaningful, by showing that the higher upper bound (of the in-context case) is tight in terms of effective rank:

**Theorem 2.** *(See proof in appendix C). For $y_{\text{in-context}}^{(p,i),L,d_x}$ as defined in corollary 1, there exists an assignment of the network weights for which the following holds:*

$$\log\left[\varepsilon\text{-seq-sep}\left(y_{\text{in-context}}^{(p,i),L,d_x}\right)\right] = \tilde{\Omega}\left(L \cdot d_x\right) \tag{12}$$

*where $\varepsilon = \tilde{O}\left(\left(\binom{3^L+d_x-1}{3^L}\right)^{-1}\right)$.*

Notably, corollary 1 and theorem 2 show that for the in-context case, the sequential separation rank asymptotically equals the regular separation rank, validating the relevance of this measure.

We now provide a high level proof sketch that captures the manner in which the theoretical framework of sections 2.1 and 2.2 is used for establishing the above gap (full proof in the appendix). For the **in-context** case, notice that each self-attention layer, defined in eq. 1, is a degree 3 polynomial over its $2N \cdot d_x$ inputs, rendering the whole network a degree $3^L$ polynomial. We write this polynomial as a sum over many monomials, and by definition, the separation rank of any monomial composing the polynomial is 1. Since the separation rank of a sum of functions is upper bounded by the sum of their separation ranks, we upper bound the separation rank by the number of these monomials, yielding eq. 10. The main difference in the **sequential representation** case is that the $S_1$ variables affect the

---

[1]To see this, note that for $s = 2$ the operation in eq. 8 includes both $M_{t+1}^V(\mathbf{a})$ and variables from $\mathbf{b}$.

[2]The upper boundedness assumption resembles practices of gradient clipping and weight decay, and the lower boundedness assumption resembles finite precision.

computation only via the gradient, so their impact is expected to be limited. However, considering that $S_2$ first encounters gradient updated vocabulary matrix entries $\left(m^{\mathsf{V}}\right)_{t+1} = \left(m^{\mathsf{V}}\right)_t - \eta \partial \mathcal{L}(S_1)/\partial \left(m^{\mathsf{V}}\right)_t$, it appears that both $S_1$ and $S_2$ variables enter the self-attention stack via its input, similarly to the in-context case. So the integration between $S_1$ and $S_2$ occurs right from the start, and indeed we show that the *separation rank* of both representations is similar. However, since any function of $S_1$ is accompanied by the learning-rate $\eta$, the monomials for which there are many $S_1$ variables will be multiplied by high powers of $\eta$. This causes many monomials to be negligibly small, and accordingly not to contribute to the *$\varepsilon$-separation rank*. By combinatorial considerations we show that the number of monomials that are not attenuated by $\eta$ (have sufficiently large magnitude) yields eq. 11. $\square$

The above theorems establish that from an expressivity perspective, the small magnitude of commonly employed learning-rates hinders the ability to integrate information across different training examples. Specifically, the established gap implies that the power of the joint representation of two sentences shown in different training examples is upper bounded by that of a network shallower by $0.5 \log_3(\eta^{-1})$ layers that has seen them in the same context. Common learning-rate values are on the order of $\eta \in [10^{-6}, 10^{-4}]$, implying a deficit of $\sim 6$ layers in the sequential case. As shown in in Levine et al. (2020); Tay et al. (2021), in many practical regimes of network size depth is crucial for expressivity, reinforcing the implications of this gap.

The weaker upper bound, of the sequential case, is not guaranteed to be tight. This means that theoretically, the sequential representation may in fact be much weaker than what we have proven, *e.g.*, that showing two sentences in the same context yields a representation that cannot be matched merely by showing them in separate contexts and adding a realistic number of layers. However, Roberts et al. (2020) show evidence supporting our indicated link between architectural parameters and the in-context bias. They show that when performing open domain question answering tasks (their defined "closed book" setting), a large T5 model that sees only the question performs comparably to smaller models that are allowed to attend to the documents that contain the answer. This directly implies a certain strength of the sequential mechanism, namely, that information which was seen during training can be accessed via the weights when the model is realistically stronger, as implied by our bounds. Notably, the large T5 model is 2-4 times the depth of the contrasted smaller models (48 versus 12-24 layers), suggesting that the upper bound can be tightened to a fraction of $L$, or that factors that are beyond expressivity also contribute to the in-context bias (*e.g.*, optimization, generalization). Investigation of these aspects is left for future work.

## 3    KNN BASED PRETRAINING EXAMPLE DESIGN

Our theoretical analysis quantifies the relation between the small magnitude of the learning rate, and the deficiency in the ability to model dependencies between different training examples. Clearly, small learning-rates are critical for optimization purposes, so the formalized phenomenon should not be solved via high learning-rates during training. Instead, our analysis makes it clear that if correlations between specific sentences are important for a given task, appending them in-context yields better representations for the task. Below, we describe two controlled experiments that demonstrate the importance of this indicated "pretraining example design" degree of freedom. In both experiments, correlated sentences are identified via kNN search in their RoBERTa-large sentence representation space (Reimers & Gurevych, 2019), performed using the FAISS library (Johnson et al., 2019).

### 3.1    KNN TASK ADAPTIVE PRETRAINING

The Task Adaptive PreTraining (TAPT) method, in which an NLM pretrains on the training set of an NLU task, leads to impressive gains (Gururangan et al., 2020). Notably, TAPT is most effective after the regular pretraining stage on a general corpus. This implies that during TAPT, the model generates improved representations by integrating the task related text with the knowledge stored in its weights from the preceding general pretraining phase. Under this premise, we postulated that performance will improve if we make relevant sentences from the general corpus more available to the model during the TAPT phase. According to the above analysis, a simple and effective way to bias the model towards representing desired correlations between sentences is to append them in context.

We thus propose the *kNN-TAPT* phase, in which the training examples are composed of task examples, concatenated with their general corpus neighbors in embedding space. We applied kNN-TAPT on the SentEval sentence similarity tasks. Showing similar sentences from Wikipedia is expected to be particularly useful on these tasks, so this is a good experimentation ground to search for effects of the in-context bias. For each SentEval example, we searched over 100M Wikipedia sentences and appended in-context neighbors that have embeddings with over 0.8 cosine similarity to the SentEval

|  | STS12 | STS13 | STS14 | STS15 | STS16 | STS-B | SICK-R | Avg. |
|---|---|---|---|---|---|---|---|---|
| Basline Roberta Model | 32.1 | 56.3 | 45.2 | 61.3 | 62.0 | 55.4 | 62.0 | 53.5 |
| TAPT | 43.0 | 62.2 | 51.6 | **70.6** | 64.9 | 63.0 | 63.5 | 59.8 |
| kNN-TAPT (random, in-batch) | 40.2 | 62.7 | 51.9 | 64.9 | 62.1 | 61.5 | 65.4 | 58.4 |
| kNN-TAPT (neighbors, in-batch) | 40.8 | 62.4 | **53.1** | 66.1 | 63.0 | 61.3 | 65.2 | 58.8 |
| kNN-TAPT (random, in-context) | 44.62 | 62.64 | 51.4 | 65.28 | 64.93 | 64.31 | 66.96 | 60.0 |
| kNN-TAPT (neighbors, in-context) | **44.9** | **63.4** | 52.1 | 66.2 | **65.3** | **66.5** | **68.3** | **61.0** |

Table 1: kNN-TAPT, which augments the Task Adaptive PreTraining (TAPT) setting of Gururangan et al. (2020), harnesses the in-context bias and improves SentEval sentence similarity scores.

example embedding, with a special token inserted between different sentences. We continued until finding no more neighbors or reaching a maximum of 256 tokens in the RoBERTa vocabulary (Liu et al., 2019). This search yielded 170K examples, over which we continued training a pretrained RoBERTa-base model for 5 epochs, using the first epoch for learning-rate warmup and examining peak learning rates of $\{1, 3, 5, 7\} \cdot 10^{-5}$. See appendix D for implementation details.

Table 1, shows zero-shot SentEval sentence similarity scores, attained by using the average word embedding of an inserted sentence as its examined sentence representation (shown by Reimers & Gurevych (2019) to be most meaningful in zero shot). All models were trained according to the above prescription, besides the baseline RoBERTa which was simply evaluated. kNN-TAPT improves over regular TAPT, by over 1 point on average, implying that the Wikipedia neighbors are indeed useful to the TAPT stage. We compared 4 kNN-TAPT variants as an ablations study. Importantly all variants labeled with kNN-TAPT train on the same training data during the TAPT stage – the SentEval sentence similarity tasks training sets and their Wikipedia nearest neighbors, and differ only in the arrangement of the data into training examples. The "neighbors" flag relates a SentEval example to its actual neighbors from the kNN search, while the "random" flag relates it to random Wikipedia sentences from the overall neighbors pool attained in the search. The "in batch" flag implies that related sentences were shown in the same batch, where every training example includes only one sentence from either SentEval or Wikipedia. In contrast, the "in context" flag implies that related sentences were shown in the same training example.

The weakness of "neighbors, in-batch" implies that the a-priori plausible approach of biasing the model to learn from these Wikipedia neighbors via placing them in the same batch is not nearly as effective as the theoretically motivated in-context approach. Leading sentence representations employ in-batch techniques (see for example the contrasive setting of Gao et al. (2021b)), and this signal strongly suggests developing in-context parallels. The fact that the original TAPT scheme outperforms the in-batch approaches implies that including the Wikipedia sentences in separate training examples is harmful. We postulate that this is because training examples that have only Wikipedia sentences actually dilute the original TAPT signal. Indeed, by this view, the reason that "random, in-context" performs comparably to TAPT, is that it does not dilute the original TAPT signal – every training example includes a SentEval example. Overall, the clear advantage of the "neighbors, in-context" kNN-TAPT variant encourages leveraging the in-context bias for TAPT in further tasks.

## 3.2 KNN PRETRAINING

We extended the above to more general *kNN-Pretraining*, designing pretraining examples with related non-neighboring sentences given only the general pretraining corpus. kNN-Pretraining is also motivated by the kNN-LM results of Khandelwal et al. (2019), who show significant benefits of using nearest neighbors in representation space at *inference* time. Their results exemplify the potential impact of integrating cross-corpus related examples; our kNN-Pretraining approach provably biases the model to learn these correlations at *pretraining time*, via the in-context bias.

Specifically, we performed kNN search over Wikipedia sentences for every sentence in Wikipedia, and created each training example similarly to the protocol in the previous subsection. During kNN-Pretraining, half of the batch contained regular pretraining examples and half contained the prepared kNN examples, in order to retain longer ranged LM abilities. To examine the effect of kNN-Pretraining, we pretrained GPT-base and GPT-medium (110M and 345M parameters) architectures from scratch over Wikipedia in the regular pretraining scheme, and switched to kNN-Pretraining at two different points during pretraining (200K and 400K). The training examples were of maximal size 256, and the batch size was 128 for the GPT-medium models and 256 for the GPT-base models.

In order to directly probe the acquired ability to integrate non-neighboring sentences, we evaluated the resultant models on the very challenging setup of zero-shot closed-book open domain question

answering. In this setup, the unidirectional pretrained model decodes an answer conditioned on the given open ended question. We evaluated the models on questions from the Natural Questions (NQ) benchmark (Kwiatkowski et al., 2019), using the same phrasing employed in Brown et al. (2020), and employing the standard "open-domain" version as used e.g. by Lee et al. (2019); Asai et al. (2019); Roberts et al. (2020). NQ is composed of questions that have answers within Wikipedia, our pretraining corpus. kNN pretraining can imtuitively improve in cases where the passage containing the answer has elucidating nearest neighbors from across wikipedia that would help the model to better internalize the answer, such that it is more accessible to the model in zero shot. As figure 1 demonstrates, 3 baseline models, pretrained with the regular scheme, achieve very low F1$< 10^{-3}$ scores on this task. In contrast, kNN-Pretraining shows a low-scoring but significant improvement.

To increase the credibility of the signal, we evaluated our models on the first 20K examples from the NQ training set (we tested zero-shot performance, so the training set was not used earlier). Indeed, the attained F1 scores are low, but they correspond to 100s of correct answers that the kNN-Pretrained model provide after roughly 10% of the overall training time, versus much less in the 3 randomly initialized baseline models. Finally, we include in appendix E NQ scores of models of different sizes when starting kNN-Pretraining at different checkpoints, and in appendix F zero-shot scores on several GLUE tasks, which demonstrate clear gains of kNN-Pretraining over the baselines.

## 4 Discussion

Modern NLM pretraining schemes have tremendously advanced the natural language landscape, since they allowed powerful models to train on huge amounts of unlabeled text. But NLMs are now challenged with tasks which require deeper and more nuanced understanding of text, and means of improving the basic pretraining process should be considered. For a given architecture, pretraining can be improved by adding more data or finding more sophisticated training objectives to apply over existing data. In this paper we highlight a parallel path for improvement, which employs the same data and objective, but redistributes the available strength of the Transformer architecture such that important connections within the pretraining corpus are learned more effectively. Specifically, we highlight the bias of the trained NLM towards modeling dependencies between chunks of text that appeared within the same training examples. In current pretraining schemes, this means that dependencies between non-neighboring chunks of text are under-favored. If such dependencies matter for the task at hand, we suggest rearranging the data into corresponding training examples.

We formalize the above notion. Our theoretical setup asks expressivity questions that pertain to the training set rather than to a single example. We thus tie the construction of the training example with the available expressivity of the architecture: we prove that the connections that can be modeled between different training examples are bounded by the connections that can learned by a shallower and weaker architecture, if these examples were inserted within the same input.

The advantage in including related text in the input of the NLM is noticed and leveraged in the empirical landscape. With that, it is clear that showing the model related data is meaningful even if it is in different training examples, and many leading methods elect to do just that. Our quantification of this trade-off is intended to aid informed decisions and highlight the expressivity advantage to be gained by smarter training example designs. We follow up on these recommendations and demonstrate the immediately available gains to be achieved by designing training examples that include nearest neighbors in embedding space. This method can be enhanced, and other more explicit biases can be introduced. For example, multiple mentions of the same entity, event, or concept can be concatenated within the same training example.

The gains achieved by using similarity in representation space indicate a path for self-improving representations, left for future work. After a first cycle of kNN-Pretraining, the representation is refined and applying a new kNN search over it can lead to more informative next round of kNN-Pretraining. This way, deeper insight can be elicited from a given pretraining corpus.

Lastly, while this paper focused on leveraging the identified in-context bias for pretraining, it can also be tied to recent successes of in-context inference methods. From the in-context few-shot prompts of Brown et al. (2020), to in context augmentations such as in Gao et al. (2020); Schick & Schütze (2020) and many others, the benefits of biasing the prediction by appending text in-context are now widely established. The tools brought forth here can assist in clarifying the theoretical advantages of such practices. Overall, our work aims to provide timely theoretical interpretations, to help guide the rapid empirical advances of our field.

ACKNOWLEDGMENTS

We thank Or Sharir, Kevin Leyton-Brown, and Ori Ram for useful discussions. This research was supported by the ERC (European Research Council) and the ISF (Israel Science Foundation). Experiments were performed with Cloud TPUs and supported by Google's TensorFlow Research Cloud (TFRC). Yoav Levine was supported by the Israel Academy of Sciences Adams fellowship.

Noga Alon, Troy Lee, Adi Shraibman, and Santosh Vempala. The approximate rank of a matrix and its algorithmic applications: approximate rank. In *Proceedings of the forty-fifth annual ACM symposium on Theory of computing*, pp. 675–684, 2013.

Akari Asai, Kazuma Hashimoto, Hannaneh Hajishirzi, Richard Socher, and Caiming Xiong. Learning to retrieve reasoning paths over wikipedia graph for question answering. *arXiv preprint arXiv:1911.10470*, 2019.

Sebastian Borgeaud, Arthur Mensch, Jordan Hoffmann, Trevor Cai, Eliza Rutherford, Katie Millican, George van den Driessche, Jean-Baptiste Lespiau, Bogdan Damoc, Aidan Clark, et al. Improving language models by retrieving from trillions of tokens. *arXiv preprint arXiv:2112.04426*, 2021.

Tom B Brown, Benjamin Mann, Nick Ryder, Melanie Subbiah, Jared Kaplan, Prafulla Dhariwal, Arvind Neelakantan, Pranav Shyam, Girish Sastry, Amanda Askell, et al. Language models are few-shot learners. *arXiv preprint arXiv:2005.14165*, 2020.

Nadav Cohen and Amnon Shashua. Inductive bias of deep convolutional networks through pooling geometry. In *5th International Conference on Learning Representations (ICLR)*, 2017.

Alexis Conneau and Douwe Kiela. Senteval: An evaluation toolkit for universal sentence representations. *arXiv preprint arXiv:1803.05449*, 2018.

Ido Dagan, Bill Dolan, Bernardo Magnini, and Dan Roth. Recognizing textual entailment: Rational, evaluation and approaches–erratum. *Natural Language Engineering*, 16(1):105–105, 2010.

Leo Gao, Jonathan Tow, Stella Biderman, Sid Black, Anthony DiPofi, Charles Foster, Laurence Golding, Jeffrey Hsu, Kyle McDonell, Niklas Muennighoff, Jason Phang, Laria Reynolds, Eric Tang, Anish Thite, Ben Wang, Kevin Wang, and Andy Zou. A framework for few-shot language model evaluation, September 2021a. URL `https://doi.org/10.5281/zenodo.5371628`.

Tianyu Gao, Adam Fisch, and Danqi Chen. Making pre-trained language models better few-shot learners. *arXiv preprint arXiv:2012.15723*, 2020.

Tianyu Gao, Xingcheng Yao, and Danqi Chen. Simcse: Simple contrastive learning of sentence embeddings. *arXiv preprint arXiv:2104.08821*, 2021b.

Jake Gipple. The volume of n-balls. *Rose-Hulman Undergraduate Mathematics Journal*, 15(1):14, 2014.

Suchin Gururangan, Ana Marasović, Swabha Swayamdipta, Kyle Lo, Iz Beltagy, Doug Downey, and Noah A Smith. Don't stop pretraining: Adapt language models to domains and tasks. *arXiv preprint arXiv:2004.10964*, 2020.

Kelvin Guu, Kenton Lee, Zora Tung, Panupong Pasupat, and Ming-Wei Chang. Realm: Retrieval-augmented language model pre-training. *arXiv preprint arXiv:2002.08909*, 2020.

Wolfgang Hackbusch. *Tensor spaces and numerical tensor calculus*, volume 42. Springer Science & Business Media, 2012.

Samuel Humeau, Kurt Shuster, Marie-Anne Lachaux, and Jason Weston. Poly-encoders: Architectures and pre-training strategies for fast and accurate multi-sentence scoring. In *International Conference on Learning Representations*, 2020. URL `https://openreview.net/forum?id=SkxgnnNFvH`.

Gautier Izacard and Edouard Grave. Leveraging passage retrieval with generative models for open domain question answering. *arXiv preprint arXiv:2007.01282*, 2020.

Jeff Johnson, Matthijs Douze, and Hervé Jégou. Billion-scale similarity search with gpus. *IEEE Transactions on Big Data*, 2019.

Urvashi Khandelwal, Omer Levy, Dan Jurafsky, Luke Zettlemoyer, and Mike Lewis. Generalization through memorization: Nearest neighbor language models. *arXiv preprint arXiv:1911.00172*, 2019.

Tom Kwiatkowski, Jennimaria Palomaki, Olivia Redfield, Michael Collins, Ankur Parikh, Chris Alberti, Danielle Epstein, Illia Polosukhin, Jacob Devlin, Kenton Lee, et al. Natural questions: a benchmark for question answering research. *Transactions of the Association for Computational Linguistics*, 7:453–466, 2019.

Kenton Lee, Ming-Wei Chang, and Kristina Toutanova. Latent retrieval for weakly supervised open domain question answering. *arXiv preprint arXiv:1906.00300*, 2019.

Hector Levesque, Ernest Davis, and Leora Morgenstern. The winograd schema challenge. In *Thirteenth International Conference on the Principles of Knowledge Representation and Reasoning*, 2012.

Yoav Levine, Or Sharir, Alon Ziv, and Amnon Shashua. Benefits of depth for long-term memory of recurrent networks. *(ICLR 2018) International Conference on Learning Representations workshop*, 2018.

Yoav Levine, Noam Wies, Or Sharir, Hofit Bata, and Amnon Shashua. The depth-to-width interplay in self-attention. In *Advances in Neural Information Processing Systems*, 2020. URL https://papers.nips.cc/paper/2020/file/ff4dfdf5904e920ce52b48c1cef97829-Paper.pdf.

Mike Lewis, Marjan Ghazvininejad, Gargi Ghosh, Armen Aghajanyan, Sida Wang, and Luke Zettlemoyer. Pre-training via paraphrasing. *arXiv preprint arXiv:2006.15020*, 2020.

Yinhan Liu, Myle Ott, Naman Goyal, Jingfei Du, Mandar Joshi, Danqi Chen, Omer Levy, Mike Lewis, Luke Zettlemoyer, and Veselin Stoyanov. Roberta: A robustly optimized bert pretraining approach. *arXiv preprint arXiv:1907.11692*, 2019.

Roshan Rao, Jason Liu, Robert Verkuil, Joshua Meier, John F Canny, Pieter Abbeel, Tom Sercu, and Alexander Rives. Msa transformer. *bioRxiv*, 2021.

Nils Reimers and Iryna Gurevych. Sentence-bert: Sentence embeddings using siamese bert-networks. *arXiv preprint arXiv:1908.10084*, 2019.

Adam Roberts, Colin Raffel, and Noam Shazeer. How much knowledge can you pack into the parameters of a language model? *arXiv preprint arXiv:2002.08910*, 2020.

Timo Schick and Hinrich Schütze. It's not just size that matters: Small language models are also few-shot learners. *arXiv preprint arXiv:2009.07118*, 2020.

David J. Smith and Mavina K. Vamanamurthy. How small is a unit ball? *Mathematics Magazine*, 62 (2):101–107, 1989. doi: 10.1080/0025570X.1989.11977419. URL https://doi.org/10.1080/0025570X.1989.11977419.

Yi Tay, Mostafa Dehghani, Samira Abnar, Yikang Shen, Dara Bahri, Philip Pham, Jinfeng Rao, Liu Yang, Sebastian Ruder, and Donald Metzler. Long range arena: A benchmark for efficient transformers. *arXiv preprint arXiv:2011.04006*, 2020.

Yi Tay, Mostafa Dehghani, Jinfeng Rao, William Fedus, Samira Abnar, Hyung Won Chung, Sharan Narang, Dani Yogatama, Ashish Vaswani, and Donald Metzler. Scale efficiently: Insights from pre-training and fine-tuning transformers. *arXiv preprint arXiv:2109.10686*, 2021.

Nandan Thakur, Nils Reimers, Johannes Daxenberger, and Iryna Gurevych. Augmented sbert: Data augmentation method for improving bi-encoders for pairwise sentence scoring tasks. *arXiv preprint arXiv:2010.08240*, 2020.

Ashish Vaswani, Noam Shazeer, Niki Parmar, Jakob Uszkoreit, Llion Jones, Aidan N Gomez, Łukasz Kaiser, and Illia Polosukhin. Attention is all you need. In *Advances in neural information processing systems*, pp. 5998–6008, 2017.

Alex Wang, Amanpreet Singh, Julian Michael, Felix Hill, Omer Levy, and Samuel R Bowman. Glue: A multi-task benchmark and analysis platform for natural language understanding. *arXiv preprint arXiv:1804.07461*, 2018.

Noam Wies, Yoav Levine, Daniel Jannai, and Amnon Shashua. Which transformer architecture fits my data? a vocabulary bottleneck in self-attention. In Marina Meila and Tong Zhang (eds.), *Proceedings of the 38th International Conference on Machine Learning*, volume 139 of *Proceedings of Machine Learning Research*, pp. 11170–11181. PMLR, 18–24 Jul 2021. URL https://proceedings.mlr.press/v139/wies21a.html.

Adina Williams, Nikita Nangia, and Samuel R Bowman. A broad-coverage challenge corpus for sentence understanding through inference. *arXiv preprint arXiv:1704.05426*, 2017.

## A  PROOF OF COROLLARY 1

**Proposition 1.** *Let $S_1 = \left\{ w_1^j \right\}_{j=1}^N$ and $S_2 = \left\{ w_2^j \right\}_{j=1}^N$ be two sentences, $\mathbf{g}_{\mathcal{W}}^{i,L,d_x}$ the Transformer operation of $\mathbf{y}_{\text{in-context}}^{i,L,d_x}(S_1, S_2)$ and $M^V$ be a vocabulary embedding matrix. Then:*

$$\textit{sep-seq}\left( \mathbf{y}_{\text{in-context}}^{i,L,d_x}(S_1, S_2) \right) \leq \textit{sep}_{([N],[2N]\setminus[N])}\left( \mathbf{g}_{\mathcal{W}}^{i,L,d_x} \right)$$

*Proof.* Assume that $\text{sep}_{([N],[2N]\setminus[N])}\left( \mathbf{g}_{\mathcal{W}}^{i,L,d_x} \right) = R$, then by definition there exist $g_1, \ldots, g_R : \left(\mathbb{R}^{d_x}\right)^N \to \mathbb{R}$ and $g_1', \ldots, g_R' : \left(\mathbb{R}^{d_x}\right)^N \to \mathbb{R}$ such that for any $\left\{ \mathbf{x}^j \right\}_{j=1}^{2N}$,

$$\mathbf{g}_{\mathcal{W}}^{i,L,d_x}\left( \mathbf{x}^1, \ldots, \mathbf{x}^{2N} \right) = \sum_{r=1}^R g_r\left( \mathbf{x}^1, \ldots, \mathbf{x}^N \right) g_r'\left( \mathbf{x}^{N+1}, \ldots, \mathbf{x}^{2N} \right)$$

Now, given $\mathbf{a}, \mathbf{b} \in \mathbb{R}^{d_x}$, we can write:

$$\mathcal{Z}_{\mathbf{y}_{\text{in-context}}^{i,L,d_x}(S_1,S_2)}(\mathbf{a}, \mathbf{b}) = \sum_{r=1}^R g_r\left( E_{M^V}\left( w_1^1 \right) \odot \mathbf{a}, \ldots, E_{M^V}\left( w_1^N \right) \odot \mathbf{a} \right)$$
$$\cdot g_r'\left( E_{M^V}\left( w_2^1 \right) \odot \mathbf{b}, \ldots, E_{M^V}\left( w_2^N \right) \odot \mathbf{b} \right)$$

Clearly, this form of presenting $\mathcal{Z}_{\mathbf{y}_{\text{in-context}}^{i,L,d_x}(S_1,S_2)}(\mathbf{a}, \mathbf{b})$ is separable with respect to $(\mathbf{a}, \mathbf{b})$, and since it has $R$ summands, we can conclude that:

$$\text{sep-seq}\left( \mathbf{y}_{\text{in-context}}^{i,L,d_x}(S_1, S_2) \right) = \text{sep}_{(\mathbf{a},\mathbf{b})}\left( \mathcal{Z}_{\mathbf{y}_{\text{in-context}}^{i,L,d_x}(S_1,S_2)} \right) \leq R$$

$\square$

Corollary 1 now follows from an upper bound on $\text{sep}_{([N],[2N]\setminus[N])}\left( \mathbf{g}_{\mathcal{W}}^{i,L,d_x} \right)$ given in Levine et al. (2020).

## B  UPPER BOUND FOR THE $\varepsilon$-SEPARATION RANK

**Definition 1.** *For an expression that can be represented as a sum of some terms,*

$$f = \sum_{n=1}^N a_n$$

denote by $f^+$ the corresponding sum, but with each term replaced by its absolute value, that is:

$$f^+ := \sum_{n=1}^{N} |a_n|$$

and note that by the triangle inequality it holds that:

$$|f| \le f^+$$

**Theorem 3.** *Let $y_{\text{sequential}}^{(p,i),L,d_x,\eta}$ be be the $p \in [d_x]$ entry of the analyzed sequential representation defined in eq. 5. Assume that all learned parameters and all gradients are bounded: $\forall \theta \in \{\mathcal{W}, M^V\} : \Lambda_{\min} \le |\theta|, |\partial \mathcal{L}(S_1)/\partial \theta| \le \Lambda_{\max}$ for some $0 < \Lambda_{\min} \le \Lambda_{\max}$, $N < d_x$, $\eta \in (0,1]$, $\frac{2(1+\eta)d_x}{\eta} < 3^L$, $2(1+\eta) d_x^2 < 3^L$, In addition, assume that there exists $M \ge 0$ for which it holds that $\mathcal{Z}^+_{y_{sequential}^{p,i,H,L,d_x,\eta}(S_1,S_2)} < M$ on its domain. Then:*

$$\log \left[ \varepsilon\text{-seq-sep} \left( y_{\text{sequential}}^{(p,i),L,d_x,\eta} \right) \right] = \tilde{\mathcal{O}} \left( [L + 0.5 \log_3(\eta)] \cdot d_x \right)$$

*Proof.* Denote:

$$\mathcal{Z}^{(S_2)}_{\Theta_{t+1}(\mathbf{a},S_1;\eta)}(\mathbf{b}) := \mathcal{Z}_{y_{sequential}^{p,i,H,L,d_x,\eta}(S_1,S_2)}(\mathbf{a},\mathbf{b})$$

The proof outline is as follows:

We start by finding a representation of $\mathcal{Z}^{(S_2)}_{\Theta_{t+1}(\mathbf{a},S_1;\eta)}$ as a sum of terms, where each term is separable with respect to $(\mathbf{a},\mathbf{b})$. We then turn to finding a subset of these terms, denoted $G$, such that the sum of all terms in $G$ is an $\varepsilon$-approximation of $\mathcal{Z}^{(S_2)}_{\Theta_{t+1}(\mathbf{a},S_1;\eta)}$. Lastly, since it follows from the definition of the $\varepsilon$-separation rank and the construction of $G$ that $\varepsilon\text{-sep}_{(\mathbf{a},\mathbf{b})} \left( \mathcal{Z}_{y_{sequential}^{p,i,H,L,d_x,\eta}(S_1,S_2)} \right)$ is upper bounded by the cardinality of $G$ (which is the number of summands in the approximation), we find an upper bound to $|G|$, which is therefore an upper bound to $\varepsilon\text{-sep}_{(\mathbf{a},\mathbf{b})} \left( \mathcal{Z}_{y_{sequential}^{p,i,H,L,d_x,\eta}(S_1,S_2)} \right)$ as well, which by definition is equal to $\varepsilon\text{-seq-sep} \left( y_{\text{sequential}}^{p,i,H,L,d_x,\eta} \right)$.

STEP 1 — A SEPARABLE REPRESENTATION OF $\mathcal{Z}_{y_{\text{SEQUENTIAL}}^{p,i,H,L,d_x,\eta}(S_1,S_2)}$

Following Levine et al. (2020); Wies et al. (2021), $\mathcal{Z}^{(S_2)}_{\Theta_{t+1}(\mathbf{a},S_1;\eta)}(\mathbf{b})$ can be written as:

$$
\begin{aligned}
&\mathcal{Z}^{(S_2)}_{\Theta_{t+1}(\mathbf{a},S_1;\eta)}(\mathbf{b}) \\
&= \sum_{j_1,\ldots,j_{C(L)}=1}^{N} \sum_{h \in [H]^{[C(L)]}} \sum_{r_1,\ldots,r_{C(L)+1}=1}^{d_a} Q^{(0,h)}_{r_1,p} \\
&\quad \cdot \left( \prod_{c=1}^{C(L)+1} \left\langle P^{(c,h)}_{r_c}, \tilde{w^{j_c}} \right\rangle \right) \left( \prod_{c=1}^{C(L)} \left\langle Q^{(c,h)}_{r_{c+1}}, \tilde{w^{j_c}} \right\rangle \right)
\end{aligned}
\tag{13}
$$

where $\tilde{w^j} := g(\mathbf{b})^j + \eta f(\mathbf{a})^j \odot \mathbf{b}$ (this form follows from $g(\mathbf{b})^j := E_{M^V}\left(w_2^j\right) \odot \mathbf{b}$ is the entry-wise product of $\mathbf{b}$ with the embedding of $w_2^j$ prior to the $t$th training step, and $f(\mathbf{a})^j := -\frac{\partial \mathcal{L}(S_1;\mathbf{a})}{\partial \left(M_t^V\right)_{w_2^j}}$ is the gradient update performed to $w_2^j$'s embedding at time $t$), the $P^{(c,h)}_{r_c}$ and $Q^{(c,h)}_{r_{c+1}}$ terms are sums of products of the networks inner (i.e., non-embedding) weights which were also updated with respect

to $\mathcal{L}(S_1; \mathbf{a})$, and for convenience we denote $j_{C(L)+1} := i$ and $P^{(C(L)+1,h)} := P^{(0,h)}$.

$$= \sum_{j_1,\ldots,j_{C(L)}=1}^{N} \sum_{h\in[H]^{[C(L)]}} \sum_{r_1,\ldots,r_{C(L)+1}=1}^{d_a} Q_{r_1,p}^{(0,h)}$$

$$\cdot \left( \prod_{c=1}^{C(L)+1} \left\langle P_{r_c}^{(c,h)}, g(\mathbf{b})^{j_c} + \eta f(\mathbf{a})^{j_c} \odot \mathbf{b} \right\rangle \right)$$

$$\cdot \left( \prod_{c=1}^{C(L)} \left\langle Q_{r_{c+1}}^{(c,h)}, g(\mathbf{b})^{j_c} + \eta f(\mathbf{a})^{j_c} \odot \mathbf{b} \right\rangle \right)$$

Separating to vocab-gradient terms and vocab terms:

$$= \underbrace{\sum_{\substack{I_P\subseteq[C(L)+1]\\I_Q\subseteq[C(L)]}}}_{\text{Indices involving both } \mathbf{a} \text{ and } \mathbf{b}} \sum_{j_1,\ldots,j_{C(L)}=1}^{N} \sum_{h\in[H]^{[C(L)]}} \sum_{r_1,\ldots,r_{C(L)+1}=1}^{d_a} Q_{r_1,p}^{(0,h)}$$

$$\cdot \underbrace{\left( \prod_{c\in I_P} \left\langle P_{r_c}^{(c,h)}, \eta f(\mathbf{a})^{j_c} \odot \mathbf{b} \right\rangle \right) \left( \prod_{c\in I_Q} \left\langle Q_{r_{c+1}}^{(c,h)}, \eta f(\mathbf{a})^{j_c} \odot \mathbf{b} \right\rangle \right)}_{\text{Terms involving both } \mathbf{a} \text{ and } \mathbf{b}}$$

$$\underbrace{\left( \prod_{c\in[C(L)+1]\backslash I_P} \left\langle P_{r_c}^{(c,h)}, g(\mathbf{b})^{j_c} \right\rangle \right) \left( \prod_{c\in[C(L)]\backslash I_Q} \left\langle Q_{r_{c+1}}^{(c,h)}, g(\mathbf{b})^{j_c} \right\rangle \right)}_{\text{Terms involving just } \mathbf{b}}$$

Opening to indices:

$$= \sum_{\substack{I_P\subseteq[C(L)+1]\\I_Q\subseteq[C(L)]}} \sum_{j_1,\ldots,j_{C(L)}=1}^{N} \sum_{h\in[H]^{[C(L)]}} \sum_{r_1,\ldots,r_{C(L)+1}=1}^{d_a} Q_{r_1,p}^{(0,h)}$$

$$\sum_{\substack{\alpha_1,\ldots,\alpha_{C(L)+1}=1\\\beta_1,\ldots,\beta_{C(L)}}}^{d_x} \left( \prod_{c\in I_P} P_{r_c,\alpha_c}^{(c,h)} \eta f(\mathbf{a})_{\alpha_c}^{j_c} \mathbf{b}_{\alpha_c} \right) \left( \prod_{c\in I_Q} Q_{r_{c+1},\beta_c}^{(c,h)} \eta f(\mathbf{a})_{\beta_c}^{j_c} \mathbf{b}_{\beta_c} \right)$$

$$\left( \prod_{c\in[C(L)+1]\backslash I_P} P_{r_c,\alpha_c}^{(c,h)} g(\mathbf{b})_{\alpha_c}^{j_c} \right) \left( \prod_{c\in[C(L)]\backslash I_Q} Q_{r_{c+1},\beta_c}^{(c,h)} g(\mathbf{b})_{\beta_c}^{j_c} \right)$$

Separating to weights and variables:

$$= \sum_{\substack{\alpha_1,\ldots,\alpha_{C(L)+1}=1\\\beta_1,\ldots,\beta_{C(L)}}}^{d_x} \tau_{\alpha_1,\ldots,\beta_{C(L)}} \sum_{\substack{I_P\subseteq[C(L)+1]\\I_Q\subseteq[C(L)]}} \sum_{j_1,\ldots,j_{C(L)}=1}^{N}$$

$$\left( \prod_{c\in I_P} \eta f(\mathbf{a})_{\alpha_c}^{j_c} \mathbf{b}_{\alpha_c} \right) \left( \prod_{c\in I_Q} \eta f(\mathbf{a})_{\beta_c}^{j_c} \mathbf{b}_{\beta_c} \right)$$

$$\cdot \left( \prod_{c\in[C(L)+1]\backslash I_P} g(\mathbf{b})_{\alpha_c}^{j_c} \right) \left( \prod_{c\in[C(L)]\backslash I_Q} g(\mathbf{b})_{\beta_c}^{j_c} \right)$$

where:

$$\tau_{\alpha_1,\ldots,\beta_{C(L)}} := \sum_{h\in[H]^{[C(L)]}} \sum_{r_1,\ldots,r_{C(L)+1}=1}^{d_a} Q_{r_1,p}^{(0,h)}$$
$$\cdot \left[ \left(\prod_{c=1}^{C(L)+1} P_{r_c,\alpha_c}^{(c,h)}\right) \left(\prod_{c=1}^{C(L)} Q_{r_{c+1},\beta_c}^{(c,h)}\right)\right]$$

Compressing summation to count variable powers:

$$= \underbrace{\sum_{N_{\mathrm{A}}=0}^{2C(L)+1}}_{\substack{\text{Total power of } f(\mathbf{a})}} \underbrace{\sum_{\substack{p_1+\cdots+p_{d_x}=N_{\mathrm{A}}\\ n_1+\cdots+n_{d_x}=2C(L)+1-N_{\mathrm{A}}}}}_{\substack{\text{How many indices}\\ \text{are equal to each } \alpha\in[d_x]}}$$

$$\underbrace{\sum_{\substack{z_1+\cdots+z_N=N_{\mathrm{A}}\\ m_1+\cdots+m_N=2C(L)+1-N_{\mathrm{A}}\\ \forall j\in[N]\backslash\{i\}:\quad z_j+m_j\equiv 0 \mod 2\\ z_i+m_i\equiv 1 \mod 2}}}_{\substack{\text{How many indices}\\ \text{are equal to each } j\in[N]}} \underbrace{\sum_{\substack{0\le p_{1,1},\ldots,p_{d_x,N}\le N_{\mathrm{A}}\\ \forall\alpha\in[d_x]\ \sum_{j=1}^N p_{\alpha,j}=p_\alpha\\ \forall j\in[N]\ \sum_{\alpha=1}^{d_x} p_{\alpha,j}=z_j}}}_{\text{How to distribute the powers of } f(\mathbf{a})}$$

$$\underbrace{\sum_{\substack{0\le n_{1,1},\ldots,n_{d_x,N}\le 2C(L)+1-N_{\mathrm{A}}\\ \forall\alpha\in[d_x]\ \sum_{j=1}^N n_{\alpha,j}=n_\alpha\\ \forall j\in[N]\ \sum_{\alpha=1}^{d_x} n_{\alpha,j}=m_j}}}_{\text{How to distribute the powers of } g(\mathbf{b})} \lambda_{N_{\mathrm{A}},\boldsymbol{p},\boldsymbol{n}}$$

$$\cdot \left(\prod_{j=1}^N \prod_{\alpha=1}^{d_x} \left(\eta f(\mathbf{a})_\alpha^j \mathbf{b}_\alpha\right)^{p_{\alpha,j}}\right) \left(\prod_{j=1}^N \prod_{\alpha=1}^{d_x} \left(g(\mathbf{b})_\alpha^j\right)^{n_{\alpha,j}}\right)$$

where:

$$\lambda_{N_{\mathrm{A}},\boldsymbol{p},\boldsymbol{n}} := \sum_{\substack{I_P\subseteq[C(L)+1]\\ I_Q\subseteq[C(L)]\\ |I_P|+|I_Q|=N_{\mathrm{A}}}}$$

$$\sum_{\substack{\alpha_1,\ldots,\alpha_{C(L)+1}\\ \beta_1,\ldots,\beta_{C(L)}=1\\ \forall\kappa\in[d_x]\,|\{c\in I_P|\alpha_c=\kappa\}|+|\{c\in I_Q|\beta_c=\kappa\}|=p_\kappa\\ \kappa\in[d_x]\,|\{c\in[C(L)+1]\backslash I_P|\alpha_c=\kappa\}|+|\{c\in[C(L)]\backslash I_Q|\beta_c=\kappa\}|=n_\kappa}}^{d_x} \tau_{\alpha_1,\ldots,\beta_{C(L)}}$$

Pushing in summations on $N$, only the parity matters:

$$= \sum_{N_\mathrm{A}=0}^{2C(L)+1} \eta^{N_\mathrm{A}} \sum_{\substack{p_1+\cdots+p_{d_x}=N_\mathrm{A} \\ n_1+\cdots+n_{d_x}=2C(L)+1-N_\mathrm{A}}} \sum_{\substack{\boldsymbol{e}\in\{0,1\}^N \\ \sum_{j=1}^N e_j \equiv N_\mathrm{A} \mod 2 \\ \sum_{j=1}^N e_j \leq \min\{N_\mathrm{A},2C(L)-N_\mathrm{A}+2e_i\}}}$$

$$\underbrace{\lambda_{N_\mathrm{A},\boldsymbol{p},\boldsymbol{n}}}_{\text{Network's weights, function of }\mathbf{a}}$$

$$\cdot \underbrace{\left( \sum_{\substack{0\leq p_{1,1},\ldots,p_{d_x,N}\leq N_\mathrm{A} \\ z_1+\cdots+z_N=N_\mathrm{A} \\ \forall\alpha\in[d_x]\ \sum_{j=1}^N p_{\alpha,j}=p_\alpha \\ \forall j\in[N]\ \sum_{\alpha=1}^{d_x} p_{\alpha,j}=z_j \\ \forall j\in[N]\ z_j\equiv e_j \mod 2}} \prod_{j=1}^N \prod_{\alpha=1}^{d_x} \left( f\left(\mathbf{a}\right)_\alpha^j \right)^{p_{\alpha,j}} \right)}_{=:\phi_{A,\boldsymbol{p},\boldsymbol{e}},\text{ function of }\mathbf{a}}$$

$$\cdot \left( \prod_{\alpha=1}^{d_x} (\mathbf{b}_\alpha)^{p_\alpha} \right) \underbrace{\left( \sum_{\substack{0\leq n_{1,1},\ldots,n_{d_x,N}\leq 2C(L)+1-N_\mathrm{A} \\ m_1+\cdots+m_N=2C(L)+1-N_\mathrm{A} \\ \forall\alpha\in[d_x]\ \sum_{j=1}^N n_{\alpha,j}=n_\alpha \\ \forall j\in[N]\ \sum_{\alpha=1}^{d_x} n_{\alpha,j}=m_j \\ \forall j\in[N]\setminus\{i\}\ m_j\equiv e_j \mod 2 \\ m_i\equiv(1-e_i) \mod 2}} \prod_{j=1}^N \prod_{\alpha=1}^{d_x} \left( g\left(\mathbf{b}\right)_\alpha^j \right)^{n_{\alpha,j}} \right)}_{=:\Psi_{B,\boldsymbol{p},\boldsymbol{n},\boldsymbol{e}},\text{ function of }\mathbf{b}}$$

$$= \sum_{N_\mathrm{A}=0}^{2C(L)+1} \eta^{N_\mathrm{A}} \sum_{\substack{p_1+\cdots+p_{d_x}=N_\mathrm{A} \\ n_1+\cdots+n_{d_x}=2C(L)+1-N_\mathrm{A}}}$$

$$\sum_{\substack{\boldsymbol{e}\in\{0,1\}^N \\ \sum_{j=1}^N e_j\equiv N_\mathrm{A} \mod 2 \\ \sum_{j=1}^N e_j\leq\min\{N_\mathrm{A},2C(L)-N_\mathrm{A}+2e_i\}}} \lambda_{N_\mathrm{A},\boldsymbol{p},\boldsymbol{n}} \cdot \phi_{A,\boldsymbol{p},\boldsymbol{e}} \cdot \Psi_{B,\boldsymbol{p},\boldsymbol{n},\boldsymbol{e}}$$

where each summand is separable with respect to $(\mathbf{a},\mathbf{b})$.

STEP 2 — AN $\varepsilon$-APPROXIMATION OF $\mathcal{Z}_{y_\mathrm{SEQUENTIAL}^{p,i,H,L,d_x,\eta}(S_1,S_2)}$

Now that we have a representation of $\mathcal{Z}_{y_\mathrm{sequential}^{p,i,H,L,d_x,\eta}(S_1,S_2)}$ as a sum of $(\mathbf{a},\mathbf{b})$-separable terms, we turn to finding a subsum that can approximate $\mathcal{Z}_{y_\mathrm{sequential}^{p,i,H,L,d_x,\eta}(S_1,S_2)}$ up to an $\varepsilon$-precision.

First, let us define the set of all legal indices in the last sum:

$$D = D_{2C(L)+1,d_x,N} := \left\{ (N_\mathrm{A},\boldsymbol{p},\boldsymbol{n},\boldsymbol{e})\in\mathbb{N}^{2d_x+N+1} \ \middle| \ \begin{array}{c} 0\leq N_\mathrm{A}\leq 2C(L)+1 \\ \sum_{\alpha=1}^{d_x} p_\alpha=N_\mathrm{A} \\ \sum_{\alpha=1}^{d_x} n_\alpha=2C(L)+1-N_\mathrm{A} \\ \forall j\in[N],\ e_j\in\{0,1\} \\ \sum_{j=1}^N e_j\equiv N_\mathrm{A} \mod 2 \\ \sum_{j=1}^N e_j\leq\min\{N_\mathrm{A},2C(L)-N_\mathrm{A}+2e_i\} \end{array} \right\}$$

And for $G \subseteq D$, denote:

$$\mathcal{Z}_G\left(\mathbf{a}, \mathbf{b}\right) := \sum_{(N_A, \boldsymbol{p}, \boldsymbol{n}, \boldsymbol{e}) \in G} \eta^{N_A} \lambda_{N_A, \boldsymbol{p}, \boldsymbol{n}} \cdot \phi_{A, \boldsymbol{p}, \boldsymbol{e}} \cdot \Psi_{B, \boldsymbol{p}, \boldsymbol{n}, \boldsymbol{e}}$$

which is the sum of all terms with indices in $G$. Clearly, summing over all possible indices gives us the original expression:

$$\mathcal{Z}_D\left(\mathbf{a}, \mathbf{b}\right) = \mathcal{Z}_{y_{\text{sequential}}^{p, i, H, L, d_x, \eta}(S_1, S_2)}\left(\mathbf{a}, \mathbf{b}\right)$$

Given $\varepsilon > 0$, we wish to find a subset of the indices, $G \subseteq D$, such that the sum of all terms whose indices are in $G$ is an $\varepsilon$-approximation of $\mathcal{Z}_{y_{\text{sequential}}^{p, i, H, L, d_x, \eta}(S_1, S_2)}\left(\mathbf{a}, \mathbf{b}\right)$. That is, we are looking for $G \subseteq D$ such that for all $\mathbf{a}, \mathbf{b}$:

$$\left|\mathcal{Z}_D\left(\mathbf{a}, \mathbf{b}\right) - \mathcal{Z}_G\left(\mathbf{a}, \mathbf{b}\right)\right| = \left|\mathcal{Z}_{D \backslash G}\left(\mathbf{a}, \mathbf{b}\right)\right| \le \varepsilon$$

Note that since we assume that $\mathcal{Z}_D^+\left(\mathbf{a}, \mathbf{b}\right) = \mathcal{Z}_{y_{\text{sequential}}^{p, i, H, L, d_x, \eta}(S_1, S_2)}^+ < M$, we can get:

$$\left|\mathcal{Z}_{D \backslash G}\left(\mathbf{a}, \mathbf{b}\right)\right| = \frac{\left|\mathcal{Z}_{D \backslash G}\left(\mathbf{a}, \mathbf{b}\right)\right|}{\mathcal{Z}_G^+\left(\mathbf{a}, \mathbf{b}\right)} \mathcal{Z}_G^+\left(\mathbf{a}, \mathbf{b}\right) \le \frac{\mathcal{Z}_{D \backslash G}^+\left(\mathbf{a}, \mathbf{b}\right)}{\mathcal{Z}_G^+\left(\mathbf{a}, \mathbf{b}\right)} M$$

and it follows that it is enough for us to show that:

$$\frac{\mathcal{Z}_{D \backslash G}^+\left(\mathbf{a}, \mathbf{b}\right)}{\mathcal{Z}_G^+\left(\mathbf{a}, \mathbf{b}\right)} \le \frac{\varepsilon}{M}$$

which is equivalent showing that:

$$\frac{\mathcal{Z}_{D \backslash G}\left(\mathbf{a}, \mathbf{b}\right)}{\mathcal{Z}_G\left(\mathbf{a}, \mathbf{b}\right)} \le \frac{\varepsilon}{M} \tag{14}$$

under the assumption:

$$\forall \theta \in \Theta \qquad \theta, -\frac{\partial \mathcal{L}\left(S_1\right)}{\partial \theta}, f\left(\mathbf{a}\right)_\alpha^j, \mathbf{b}_\alpha, g\left(\mathbf{b}\right)_\alpha^j \in \left[\Lambda_{\min}, \Lambda_{\max}\right] \tag{15}$$

which we make going forward. This will ensure that $\mathcal{Z}_G$ is an $\varepsilon$-approximation of $\mathcal{Z}_{y_{\text{sequential}}^{p, i, H, L, d_x, \eta}(S_1, S_2)}\left(\mathbf{a}, \mathbf{b}\right)$.

Now, we assume that $\forall \theta \in \Theta : \theta, -\frac{\partial \mathcal{L}(S_1)}{\partial \theta} \in \left[\Lambda_{\min}, \Lambda_{\max}\right]$, and by Levine et al. (2020), the $P$s and $Q$s in eq. 13 are products of up to $L$ matrices, so each of their coordinates is bounded in $\left[\Lambda_{\min}^L, \Lambda_{\max}^L\right]$ and we assume without loss of generality that $\Lambda_{\min} \le 1 \le \Lambda_{\max}$ (otherwise we could have picked a smaller $\Lambda_{\min}$ and a larger $\Lambda_{\max}$). Then for each $(N_A, \boldsymbol{p}, \boldsymbol{n}, \boldsymbol{e}) \in D$ the following inequalities hold:

$$\tau_{\alpha_1, \ldots, \beta_{C(L)}} \le \sum_{h \in [H]^{[C(L)]}} \sum_{r_1, \ldots, r_{C(L)+1}=1}^{d_a} \Lambda_{\max} \left[\left(\prod_{c=1}^{C(L)+1} \Lambda_{\max}\right)\left(\prod_{c=1}^{C(L)} \Lambda_{\max}\right)\right]$$
$$= d_x^{C(L)} d_a \Lambda_{\max}^{2C(L)+2}$$

$$\lambda_{N_A, \boldsymbol{p}, \boldsymbol{n}}$$

$$\le \sum_{\substack{I_P \subseteq [C(L)+1] \\ I_Q \subseteq [C(L)] \\ |I_P| + |I_Q| = N_A}} \sum_{\substack{\alpha_1, \ldots, \alpha_{C(L)+1}=1 \\ \beta_1, \ldots, \beta_{C(L)} \\ \forall \kappa \in [d_x] \, |\{c \in I_P | \alpha_c = \kappa\}| + |\{c \in I_Q | \beta_c = \kappa\}| = p_\kappa \\ \kappa \in [d_x] \, |\{c \in [C(L)+1] \backslash I_P | \alpha_c = \kappa\}| + |\{c \in [C(L)] \backslash I_Q | \beta_c = \kappa\}| = n_\kappa}}^{d_x} d_x^{C(L)} d_a \Lambda_{\max}^{2C(L)+2}$$

$$= \binom{2C(L)+1}{N_A} \binom{N_A}{p_1, \ldots, p_{d_x}} \binom{2C(L)+1-N_A}{n_1, \ldots, n_{d_x}} d_x^{C(L)} d_a \Lambda_{\max}^{2C(L)+2}$$

$$\phi_{A,\boldsymbol{p},\boldsymbol{e}} \le \sum_{\substack{0 \le p_{1,1},\ldots,p_{d_x,N} \le N_A \\ z_1+\cdots+z_N=N_A \\ \forall \alpha \in [d_x]\ \sum_{j=1}^{N} p_{\alpha,j}=p_\alpha \\ \forall j \in [N]\ \sum_{\alpha=1}^{d_x} p_{\alpha,j}=z_j \\ \forall j \in [N]\ z_j \equiv e_j \mod 2}} \prod_{j=1}^{N} \prod_{\alpha=1}^{d_x} \Lambda_{\max}^{p_{\alpha,j}}$$

$$= \left( \sum_{\substack{0 \le p_{1,1},\ldots,p_{d_x,N} \le N_A \\ z_1+\cdots+z_N=N_A \\ \forall \alpha \in [d_x]\ \sum_{j=1}^{N} p_{\alpha,j}=p_\alpha \\ \forall j \in [N]\ \sum_{\alpha=1}^{d_x} p_{\alpha,j}=z_j \\ \forall j \in [N]\ z_j \equiv e_j \mod 2}} 1 \right) \Lambda_{\max}^{N_A}$$

$$\Psi_{B,\boldsymbol{p},\boldsymbol{n},\boldsymbol{e}} \le \left( \prod_{\alpha=1}^{d_x} \Lambda_{\max}^{p_\alpha} \right) \sum_{\substack{0 \le n_{1,1},\ldots,n_{d_x,N} \le 2C(L)+1-N_A \\ m_1+\cdots+m_N=2C(L)+1-N_A \\ \forall \alpha \in [d_x]\ \sum_{j=1}^{N} n_{\alpha,j}=n_\alpha \\ \forall j \in [N]\ \sum_{\alpha=1}^{d_x} n_{\alpha,j}=m_j \\ \forall j \in [N]\setminus\{i\}\ m_j \equiv e_j \mod 2 \\ m_i \equiv (1-e_i) \mod 2}} \prod_{j=1}^{N} \prod_{\alpha=1}^{d_x} \Lambda_{\max}^{n_{\alpha,j}}$$

$$= \left( \sum_{\substack{0 \le n_{1,1},\ldots,n_{d_x,N} \le 2C(L)+1-N_A \\ m_1+\cdots+m_N=2C(L)+1-N_A \\ \forall \alpha \in [d_x]\ \sum_{j=1}^{N} n_{\alpha,j}=n_\alpha \\ \forall j \in [N]\ \sum_{\alpha=1}^{d_x} n_{\alpha,j}=m_j \\ \forall j \in [N]\setminus\{i\}\ m_j \equiv e_j \mod 2 \\ m_i \equiv (1-e_i) \mod 2}} 1 \right) \Lambda_{\max}^{2C(L)+1}$$

and therefore:

$$\lambda_{N_A,\boldsymbol{p},\boldsymbol{n}} \cdot \phi_{A,\boldsymbol{p},\boldsymbol{e}} \cdot \Psi_{B,\boldsymbol{p},\boldsymbol{n},\boldsymbol{e}}$$

$$\le \left( \sum_{\substack{0 \le p_{1,1},\ldots,p_{d_x,N} \le N_A \\ z_1+\cdots+z_N=N_A \\ \forall \alpha \in [d_x]\ \sum_{j=1}^{N} p_{\alpha,j}=p_\alpha \\ \forall j \in [N]\ \sum_{\alpha=1}^{d_x} p_{\alpha,j}=z_j \\ \forall j \in [N]\ z_j \equiv e_j \mod 2}} 1 \right) \left( \sum_{\substack{0 \le n_{1,1},\ldots,n_{d_x,N} \le 2C(L)+1-N_A \\ m_1+\cdots+m_N=2C(L)+1-N_A \\ \forall \alpha \in [d_x]\ \sum_{j=1}^{N} n_{\alpha,j}=n_\alpha \\ \forall j \in [N]\ \sum_{\alpha=1}^{d_x} n_{\alpha,j}=m_j \\ \forall j \in [N]\setminus\{i\}\ m_j \equiv e_j \mod 2 \\ m_i \equiv (1-e_i) \mod 2}} 1 \right)$$

$$\cdot \binom{2C(L)+1}{N_A} \binom{N_A}{p_1,\ldots,p_{d_x}} \binom{2C(L)+1-N_A}{n_1,\ldots,n_{d_x}} d_x^{C(L)} d_a \Lambda_{\max}^{L(2C(L)+2)+2C(L)+1+N_A}$$

Relaxing the parity constraints inside the brackets and recalling that $\Lambda_{\max} \geq 1$ gives us an upper bound:

$$\leq \left( \prod_{\alpha=1}^{d_x} \left( \binom{N}{p_\alpha} \right) \right) \cdot \left( \prod_{\alpha=1}^{d_x} \left( \binom{N}{n_\alpha} \right) \right)$$
$$\cdot \binom{2C(L)+1}{N_A} \binom{N_A}{p_1, \ldots, p_{d_x}} \binom{2C(L)+1-N_A}{n_1, \ldots, n_{d_x}} d_x^{C(L)} d_a \Lambda_{\max}^{(L+2)(2C(L)+2)}$$

which we can further bound using lemmas 3 and 4 in Levine et al. (2020) until we are left with:

$$\leq \left( \frac{e(2d_x N + 2C(L) + 1)}{d_x N} \right)^{2d_x N}$$
$$\cdot \binom{2C(L)+1}{N_A} \binom{N_A}{p_1, \ldots, p_{d_x}} \binom{2C(L)+1-N_A}{n_1, \ldots, n_{d_x}} d_x^{C(L)} d_a \Lambda_{\max}^{(L+2)(2C(L)+2)}$$

On the other hand:

$$\lambda_{N_A, \boldsymbol{p}, \boldsymbol{n}} \cdot \phi_{A, \boldsymbol{p}, \boldsymbol{e}} \cdot \Psi_{B, \boldsymbol{p}, \boldsymbol{n}, \boldsymbol{e}}$$
$$\geq \binom{2C(L)+1}{N_A} \binom{N_A}{p_1, \ldots, p_{d_x}} \binom{2C(L)+1-N_A}{n_1, \ldots, n_{d_x}}$$
$$\cdot d_x^{C(L)} d_a \Lambda_{\min}^{L(2C(L)+2)}$$

$$\cdot \left( \sum_{\substack{0 \leq p_{1,1}, \ldots, p_{d_x, N} \leq N_A \\ z_1 + \cdots + z_N = N_A \\ \forall \alpha \in [d_x] \sum_{j=1}^{N} p_{\alpha, j} = p_\alpha \\ \forall j \in [N] \sum_{\alpha=1}^{d_x} p_{\alpha, j} = z_j \\ \forall j \in [N] \, z_j \equiv e_j \mod 2}} 1 \right) \Lambda_{\min}^{N_A}$$

$$\cdot \left( \sum_{\substack{0 \leq n_{1,1}, \ldots, n_{d_x, N} \leq 2C(L)+1-N_A \\ m_1 + \cdots + m_N = 2C(L)+1-N_A \\ \forall \alpha \in [d_x] \sum_{j=1}^{N} n_{\alpha, j} = n_\alpha \\ \forall j \in [N] \sum_{\alpha=1}^{d_x} n_{\alpha, j} = m_j \\ \forall j \in [N] \setminus \{i\} \, m_j \equiv e_j \mod 2 \\ m_i \equiv (1 - e_i) \mod 2}} 1 \right) \Lambda_{\min}^{2C(L)+1}$$
$$\geq \binom{2C(L)+1}{N_A} \binom{N_A}{p_1, \ldots, p_{d_x}} \binom{2C(L)+1-N_A}{n_1, \ldots, n_{d_x}} d_x^{C(L)} d_a \Lambda_{\min}^{L(2C(L)+2)+2C(L)+1+N_A}$$
$$\geq \binom{2C(L)+1}{N_A} \binom{N_A}{p_1, \ldots, p_{d_x}} \binom{2C(L)+1-N_A}{n_1, \ldots, n_{d_x}} d_x^{C(L)} d_a \Lambda_{\min}^{(L+2)(2C(L)+2)}$$

Combining the upper and lower bound for $\lambda_{N_A, \boldsymbol{p}, \boldsymbol{n}} \cdot \phi_{A, \boldsymbol{p}, \boldsymbol{e}} \cdot \Psi_{B, \boldsymbol{p}, \boldsymbol{n}, \boldsymbol{e}}$, we get that for all $G \subseteq D$:

$$\frac{\mathcal{Z}_{D \setminus G}(\mathbf{a}, \mathbf{b})}{\mathcal{Z}_G(\mathbf{a}, \mathbf{b})}$$
$$\leq \left( \frac{e(2d_x N + 2C(L) + 1)}{d_x N} \right)^{2d_x N} \cdot \left( \frac{\Lambda_{\max}}{\Lambda_{\min}} \right)^{(L+2)(2C(L)+2)}$$
$$\cdot \frac{\sum_{(N_A, \boldsymbol{p}, \boldsymbol{n}, \boldsymbol{e}) \in D \setminus G} \eta^{N_A} \binom{2C(L)+1}{N_A} \binom{N_A}{p_1, \ldots, p_{d_x}} \binom{2C(L)+1-N_A}{n_1, \ldots, n_{d_x}}}{\sum_{(N_A, \boldsymbol{p}, \boldsymbol{n}, \boldsymbol{e}) \in G} \eta^{N_A} \binom{2C(L)+1}{N_A} \binom{N_A}{p_1, \ldots, p_{d_x}} \binom{2C(L)+1-N_A}{n_1, \ldots, n_{d_x}}}$$

so in order to show that (14) holds, it suffices to show that:

$$\frac{\sum_{(N_A,\boldsymbol{p},\boldsymbol{n},\boldsymbol{e})\in D\setminus G}\eta^{N_A}\binom{2C(L)+1}{N_A}\binom{N_A}{p_1,\ldots,p_{d_x}}\binom{2C(L)+1-N_A}{n_1,\ldots,n_{d_x}}}{\sum_{(N_A,\boldsymbol{p},\boldsymbol{n},\boldsymbol{e})\in G}\eta^{N_A}\binom{2C(L)+1}{N_A}\binom{N_A}{p_1,\ldots,p_{d_x}}\binom{2C(L)+1-N_A}{n_1,\ldots,n_{d_x}}}$$

$$\leq\left(\frac{d_x N}{e\left(2d_x N+2C\left(L\right)+1\right)}\right)^{2d_x N}\cdot\left(\frac{\Lambda_{\min}}{\Lambda_{\max}}\right)^{(L+2)(2C(L)+2)}\cdot\frac{\varepsilon}{M}$$

Now, we can limit ourselves to subsets $G\subseteq D$ of the form:

$$G\left(T\right)=\left\{(N_A,\boldsymbol{p},\boldsymbol{n},\boldsymbol{e})\in D\left|\eta^{N_A}\binom{2C(L)+1}{N_A}\binom{N_A}{p_1,\ldots,p_{d_x}}\binom{2C(L)+1-N_A}{n_1,\ldots,n_{d_x}}\geq T\right.\right\}$$

and in this case we get that:

$$\frac{\sum_{(N_A,\boldsymbol{p},\boldsymbol{n},\boldsymbol{e})\in D\setminus G(T)}\eta^{N_A}\binom{2C(L)+1}{N_A}\binom{N_A}{p_1,\ldots,p_{d_x}}\binom{2C(L)+1-N_A}{n_1,\ldots,n_{d_x}}}{\sum_{(N_A,\boldsymbol{p},\boldsymbol{n},\boldsymbol{e})\in G(T)}\eta^{N_A}\binom{2C(L)+1}{N_A}\binom{N_A}{p_1,\ldots,p_{d_x}}\binom{2C(L)+1-N_A}{n_1,\ldots,n_{d_x}}}$$

$$\leq\frac{\sum_{(N_A,\boldsymbol{p},\boldsymbol{n},\boldsymbol{e})\in D\setminus G(T)}T}{\sum_{(N_A,\boldsymbol{p},\boldsymbol{n},\boldsymbol{e})\in G(T)}T}$$

$$=\frac{|D|-|G\left(T\right)|}{|G\left(T\right)|}$$

Let us define:

$$\tilde{D}=\tilde{D}_{2C(L)+1,d_x}:=\left\{(N_A,\boldsymbol{p},\boldsymbol{n})\in\mathbb{N}^{2d_x+1}\left|\begin{array}{c}0\leq N_A\leq 2C(L)+1\\\sum_{\alpha=1}^{d_x}p_\alpha=N_A\\\sum_{\alpha=1}^{d_x}n_\alpha=2C(L)+1-N_A\end{array}\right.\right\}$$

$$\tilde{G}\left(T\right):=\left\{(N_A,\boldsymbol{p},\boldsymbol{n})\in\tilde{D}\left|\eta^{N_A}\binom{2C(L)+1}{N_A}\binom{N_A}{p_1,\ldots,p_{d_x}}\binom{2C(L)+1-N_A}{n_1,\ldots,n_{d_x}}\geq T\right.\right\}$$

and note that:

$$\left|\tilde{G}\left(T\right)\right|\leq|G\left(T\right)|\leq 2^N\cdot\left|\tilde{G}\left(T\right)\right|$$

$$\left|\tilde{D}\right|\leq|D|\leq 2^N\cdot\left|\tilde{D}\right|$$

and also:

$$\left|\tilde{D}\right|=\left(\!\!\binom{2d_x}{2C\left(L\right)+1}\!\!\right)$$

$$\leq\left(\frac{e\left(2d_x+2C\left(L\right)+1\right)}{d_x}\right)^{2d_x}$$

where the inequality is due to lemma 3 in Levine et al. (2020).

Hence:

$$\frac{|D|-|G\left(T\right)|}{|G\left(T\right)|}\leq\frac{2^N\cdot\left(\frac{e(2d_x+2C(L)+1)}{d_x}\right)^{2d_x}-\left|\tilde{G}\left(T\right)\right|}{\left|\tilde{G}\left(T\right)\right|}$$

And after rearranging we get that for each $T\geq 0$ such that:

$$\left|\tilde{G}\left(T\right)\right|\geq\frac{2^N\cdot\left(e\left(2d_x+2C\left(L\right)+1\right)\right)^{2d_x}}{\left(1+\left(\frac{d_x N}{e(2d_x N+2C(L)+1)}\right)^{2d_x N}\cdot\left(\frac{\Lambda_{\min}}{\Lambda_{\max}}\right)^{(L+2)(2C(L)+2)}\cdot\frac{\varepsilon}{M}\right)d_x^{2d_x}} \tag{16}$$

In order for $\mathcal{Z}_{G(T)}\left(\mathbf{a},\mathbf{b}\right)$ to be an $\varepsilon$-approximation of $\mathcal{Z}_{y_{\text{sequential}}^{p,i,H,L,d_x,\eta}(S_1,S_2)}\left(\mathbf{a},\mathbf{b}\right)$.

STEP 3 — AN UPPER BOUND TO $\varepsilon$-SEP$_{(\mathbf{a},\mathbf{b})}\left(\mathcal{Z}_{y^{p,i,H,L,d_x,\eta}_{\text{SEQUENTIAL}}(S_1,S_2)}\right)$

In the last step we have found a condition on subsets of indices, such that summing over any subset who meets this condition will yield an $\varepsilon$-approximation of $\mathcal{Z}_{y^{p,i,H,L,d_x,\eta}_{\text{sequential}}(S_1,S_2)}$. We will now find a specific subset who meets this condition, and use it in order to bound $\varepsilon$-sep$_{(\mathbf{a},\mathbf{b})}\left(\mathcal{Z}_{y^{p,i,H,L,d_x,\eta}_{\text{sequential}}(S_1,S_2)}\right)$ from above.

We will focus our attention on $T$s of the form:

$$T\left(s\right)=s\cdot\eta^{\frac{\eta(2C(L)+1)}{1+\eta}}\left(\frac{2C\left(L\right)+1}{\frac{\eta(2C(L)+1)}{1+\eta}}\right)$$

$$\cdot\left(\frac{\frac{\eta(2C(L)+1)}{1+\eta}}{\frac{\eta(2C(L)+1)}{(1+\eta)d_x}},\ldots,\frac{\eta(2C(L)+1)}{(1+\eta)d_x}\right)\left(\frac{\frac{2C(L)+1}{1+\eta}}{\frac{2C(L)+1}{(1+\eta)d_x}},\ldots,\frac{2C(L)+1}{(1+\eta)d_x}\right)$$

for some $s\in\left(0,e^{-1.5}\right]$ which we'll determine later.

By lemma 9 we have that:

$$\left|\tilde{G}\left(T\left(s\right)\right)\right|\geq\frac{1}{d_x\sqrt{\pi}}\left(\frac{\pi e\left(2C\left(L\right)+1\right)}{2d_x^2\left(1+\eta\right)}\right)^{\frac{d_x-1}{2}}\cdot\left(\ln\left(s^{-1}\right)\right)^{\frac{d_x-1}{2}}$$

so we can choose:

$$s^*=\exp\left(-\left(\frac{2^{2N}\left(e(2d_x+2C(L)+1)\right)^{4d_x}\pi(2(1+\eta))^{d_x-1}}{\left(1+\left(\frac{d_xN}{e(2d_xN+2C(L)+1)}\right)^{2d_xN}\left(\frac{\Lambda_{\min}}{\Lambda_{\max}}\right)^{(L+2)(2C(L)+2)}\frac{\varepsilon}{M}\right)^2 d_x^{2d_x}\left(\pi e(2C(L)+1)\right)^{d_x-1}}\right)^{\frac{1}{d_x-1}}\right)$$

and since this upholds $s^*<e^{-1.5}$, we get that (16) indeed holds for $\tilde{G}\left(T\left(s^*\right)\right)$, and therefore that $\mathcal{Z}_{G(T(s^*))}$ is an $\varepsilon$-approximation of $\mathcal{Z}_{y^{p,i,H,L,d_x,\eta}_{\text{sequential}}(S_1,S_2)}$.

Now, from lemma 8 we get:

$$\left|\tilde{G}\left(T\left(s^*\right)\right)\right|\leq\frac{2\left(2C\left(L\right)+1\right)}{\left(d_x-1\right)^2}\cdot\left(\frac{25\sqrt{\eta}}{\left(d_x-1\right)}\right)^{d_x-1}$$

$$\cdot\frac{2^{2N}\cdot\left(e\left(2d_x+2C\left(L\right)+1\right)\right)^{4d_x}}{\left(1+\left(\frac{d_xN}{e(2d_xN+2C(L)+1)}\right)^{2d_xN}\cdot\left(\frac{\Lambda_{\min}}{\Lambda_{\max}}\right)^{(L+2)(2C(L)+2)}\cdot\frac{\varepsilon}{M}\right)^2\cdot d_x^{2d_x}}$$

and therefore:

$$\varepsilon\text{-seq-sep}\left(y^{p,i,H,L,d_x,\eta}_{\text{sequential}}\right)$$

$$=\varepsilon\text{-sep}_{(\mathbf{a},\mathbf{b})}\left(\mathcal{Z}_{y^{p,i,H,L,d_x,\eta}_{\text{sequential}}(S_1,S_2)}\right)$$

$$\leq\left|G\left(T\left(s^*\right)\right)\right|$$

$$\leq 2^N\cdot\left|\tilde{G}\left(T\left(s^*\right)\right)\right|$$

$$\leq\frac{2\left(2C\left(L\right)+1\right)}{\left(d_x-1\right)^2}\cdot\left(\frac{25\sqrt{\eta}}{\left(d_x-1\right)}\right)^{d_x-1}$$

$$\cdot\frac{2^{3N}\cdot\left(e\left(2d_x+2C\left(L\right)+1\right)\right)^{4d_x}}{\left(1+\left(\frac{d_xN}{e(2d_xN+2C(L)+1)}\right)^{2d_xN}\cdot\left(\frac{\Lambda_{\min}}{\Lambda_{\max}}\right)^{(L+2)(2C(L)+2)}\cdot\frac{\varepsilon}{M}\right)^2\cdot d_x^{2d_x}}$$

$\square$

## B.1 Lemmas for estimating the number of coefficients

**Remark 1.** *For brevity and clarity we will use expressions of the form $\left(\begin{smallmatrix} K \\ \frac{K}{M},\dots,\frac{K}{M} \end{smallmatrix}\right)$, regardless of whether $K$ is divisible by $M$ or not. For the latter case, this expression should actually be:*

$$\left( \begin{matrix} K \\ \lfloor \frac{K}{M} \rfloor, \dots, \lfloor \frac{K}{M} \rfloor, \underbrace{\lfloor \frac{K}{M} \rfloor + 1, \dots, \lfloor \frac{K}{M} \rfloor + 1}_{(K \mod M) \text{ times}} \end{matrix} \right)$$

*expressions of the form $\left(\begin{smallmatrix} K \\ \frac{K}{M},\dots,\frac{K}{M}+1 \end{smallmatrix}\right)$ should be read as:*

$$\left( \begin{matrix} K \\ \lfloor \frac{K}{M} \rfloor, \dots, \lfloor \frac{K}{M} \rfloor, \underbrace{\lfloor \frac{K}{M} \rfloor + 1, \dots, \lfloor \frac{K}{M} \rfloor + 1}_{(K \mod M)+1 \text{ times}} \end{matrix} \right)$$

*and expressions of the form $\left(\begin{smallmatrix} K \\ \frac{K}{M},\dots,\frac{K}{M}-1 \end{smallmatrix}\right)$ should be read as:*

$$\begin{cases} \left( \begin{matrix} K \\ \frac{K}{M},\dots,\frac{K}{M}, \underbrace{\frac{K}{M}-1}_{\text{just once}} \end{matrix} \right) & \text{if } K \mod M \equiv 0 \\[2em] \left( \begin{matrix} K \\ \lfloor \frac{K}{M} \rfloor, \dots, \lfloor \frac{K}{M} \rfloor, \underbrace{\lfloor \frac{K}{M} \rfloor-1, \dots, \lfloor \frac{K}{M} \rfloor-1}_{(K \mod M)-1 \text{ times}} \end{matrix} \right) & \text{otherwise} \end{cases}$$

**Lemma 1.** *For all $K, M \in \mathbb{N}$, the maximal value of $\left(\begin{smallmatrix} K \\ a_1,\dots,a_M \end{smallmatrix}\right)$ is achieved when $\forall j_1, j_2 \in [M]$ it holds that $|a_{j_1} - a_{j_2}| \leq 1$.*

*Proof.* Let $a_1, \dots, a_M$ be a sequence of non-negative integers such that $a_1 + \dots + a_M = K$ and $\left(\begin{smallmatrix} K \\ a_1,\dots,a_M \end{smallmatrix}\right)$ is maximal. Assume towards a contradiction there exist $j_1, j_2 \in [M]$ such that $a_{j_1} - a_{j_2} > 1 \iff \frac{a_{j_2}+1}{a_{j_1}} < 1$, then:

$$\begin{aligned} \left( \begin{matrix} K \\ a_1, \dots, a_M \end{matrix} \right) &= \frac{K!}{\prod_{i=1}^{M} a_i!} \\ &= \frac{K!}{\left( \prod_{i \in [M] \setminus \{j_1, j_2\}} a_i! \right) \cdot (a_{j_1} - 1)! \cdot (a_{j_2} + 1)!} \cdot \frac{a_{j_2} + 1}{a_{j_1}} \\ &< \frac{K!}{\left( \prod_{i \in [M] \setminus \{j_1, j_2\}} a_i! \right) \cdot (a_{j_1} - 1)! \cdot (a_{j_2} + 1)!} \\ &= \left( \begin{matrix} K \\ a_1, \dots, a_{j_1} - 1, \dots, a_{j_2} + 1, \dots, a_M \end{matrix} \right) \end{aligned}$$

in contrary to the maximality of $\left(\begin{smallmatrix} K \\ a_1,\dots,a_M \end{smallmatrix}\right)$. Therefore, $\forall j_1, j_2 \in [M], |a_{j_1} - a_{j_2}| \leq 1$. $\square$

**Lemma 2.** *Let $K, M$ be two fixed natural numbers, $\eta \in (0, 1]$ and denote*

$$S(n) := \binom{K}{n} \eta^n \left( \begin{matrix} n \\ \frac{n}{M}, \dots, \frac{n}{M} \end{matrix} \right) \left( \begin{matrix} K-n \\ \frac{K-n}{M}, \dots, \frac{K-n}{M} \end{matrix} \right)$$

*Then for all $n \in [K] \cup \{0\}$:*

$$\begin{aligned} S(n) &= \left( \prod_{j=0}^{\lfloor \frac{n}{M} \rfloor - 1} \left( \frac{\eta(K - jM)}{(j+1)M} \right)^M \right) \\ &= \cdot \left( \frac{\eta(K - \lfloor \frac{n}{M} \rfloor M)}{(\lfloor \frac{n}{M} \rfloor + 1)M} \right)^{n \mod M} \left( \begin{matrix} K \\ \frac{K}{M}, \dots, \frac{K}{M} \end{matrix} \right) \end{aligned} \tag{17}$$

*Proof.* We will prove by induction.

- **Base case:** $n = 0$.

$$S(0) = \binom{K}{0} \eta^0 \binom{n}{\frac{0}{M}, \ldots, \frac{0}{M}} \binom{K}{\frac{K-0}{M}, \ldots, \frac{K-0}{M}}$$

$$= \left( \prod_{j=0}^{\lfloor \frac{0}{M} \rfloor - 1} \left( \frac{\eta(K - jM)}{(j+1)M} \right)^M \right)$$

$$\cdot \left( \frac{\eta \left( K - \lfloor \frac{0}{M} \rfloor M \right)}{\left( \lfloor \frac{0}{M} \rfloor + 1 \right) M} \right)^{0 \mod M} \cdot \binom{K}{\frac{K}{M}, \ldots, \frac{K}{M}}$$

where the second equality is due to the fact that:

$$\left( \prod_{j=0}^{\lfloor \frac{0}{M} \rfloor - 1} \left( \frac{\eta(K - jM)}{(j+1)M} \right)^M \right) = 1$$

as an empty product, and:

$$\left( \frac{\eta(K - 0 \cdot M)}{1 \cdot M} \right)^{0 \mod M} = 1$$

Therefore, (17) is true for $n = 0$.

- **Induction step:** Let $n \geq 0$ such that (17) holds for $n$.

$$S(n+1) = \binom{K}{n+1} \eta^{n+1} \binom{n+1}{\frac{n}{M}, \ldots, \frac{n}{M} + 1} \binom{K - n - 1}{\frac{K-n}{M}, \ldots, \frac{K-n}{M} - 1}$$

$$= \frac{\eta(K-n)}{\left( \frac{n}{M} + 1 \right) M} \cdot S(n)$$

$$= \frac{\eta(K-n)}{\left( \frac{n}{M} + 1 \right) M} \cdot \left( \prod_{j=0}^{\lfloor \frac{n}{M} \rfloor - 1} \left( \frac{\eta(K - jM)}{(j+1)M} \right)^M \right)$$

$$\cdot \left( \frac{\eta \left( K - \lfloor \frac{n}{M} \rfloor M \right)}{\left( \lfloor \frac{n}{M} \rfloor + 1 \right) M} \right)^{n \mod M} \binom{K}{\frac{K}{M}, \ldots, \frac{K}{M}}$$

$$= \left( \prod_{j=0}^{\lfloor \frac{n+1}{M} \rfloor - 1} \left( \frac{\eta(K - jM)}{(j+1)M} \right)^M \right)$$

$$\cdot \left( \frac{\eta \left( K - \lfloor \frac{n+1}{M} \rfloor M \right)}{\left( \lfloor \frac{n+1}{M} \rfloor + 1 \right) M} \right)^{(n+1) \mod M} \binom{K}{\frac{K}{M}, \ldots, \frac{K}{M}}$$

Thus, (17) holds for $n + 1$, and the proof of the induction step is complete. Hence, by induction, (17) is correct for all $n \in [K] \cup \{0\}$.

$\square$

**Lemma 3.** *Let $K, M$ be two fixed natural numbers, and $\eta \in (0, 1]$. Then the maximum of:*

$$\binom{K}{n} \eta^n \binom{n}{a_1, \ldots, a_M} \binom{K - n}{b_1, \ldots, b_M}$$

*is achieved when:*

$$n = \left( \left\lfloor \frac{\eta K - M}{M(1+\eta)} \right\rfloor + 1 \right) M \simeq \frac{\eta K}{1+\eta}$$

*and for all* $i \in [M]$, $\left| a_i - \frac{n}{M} \right| \leq 1$ *and* $\left| b_i - \frac{K-n}{M} \right| \leq 1$.

*Proof.* From lemma 1 we know that a multinumial coefficient reaches its maximum when the sum is evenly distributed between all indices. Since the $a_i$s and $b_i$ can be chosen independently of each other given $K$ and $n$, we may assume without loss of generality that no matter the value of $n$, for all $i \in [M]$, it holds that $\left| a_i - \frac{n}{M} \right| \leq 1$ and $\left| b_i - \frac{K-n}{M} \right| \leq 1$.

Define $S$ as in lemma 2.

Note that the function:

$$T(x) = \binom{K}{x} \binom{x}{\frac{x}{M}, \ldots, \frac{x}{M}} \binom{K-x}{\frac{K-x}{M}, \ldots, \frac{K-x}{M}}$$

is Gaussian-shaped and therefore unimodal and has a unique maximum. $\eta^x$ for $\eta \in (0,1]$ is monotonically decreasing, and therefore their product is also unimodal.

Since we are only interested in solutions for $n \in \mathbb{N}$, we can reduce our problem to finding the maximal $n$ such that:

$$S(n-1) \leq S(n) \iff \frac{S(n)}{S(n-1)} \geq 1$$

and from lemma 2 we have:

$$\frac{S(n)}{S(n-1)} = \frac{\eta \left( K - \left\lfloor \frac{n-1}{M} \right\rfloor M \right)}{\left( \left\lfloor \frac{n-1}{M} \right\rfloor + 1 \right) M} \geq 1$$

$$\iff \left\lfloor \frac{n-1}{M} \right\rfloor M \leq \frac{\eta K - M}{1+\eta}$$

So we get that $S(n)$ is monotonically increasing as long as $\left\lfloor \frac{n-1}{M} \right\rfloor M \leq \frac{\eta K - M}{1+\eta}$, and the largest integer for which this condition holds is:

$$n = \left( \left\lfloor \frac{\eta K - M}{M(1+\eta)} \right\rfloor + 1 \right) M \simeq \frac{\eta K}{1+\eta}$$

$\square$

**Lemma 4.** *Denote by* $\mathcal{N}\left(\mathcal{B}_R^d\right)$ *the number of integer lattice points in* $\mathcal{B}_R^d$ *(the d-dimensional zero-centered ball of radius R). Then:*

$$\frac{\left(\frac{\pi e}{2}\right)^{\frac{d}{2}}}{\sqrt{\pi d}} \left( \frac{2R}{\sqrt{d}} - 1 \right)^d \leq \mathcal{N}\left(\mathcal{B}_R^d\right) \leq \frac{\left(\frac{\pi e}{2}\right)^{\frac{d}{2}}}{\sqrt{\pi d}} \left( \frac{2R}{\sqrt{d}} + 1 \right)^d$$

*Proof.* Let $\mathcal{I}\left(\mathcal{B}_R^d\right)$ be the set of all integer lattice points in $\mathcal{B}_R^d$. For $\boldsymbol{x} \in \mathcal{I}\left(\mathcal{B}_R^d\right)$, define:

$$C_{\boldsymbol{x}} := \times_{i=1}^d \left[ x_i - \frac{1}{2}, x_i + \frac{1}{2} \right] = \left\{ \boldsymbol{y} \in \mathbb{R}^d \,\middle|\, \forall i \in [d], x_i - \frac{1}{2} \leq y_i \leq x_i + \frac{1}{2} \right\}$$

Let $\boldsymbol{y} \in \bigcup_{\boldsymbol{x} \in \mathcal{I}\left(\mathcal{B}_R^d\right)} C_{\boldsymbol{x}}$, so there exists $\boldsymbol{x} \in \mathcal{I}\left(\mathcal{B}_R^d\right)$ such that $\boldsymbol{y} \in C_{\boldsymbol{x}}$, and from the triangle inequality we get:

$$\|\boldsymbol{y}\| \leq \|\boldsymbol{x}\| + \|\boldsymbol{y} - \boldsymbol{x}\| \leq R + \sqrt{\left(\frac{1}{2}\right)^2 + \ldots + \left(\frac{1}{2}\right)^2} = R + \frac{\sqrt{d}}{2}$$

and therefore $\boldsymbol{y} \in \mathcal{B}_{R+\frac{\sqrt{d}}{2}}^d$. Since $\boldsymbol{y}$ was chosen arbitrarily, we get that:

$$\bigcup_{\boldsymbol{x} \in \mathcal{I}\left(\mathcal{B}_R^d\right)} C_{\boldsymbol{x}} \subseteq \mathcal{B}_{R+\frac{\sqrt{d}}{2}}^d$$

Note that $Vol\left(C_{\boldsymbol{x}}\right) = 1$ for all $\boldsymbol{x} \in \mathcal{I}\left(\mathcal{B}_R^d\right)$ and that for $\boldsymbol{x}, \boldsymbol{x}' \in \mathcal{I}\left(\mathcal{B}_R^d\right)$ such that $\boldsymbol{x} \neq \boldsymbol{x}'$, $C_{\boldsymbol{x}} \cap C_{\boldsymbol{x}'}$ is a set of measure zero, hence:

$$\mathcal{N}\left(\mathcal{B}_R^d\right) = \left|\mathcal{I}\left(\mathcal{B}_R^d\right)\right|$$

$$= Vol\left(\bigcup_{\boldsymbol{x} \in \mathcal{I}\left(\mathcal{B}_R^d\right)} C_{\boldsymbol{x}}\right)$$

$$\leq Vol\left(\mathcal{B}_{R+\frac{\sqrt{d}}{2}}^d\right)$$

$$= \frac{\pi^{\frac{d}{2}}}{\Gamma\left(\frac{d}{2}+1\right)}\left(R + \frac{\sqrt{d}}{2}\right)^d$$

Assume for convenience that $d \mod 2 \equiv 0$, so $\Gamma\left(\frac{d}{2}+1\right) = \left(\frac{d}{2}\right)!$, and Stirling's approximation yields:

$$\simeq \frac{\pi^{\frac{d}{2}}}{\sqrt{\pi d}\cdot\left(\frac{d}{2e}\right)^{\frac{d}{2}}}\left(R + \frac{\sqrt{d}}{2}\right)^d = \frac{\left(\frac{\pi e}{2}\right)^{\frac{d}{2}}}{\sqrt{\pi d}}\left(\frac{2R}{\sqrt{d}}+1\right)^d$$

On the other hand, note that $\mathcal{B}_{R-\frac{\sqrt{d}}{2}}^d \subseteq \bigcup_{\boldsymbol{x}\in\mathcal{I}\left(\mathcal{B}_R^d\right)} C_{\boldsymbol{x}}$, and therefore:

$$\mathcal{N}\left(\mathcal{B}_R^d\right) = Vol\left(\bigcup_{\boldsymbol{x} \in \mathcal{I}\left(\mathcal{B}_R^d\right)} C_{\boldsymbol{x}}\right)$$

$$\geq Vol\left(\mathcal{B}_{R-\frac{\sqrt{d}}{2}}^d\right)$$

$$= \frac{\pi^{\frac{d}{2}}}{\Gamma\left(\frac{d}{2}+1\right)}\left(R - \frac{\sqrt{d}}{2}\right)^d$$

$$\simeq \frac{\left(\frac{\pi e}{2}\right)^{\frac{d}{2}}}{\sqrt{\pi d}}\left(\frac{2R}{\sqrt{d}}-1\right)^d$$

$\square$

**Lemma 5.** *Let $K, M$ be two fixed natural numbers, $s \in (0,1)$ a constant sensitivity parameter, and let"*

$$T_{K,M} := \left\{\boldsymbol{a} \in \mathbb{N}^M \left| \binom{K}{a_1,\ldots,a_M} \geq s\cdot\binom{K}{\frac{K}{M},\ldots,\frac{K}{M}} \right.\right\}$$

*Then:*

$$\left\{\boldsymbol{a} \in \mathbb{N}^M \left| \boldsymbol{a}^T\vec{\mathbf{1}} = K, \left\|\boldsymbol{a} - \frac{K}{M}\cdot\vec{\mathbf{1}}\right\|^2 \leq \frac{K}{M}\ln\left(s^{-1}\right) \right.\right\}$$

$$\subseteq T_{K,M}$$

$$\subseteq \left\{\boldsymbol{a} \in \mathbb{N}^M \left| \boldsymbol{a}^T\vec{\mathbf{1}} = K, \left\|\boldsymbol{a} - \frac{K}{M}\cdot\vec{\mathbf{1}}\right\|^2 \leq 4K\ln\left(s^{-1}\right) \right.\right\}$$

*Proof.* By Stirling's approximation, it holds that:

$$x! \simeq \sqrt{2\pi x}\cdot\left(\frac{x}{e}\right)^x$$

and a slightly more accurate version is:

$$x! \simeq \sqrt{2\pi\left(x+\frac{1}{6}\right)}\cdot\left(\frac{x}{e}\right)^x$$

and by plugging this approximation to the definition of a multinomial coefficient we get after some rearranging:

$$\binom{K}{a_1, \ldots, a_M} \simeq (2\pi)^{\frac{1-M}{2}} \cdot \frac{\sqrt{K + \frac{1}{6}}}{\prod_{i=1}^{M} \sqrt{a_i + \frac{1}{6}}} \cdot \frac{M^K}{\exp\left(\sum_{i=1}^{M} a_i \ln\left(\frac{M}{K} a_i\right)\right)}$$

Note that using this approximation and some rearrangements, we get that the following are equivalent:

$$\binom{K}{a_1, \ldots, a_M} \geq s \cdot \binom{K}{\frac{K}{M}, \ldots, \frac{K}{M}}$$

$$\iff (2\pi)^{\frac{1-M}{2}} \cdot \frac{\sqrt{K + \frac{1}{6}}}{\prod_{i=1}^{M} \sqrt{a_i + \frac{1}{6}}} \cdot \frac{M^K}{\exp\left(\sum_{i=1}^{M} a_i \ln\left(\frac{M}{K} a_i\right)\right)}$$

$$\geq s \cdot (2\pi)^{\frac{1-M}{2}} \cdot \frac{\sqrt{K + \frac{1}{6}}}{\prod_{i=1}^{M} \sqrt{\frac{K}{M} + \frac{1}{6}}} \cdot \frac{M^K}{\exp\left(\sum_{i=1}^{M} \frac{K}{M} \ln\left(\frac{M}{K} \cdot \frac{K}{M}\right)\right)} \tag{18}$$

$$\iff \left(\prod_{i=1}^{M} \sqrt{a_i + \frac{1}{6}}\right) \cdot \exp\left(\sum_{i=1}^{M} a_i \ln\left(\frac{M}{K} a_i\right)\right) \leq s^{-1} \cdot \left(\frac{K}{M} + \frac{1}{6}\right)^{\frac{M}{2}} \tag{19}$$

so we can characterize the set:

$$\tilde{T}_{K,M} := \left\{ \boldsymbol{a} \in \mathbb{N}^M \,\middle|\, \boldsymbol{a}^T \vec{\mathbf{1}} = K, \left(\prod_{i=1}^{M} \sqrt{a_i + \frac{1}{6}}\right) \cdot \exp\left(\sum_{i=1}^{M} a_i \ln\left(\frac{M}{K} a_i\right)\right) \leq s^{-1} \cdot \left(\frac{K}{M} + \frac{1}{6}\right)^{\frac{M}{2}} \right\}$$

as $\tilde{T}_{K,M} \simeq T_{K,M}$.

We'll start by finding a condition that will assure us that $(a_1, \ldots, a_M) \in T_{K,M}$ (i.e., we will characterize a subset of $T_{K,M}$).

First, note that by the AM-GM inequality,

$$\sqrt[M]{\prod_{i=1}^{M} \left(a_i + \frac{1}{6}\right)} \leq \frac{\sum_{i=1}^{M} \left(a_i + \frac{1}{6}\right)}{M} = \frac{K + \frac{1}{6} \cdot M}{M} = \frac{K}{M} + \frac{1}{6}$$

and therefore:

$$\prod_{i=1}^{M} \sqrt{a_i + \frac{1}{6}} = \left(\sqrt[M]{\prod_{i=1}^{M} \left(a_i + \frac{1}{6}\right)}\right)^{\frac{M}{2}} \leq \left(\frac{K}{M} + \frac{1}{6}\right)^{\frac{M}{2}}$$

Since we're interested in a subset of $T_{K,M}$, we can show that the last inequality holds when we replace the first term in the left-hand side ($\prod_{i=1}^{M} \sqrt{a_i + \frac{1}{6}}$) with $\left(\frac{K}{M} + \frac{1}{6}\right)^{\frac{M}{2}}$ (if the new inequality holds, (19) must hold as well), and we're left with:

$$\left(\frac{K}{M} + \frac{1}{6}\right)^{\frac{M}{2}} \cdot \exp\left(\sum_{i=1}^{M} a_i \ln\left(\frac{M}{K} a_i\right)\right) \leq s^{-1} \cdot \left(\frac{K}{M} + \frac{1}{6}\right)^{\frac{M}{2}}$$

$$\iff \sum_{i=1}^{M} a_i \ln\left(\frac{M}{K} a_i\right) \leq \ln\left(s^{-1}\right) \tag{20}$$

Now, for $i \in [M]$, let $h_i := \frac{M}{K} a_i - 1$, so:

$$\sum_{i=1}^{M} a_i \ln\left(\frac{M}{K} a_i\right) = \frac{K}{M} \sum_{i=1}^{M} (1 + h_i) \ln(1 + h_i)$$

and the last inequality becomes:

$$\frac{K}{M} \sum_{i=1}^{M} (1 + h_i) \ln(1 + h_i) \leq \ln\left(s^{-1}\right)$$

$$\iff \sum_{i=1}^{M} (1 + h_i) \ln(1 + h_i) \leq \frac{M}{K} \ln\left(s^{-1}\right)$$

Observe that for all $x \geq -1$, it holds that:

$$\ln(1 + x) \leq x$$

and therefore:

$$(1 + x)\ln(1 + x) \leq x^2 + x$$

so it suffices (again, we're only interested in a subset of $T_{K,M}$) to show that:

$$\sum_{i=1}^{M} \left(h_i^2 + h_i\right) \leq \frac{M}{K}\ln\left(s^{-1}\right) \tag{21}$$

Now, observe that:

$$\sum_{i=1}^{M} h_i = \sum_{i=1}^{M} \left(\frac{M}{K}a_i - 1\right)$$

$$= \frac{M}{K} \overbrace{\sum_{i=1}^{M} a_i}^{=K} - M$$

$$= 0$$

so (21) becomes:

$$\sum_{i=1}^{M} h_i^2 \leq \frac{M}{K}\ln\left(s^{-1}\right)$$

$$\iff \sum_{i=1}^{M} \left(a_i - \frac{K}{M}\right)^2 \leq \frac{K}{M}\ln\left(s^{-1}\right)$$

and we get that:

$$\left\{ \boldsymbol{a} \in \mathbb{N}^M \, \middle| \, \boldsymbol{a}^T \vec{\mathbf{1}} = K, \left\|\boldsymbol{a} - \frac{K}{M} \cdot \vec{\mathbf{1}}\right\|^2 \leq \frac{K}{M}\ln\left(s^{-1}\right) \right\} \subseteq T_{K,M}$$

Let us now turn to finding a condition that will assure us that $(a_1, \ldots, a_M) \notin T_{K,M}$ (i.e., we will characterize a subset of the complement of $T_{K,M}$). Note that:

$$\left(\prod_{i=1}^{M} \sqrt{a_i + \frac{1}{6}}\right) \cdot \exp\left(\sum_{i=1}^{M} a_i \ln\left(\frac{M}{K}a_i\right)\right)$$

$$= \left(\frac{K}{M} + \frac{1}{6}\right)^{\frac{M}{2}} \cdot \exp\left(\sum_{i=1}^{M} \left[a_i \ln\left(\frac{M}{K}a_i\right) + \frac{1}{2}\ln\left(\frac{6Ma_i + M}{6K + M}\right)\right]\right)$$

so if $(a_1, \ldots, a_M) \notin T_{K,M}$, we must have that:

$$\left(\frac{K}{M} + \frac{1}{6}\right)^{\frac{M}{2}} \cdot \exp\left(\sum_{i=1}^{M} \left[a_i \ln\left(\frac{M}{K}a_i\right) + \frac{1}{2}\ln\left(\frac{6Ma_i + M}{6K + M}\right)\right]\right)$$

$$> s^{-1} \cdot \left(\frac{K}{M} + \frac{1}{6}\right)^{\frac{M}{2}}$$

$$\iff \sum_{i=1}^{M} \left[a_i \ln\left(\frac{M}{K}a_i\right) + \frac{1}{2}\ln\left(\frac{6Ma_i + M}{6K + M}\right)\right] > \ln\left(s^{-1}\right)$$

and using the same definition of $h_i$ as before, we get:

$$\sum_{i=1}^{M} \left[ a_i \ln \left( \frac{M}{K} a_i \right) + \frac{1}{2} \ln \left( \frac{6M a_i + M}{6K + M} \right) \right]$$

$$= \sum_{i=1}^{M} \left[ \frac{K}{M} \left( 1 + h_i \right) \ln \left( 1 + h_i \right) + \frac{1}{2} \ln \left( \frac{6K \left( 1 + h_i \right) + M}{6K + M} \right) \right]$$

$$= \sum_{i=1}^{M} \left[ \frac{K}{M} \left( 1 + h_i \right) \ln \left( 1 + h_i \right) \right.$$

$$\left. + \frac{1}{2} \ln \left( \frac{6K \left( 1 + h_i \right) + M}{6K + M} \right) - \left( \frac{K}{M} + \frac{3K}{6K + M} \right) h_i \right]$$

where the last equality is due to the fact that $\sum_{i=1}^{M} h_i = 0$.

Observing the first order Taylor polynomial of the function:

$$f(x) = \frac{K}{M} \left( 1 + x \right) \ln \left( 1 + x \right) + \frac{1}{2} \ln \left( \frac{6K \left( 1 + x \right) + M}{6K + M} \right) - \left( \frac{K}{M} + \frac{3K}{6K + M} \right) x$$

at $x = 0$ with the remainder in the Lagrange form, we get:

$$f(h_i) = \left( \frac{K}{M} + \frac{3K}{6K + M} \right) h_i + f''(\xi_i) \cdot \frac{h_i^2}{2} - \left( \frac{K}{M} + \frac{3K}{6K + M} \right) h_i$$

$$= f''(\xi_i) \cdot \frac{h_i^2}{2}$$

$$= \left( \frac{K}{M \left( 1 + \xi_i \right)} - \frac{18K^2}{\left( 6K \left( 1 + \xi_i \right) + M \right)^2} \right) \cdot \frac{h_i^2}{2} \tag{22}$$

for some $\xi_i$ between $0$ and $h_i$.

Note that $f''(\xi_i)$ is monotonically decreasing with $\xi_i$ for $\xi_i \in (-1, M - 1]$, and using this fact and the fact that $K \geq 1$, (22) is lower bounded by:

$$\frac{K}{4M^2} \cdot h_i^2$$

So we can limit ourselves to looking at the cases where:

$$\sum_{i=1}^{M} \frac{K}{4M^2} \cdot h_i^2 > \ln \left( s^{-1} \right)$$

$$\iff \sum_{i=1}^{M} \left( a_i - \frac{K}{M} \right)^2 > 4K \ln \left( s^{-1} \right)$$

and we get that:

$$\left\{ \boldsymbol{a} \in \mathbb{N}^M \, \middle| \, \boldsymbol{a}^T \vec{\boldsymbol{1}} = K, \left\| \boldsymbol{a} - \frac{K}{M} \cdot \vec{\boldsymbol{1}} \right\|^2 > 4K \ln \left( s^{-1} \right) \right\}$$

$$\subseteq \left\{ \boldsymbol{a} \in \mathbb{N}^M \, \middle| \, \boldsymbol{a}^T \vec{\boldsymbol{1}} = K \right\} \backslash T_{K,M}$$

$$\iff T_{K,M} \subseteq \left\{ \boldsymbol{a} \in \mathbb{N}^M \, \middle| \, \boldsymbol{a}^T \vec{\boldsymbol{1}} = K, \left\| \boldsymbol{a} - \frac{K}{M} \cdot \vec{\boldsymbol{1}} \right\|^2 \leq 4K \ln \left( s^{-1} \right) \right\}$$

Combining the two results together we get:

$$\left\{ \boldsymbol{a} \in \mathbb{N}^M \, \middle| \, \boldsymbol{a}^T \vec{\boldsymbol{1}} = K, \left\| \boldsymbol{a} - \frac{K}{M} \cdot \vec{\boldsymbol{1}} \right\|^2 \leq \frac{K}{M} \ln \left( s^{-1} \right) \right\}$$

$$\subseteq T_{K,M}$$

$$\subseteq \left\{ \boldsymbol{a} \in \mathbb{N}^M \, \middle| \, \boldsymbol{a}^T \vec{\boldsymbol{1}} = K, \left\| \boldsymbol{a} - \frac{K}{M} \cdot \vec{\boldsymbol{1}} \right\|^2 \leq 4K \ln \left( s^{-1} \right) \right\}$$

$\square$

**Lemma 6.** *Let $K, M$ be two fixed natural numbers, and $s \in (0, 1)$ a constant sensitivity parameter. Then the number of multinomial coefficients, $\binom{K}{a_1, \dots, a_M}$, which uphold:*

$$\binom{K}{a_1, \dots, a_M} \geq s \cdot \binom{K}{\frac{K}{M}, \dots, \frac{K}{M}}$$

*is upper bounded by:*

$$\frac{\left(\frac{\pi e}{2}\right)^{\frac{M-1}{2}}}{(M-1)\sqrt{\pi}} \left(4\sqrt{\frac{K \ln(s^{-1})}{M-1}} + 1\right)^{M-1}$$

*and lower bounded by:*

$$\frac{\left(\frac{\pi e}{2}\right)^{\frac{M-1}{2}}}{M\sqrt{\pi}} \left(2\frac{\sqrt{K \ln(s^{-1})}}{M} - 1\right)^{M-1}$$

*Proof.* Let $\binom{K}{a_1, \dots, a_M}$ be a multinomial coefficient for which it holds that:

$$\binom{K}{a_1, \dots, a_M} \geq s \cdot \binom{K}{\frac{K}{M}, \dots, \frac{K}{M}}$$

and denote:

$$T_{K,M} := \left\{ \boldsymbol{a} \in \mathbb{N}^M \left| \binom{K}{a_1, \dots, a_M} \geq s \cdot \binom{K}{\frac{K}{M}, \dots, \frac{K}{M}} \right. \right\}$$

$$T_{K,M}^L := \left\{ \boldsymbol{a} \in \mathbb{N}^M \left| \boldsymbol{a}^T \vec{\mathbf{1}} = K, \left\| \boldsymbol{a} - \frac{K}{M} \cdot \vec{\mathbf{1}} \right\|^2 \leq \frac{K}{M} \ln(s^{-1}) \right. \right\}$$

$$T_{K,M}^U := \left\{ \boldsymbol{a} \in \mathbb{N}^M \left| \boldsymbol{a}^T \vec{\mathbf{1}} = K, \left\| \boldsymbol{a} - \frac{K}{M} \cdot \vec{\mathbf{1}} \right\|^2 \leq 4K \ln(s^{-1}) \right. \right\}$$

then by lemma 5,

$$T_{K,M}^L \subseteq T_{K,M} \subseteq T_{K,M}^U$$

and therefore:

$$\left| T_{K,M}^L \right| \leq |T_{K,M}| \leq \left| T_{K,M}^U \right|$$

so in order to bound the cardinality of $T_{K,M}$ (which is the quantity we are interested in), we can find an upper bound on the cardinality of $T_{K,M}^U$ and a lower bound on the cardinality of $T_{K,M}^L$.

Let $B \in \{L, U\}$ and $\boldsymbol{a} \in T_{K,M}^B$, and denote:

$$R(B) := \begin{cases} \sqrt{\frac{K}{M} \ln(s^{-1})} & \text{if } B = L \\ \sqrt{4K \ln(s^{-1})} & \text{if } B = U \end{cases}$$

For $i \in [M]$, denote $x_i := a_i - \frac{K}{M}$. So the problem has changed to finding the number of integer $M$-tuples $x_1, \dots, x_M$ such that $\sum_{i=1}^{M} x_i = 0$ and $\|\boldsymbol{x}\| \leq R(B)$, which is the number of integer lattice points $\boldsymbol{x}$ in the zero-centered $M$-dimensional ball of radius $R(B)$ that uphold $\sum_{i=1}^{M} x_i = 0$.

Note that the intersection of a $d$-dimensional ball of radius $R$ with the hyperplane $\mathcal{H} = \left\{ \boldsymbol{y} \in \mathbb{R}^d \left| \langle \boldsymbol{y}, \vec{\mathbf{1}} \rangle = 0 \right. \right\}$ is a $(d-1)$-dimensional ball, so we can assume that the number of integer lattice points in the zero-centered $d$-dimensional ball of radius $R$ whose coordinates add-up to zero is $\sim \frac{1}{\sqrt{d}} \cdot \mathcal{N}\left(\mathcal{B}_R^{d-1}\right)$, where $\frac{1}{\sqrt{d}}$ is the cosine of the angle between $\mathcal{H}$ and one of the axis-aligned $(d-1)$-dimensional hyperplanes in $\mathbb{R}^d$. In our case, $R = R(B)$ and $d = M$, so by lemma 4, the

cardinality of $T_{K,M}^U$ is upper bounded by:

$$\frac{1}{\sqrt{M}} \cdot \mathcal{N}\left(\mathcal{B}_{\sqrt{4K \ln(s^{-1})}}^{M-1}\right) \leq \frac{1}{\sqrt{M}} \cdot \frac{\left(\frac{\pi e}{2}\right)^{\frac{M-1}{2}}}{\sqrt{\pi(M-1)}} \left(\frac{2\sqrt{4K \ln(s^{-1})}}{\sqrt{M-1}} + 1\right)^{M-1}$$

$$\leq \frac{\left(\frac{\pi e}{2}\right)^{\frac{M-1}{2}}}{(M-1)\sqrt{\pi}} \left(4\sqrt{\frac{K \ln(s^{-1})}{M-1}} + 1\right)^{M-1}$$

and the cardinality of $T_{K,M}^L$ is lower bounded by:

$$\frac{1}{\sqrt{M}} \cdot \mathcal{N}\left(\mathcal{B}_{\sqrt{\frac{K}{M} \ln(s^{-1})}}^{M-1}\right) \geq \frac{1}{\sqrt{M}} \cdot \frac{\left(\frac{\pi e}{2}\right)^{\frac{M-1}{2}}}{\sqrt{\pi(M-1)}} \left(\frac{2\sqrt{\frac{K}{M} \ln(s^{-1})}}{\sqrt{M-1}} - 1\right)^{M-1}$$

$$\geq \frac{\left(\frac{\pi e}{2}\right)^{\frac{M-1}{2}}}{M\sqrt{\pi}} \left(2\frac{\sqrt{K \ln(s^{-1})}}{M} - 1\right)^{M-1}$$

$\square$

**Lemma 7.** *Let $K$ be a fixed natural number, $\eta \in (0,1]$, and $s \in (0,1)$ a constant sensitivity parameter.Then number of integer $n$s such that $\binom{K}{n}\eta^n \geq s \cdot \binom{K}{\frac{\eta K}{1+\eta}}\eta^{\frac{\eta K}{1+\eta}}$ is upper bounded by:*

$$\frac{K\sqrt{(1+2\ln(s))^2(1+\eta)^2 - 4\eta}}{2(1+\eta)(\ln(s^{-1}) - 1)}$$

*Proof.* Denote $n := \frac{\eta K}{1+\eta} + x$, so:

$$\binom{K}{n}\eta^n \geq s \cdot \binom{K}{\frac{\eta K}{1+\eta}}\eta^{\frac{\eta K}{1+\eta}}$$

$$\iff \binom{K}{\frac{\eta K}{1+\eta} + x}\eta^x \geq s \cdot \binom{K}{\frac{\eta K}{1+\eta}}$$

Recall that by Stirling's approximation we know that:

$$\binom{K}{n} \simeq \sqrt{\frac{K}{2\pi n(K-n)}} \cdot \frac{K^K}{n^n (K-n)^{K-n}}$$

So the number $x$s which uphold:

$$\sqrt{\frac{K}{2\pi\left(\frac{\eta K}{1+\eta} + x\right)\left(\frac{K}{1+\eta} - x\right)}} \cdot \frac{K^K}{\left(\frac{\eta K}{1+\eta} + x\right)^{\frac{\eta K}{1+\eta}+x}\left(\frac{K}{1+\eta} - x\right)^{\frac{K}{1+\eta}-x}}\eta^x$$

$$\geq s \cdot \sqrt{\frac{K}{2\pi\left(\frac{\eta K}{1+\eta}\right)\left(\frac{K}{1+\eta}\right)}} \cdot \frac{K^K}{\left(\frac{\eta K}{1+\eta}\right)^{\frac{\eta K}{1+\eta}}\left(\frac{K}{1+\eta}\right)^{\frac{K}{1+\eta}}}$$

$$\iff \begin{array}{l} \ln\left(s^{-1} \cdot \left(\frac{\eta K}{1+\eta}\right)^{\frac{\eta K}{1+\eta}+\frac{1}{2}}\left(\frac{K}{1+\eta}\right)^{\frac{K}{1+\eta}+\frac{1}{2}}\right) \\ \geq \ln\left(\left(\frac{\eta K}{1+\eta} + x\right)^{\frac{\eta K}{1+\eta}+x+\frac{1}{2}}\left(\frac{K}{1+\eta} - x\right)^{\frac{K}{1+\eta}-x+\frac{1}{2}} \cdot \eta^{-x}\right) \end{array} \tag{23}$$

approximates the number we are trying to quantify.

After some rearranging, one can observe that (23) is equivalent to:

$$\ln\left(s^{-1}\right) \geq \left(\frac{\eta K}{1+\eta} + x + \frac{1}{2}\right) \ln\left(1 + \frac{(1+\eta)\,x}{\eta K}\right)$$
$$+ \left(\frac{K}{1+\eta} - x + \frac{1}{2}\right) \ln\left(1 - \frac{(1+\eta)\,x}{K}\right) \tag{24}$$

and since for all $t > -1$, $\frac{t}{1+t} \leq \ln(1+t)$, the number of $x$s which uphold (24) is upper bounded by the number of integer $x$s for which it holds that:

$$\ln\left(s^{-1}\right) \geq \frac{\left(1 - \eta^2\right)Kx - 2\left(1+\eta\right)^2 x^2}{2\eta K^2 + 2\left(1-\eta^2\right)Kx - 2\left(1+\eta\right)^2 x^2}$$
$$\iff 2\left(1+\eta\right)^2 \left(\ln\left(s^{-1}\right) - 1\right) x^2$$
$$+ \left(1 - 2\ln\left(s^{-1}\right)\right)\left(1-\eta^2\right)Kx - 2\ln\left(s^{-1}\right)\eta K^2 \leq 0 \tag{25}$$

Recall that for inequalities of the form $ax^2 + bx + c \leq 0$ where $a > 0$, the set of all values of $x$ which satisfy this inequality is $\left(\frac{-b-\sqrt{b^2-4ac}}{2a}, \frac{-b+\sqrt{b^2-4ac}}{2a}\right)$ and the number of integer values of $x$ which satisfy this condition is approximately $\frac{\sqrt{b^2-4ac}}{a}$ (the interval's length).

In our case, (25) gives us:

$$a = 2\left(1+\eta\right)^2 \left(\ln\left(s^{-1}\right) - 1\right)$$
$$b = \left(1 - 2\ln\left(s^{-1}\right)\right)\left(1-\eta^2\right)K$$
$$c = -2\ln\left(s^{-1}\right)\eta K^2$$

and therefore:

$$\frac{\sqrt{b^2 - 4ac}}{a} = \frac{K\sqrt{\left(2\ln\left(s^{-1}\right) - 1\right)^2 \left(1+\eta\right)^2 - 4\eta}}{2\left(1+\eta\right)\left(\ln\left(s^{-1}\right) - 1\right)}$$

$\square$

**Lemma 8.** *Let $K, M$ be two fixed natural numbers, $\eta \in (0, 1]$, and $s \in \left(0, e^{-1.5}\right]$ a constant sensitivity parameter. Denote:*

$$D_{K,M} := \left\{(n, \boldsymbol{a}, \boldsymbol{b}) \in \mathbb{N}^{2M+1} \,\middle|\, \begin{matrix} 0 \leq n \leq K \\ a_1 + \ldots + a_M = n \\ b_1 + \ldots + b_M = K - n \end{matrix}\right\}$$

*define $F : D_{K,M} \longrightarrow \mathbb{R}$ as:*

$$F\left(n, \boldsymbol{a}, \boldsymbol{b}\right) = \binom{K}{n}\eta^n \binom{n}{a_1, \ldots, a_M}\binom{K-n}{b_1, \ldots, b_M}$$

*and let $\mathbf{x}^\star := \underset{\mathbf{x} \in D_{K,M}}{\arg\max} F\left(\mathbf{x}\right)$. If $\frac{\eta K}{1+\eta} \geq (M-1)$, then the number of $\mathbf{x} \in D_{K,M}$ which uphold $F\left(\mathbf{x}\right) \geq s \cdot F\left(\mathbf{x}^\star\right)$ is bounded from above by:*

$$\frac{2K}{(M-1)^2 \pi} \cdot \left(\frac{25\pi e K \ln\left(s^{-1}\right)\sqrt{\eta}}{2\left(M-1\right)\left(1+\eta\right)}\right)^{M-1}$$

*Proof.* By lemma 3, $\mathbf{x}^\star \simeq \left(\frac{\eta K}{M(1+\eta)}, \frac{\eta K}{M(1+\eta)}, \ldots, \frac{\eta K}{M(1+\eta)}, \frac{K}{M(1+\eta)}, \ldots, \frac{K}{M(1+\eta)}\right)$. By lemma 7, the number of $n$s between $0 \leq n \leq K$ such that $\binom{K}{n}\eta^n \geq s \cdot \left(\frac{K}{\frac{\eta K}{1+\eta}}\right)\eta^{\frac{\eta K}{1+\eta}}$ is upper bounded by $\frac{K\sqrt{(2\ln(s^{-1})-1)^2(1+\eta)^2-4\eta}}{2(1+\eta)(\ln(s^{-1})-1)}$, by lemma 6 the number of non-negative integer $M$-tuples $a_1, \ldots, a_M$ such that $a_1 + \ldots + a_M = \frac{\eta K}{1+\eta}$ and $\binom{\frac{\eta K}{1+\eta}}{a_1, \ldots, a_M} \geq s \cdot \left(\frac{\frac{\eta K}{1+\eta}}{\frac{\eta K}{M(1+\eta)}, \ldots, \frac{\eta K}{M(1+\eta)}}\right)$ is bounded from above by

$\frac{\left(\frac{\pi e}{2}\right)^{\frac{M-1}{2}}}{(M-1)\sqrt{\pi}}\left(4\sqrt{\frac{\eta K \ln(s^{-1})}{(M-1)(1+\eta)}}+1\right)^{M-1}$, and the number of non-negative integer $M$-tuples $b_1, \ldots, b_M$ such that $b_1 + \ldots + b_M = \frac{K}{1+\eta}$ and $\binom{\frac{K}{1+\eta}}{b_1, \ldots, b_M} \geq s \cdot \binom{\frac{K}{1+\eta}}{\frac{K}{M(1+\eta)}, \ldots, \frac{K}{M(1+\eta)}}$ is bounded from above by

$\frac{\left(\frac{\pi e}{2}\right)^{\frac{M-1}{2}}}{(M-1)\sqrt{\pi}}\left(4\sqrt{\frac{K \ln(s^{-1})}{(M-1)(1+\eta)}}+1\right)^{M-1}$. In total, without taking into consideration the interactions between the three multiplicands (so our bound is not tight), the number of $\mathbf{x} \in D_{K,M}$ which uphold $F(\mathbf{x}) \geq s \cdot F(\mathbf{x}^\star)$ is upper bounded by:

$$\underbrace{\frac{K\sqrt{\left(2\ln\left(s^{-1}\right)-1\right)^2\left(1+\eta\right)^2-4\eta}}{2\left(1+\eta\right)\left(\ln\left(s^{-1}\right)-1\right)}}_{\text{binomial} + \eta\text{-exponent}}$$

$$\cdot \underbrace{\frac{\left(\frac{\pi e}{2}\right)^{\frac{M-1}{2}}}{(M-1)\sqrt{\pi}}\left(4\sqrt{\frac{\eta K \ln\left(s^{-1}\right)}{(M-1)(1+\eta)}}+1\right)^{M-1}}_{a-\text{multinomial}}$$

$$\cdot \underbrace{\frac{\left(\frac{\pi e}{2}\right)^{\frac{M-1}{2}}}{(M-1)\sqrt{\pi}}\left(4\sqrt{\frac{K \ln\left(s^{-1}\right)}{(M-1)(1+\eta)}}+1\right)^{M-1}}_{b-\text{multinomial}}$$

and since $\frac{\eta K}{(M-1)(1+\eta)} \geq 1$ and $s \leq e^{-1.5}$ (and hence $\frac{2\ln(s^{-1})-1}{2(\ln(s^{-1})-1)} \leq 2$), this can be further bounded by:

$$\frac{2K}{(M-1)^2 \pi} \cdot \left(\frac{25\pi e K \ln\left(s^{-1}\right)\sqrt{\eta}}{2(M-1)(1+\eta)}\right)^{M-1}$$

$\square$

**Lemma 9.** *Let $K, M$ be two fixed natural numbers, $\eta \in (0, 1]$, and $s \in \left(0, e^{-1.5}\right]$ a constant sensitivity parameter. Denote:*

$$D_{K,M} := \left\{(n, \boldsymbol{a}, \boldsymbol{b}) \in \mathbb{N}^{2M+1} \, \middle| \, \begin{matrix} 0 \leq n \leq K \\ a_1 + \ldots + a_M = n \\ b_1 + \ldots + b_M = K - n \end{matrix} \right\}$$

*define $F : D_{K,M} \longrightarrow \mathbb{R}$ as:*

$$F(n, \boldsymbol{a}, \boldsymbol{b}) = \binom{K}{n}\eta^n\binom{n}{a_1, \ldots, a_M}\binom{K-n}{b_1, \ldots, b_M}$$

*and let $\mathbf{x}^\star := \arg\max_{\mathbf{x} \in D_{K,M}} F(\mathbf{x})$. If $\frac{K}{1+\eta} \geq M^2$, then the number of $\mathbf{x} \in D_{K,M}$ which uphold $F(\mathbf{x}) \geq s \cdot F(\mathbf{x}^\star)$ is bounded from below by:*

$$\frac{1}{M\sqrt{\pi}}\left(\frac{\pi e K \ln\left(s^{-1}\right)}{2M^2(1+\eta)}\right)^{\frac{M-1}{2}}$$

*Proof.* By lemma 6, the number of non-negative integer $M$-tuples $b_1, \ldots, b_M$ such that $b_1 + \ldots + b_M = \frac{K}{1+\eta}$ and $\binom{\frac{K}{1+\eta}}{b_1, \ldots, b_M} \geq s \cdot \binom{\frac{K}{1+\eta}}{\frac{K}{M(1+\eta)}, \ldots, \frac{K}{M(1+\eta)}}$ is bounded from below by

$\frac{\left(\frac{\pi e}{2}\right)^{\frac{M-1}{2}}}{M\sqrt{\pi}}\left(2^{\frac{\sqrt{\frac{K}{1+\eta}\ln(s^{-1})}}{M}}-1\right)^{M-1}$. Since these $b$s are only a subset of the elements of $D_{K,M}$ which $F$ takes into consideration, this is also a (quite loose) lower bound on the number of $\mathbf{x} \in D_{K,M}$ which uphold $F(\mathbf{x}) \geq s \cdot F(\mathbf{x}^\star)$.

Now, we know that:

$$\frac{K}{1+\eta} \geq M^2$$

and since $s \leq e^{-1.5}$, it holds that $\ln\left(s^{-1}\right) > 1$, we get:

$$\frac{K}{1+\eta}\ln\left(s^{-1}\right) > \frac{K}{1+\eta} \geq M^2$$

$$\iff -1 > -\frac{\sqrt{\frac{K}{1+\eta}\ln\left(s^{-1}\right)}}{M}$$

and therefore:

$$\frac{\left(\frac{\pi e}{2}\right)^{\frac{M-1}{2}}}{M\sqrt{\pi}}\left(2\frac{\sqrt{\frac{K}{1+\eta}\ln\left(s^{-1}\right)}}{M} - 1\right)^{M-1}$$

$$> \frac{\left(\frac{\pi e}{2}\right)^{\frac{M-1}{2}}}{M\sqrt{\pi}}\left(2\frac{\sqrt{\frac{K}{1+\eta}\ln\left(s^{-1}\right)}}{M} - \frac{\sqrt{\frac{K}{1+\eta}\ln\left(s^{-1}\right)}}{M}\right)^{M-1}$$

$$= \frac{1}{M\sqrt{\pi}}\left(\frac{\pi e K \ln\left(s^{-1}\right)}{2M^2\left(1+\eta\right)}\right)^{\frac{M-1}{2}}$$

$\square$

## C  LOWER BOUNDS ON THE $\varepsilon$-SEPARATION RANK

### C.1  PRELIMINARIES

#### C.1.1  TENSORS AND THEIR MATRICIZATION

We begin by laying out basic concepts in tensor theory required for the upcoming analysis. The core concept of a *tensor* may be thought of as a multi-dimensional array. The *order* of a tensor is defined to be the number of indexing entries in the array, referred to as *modes*. The *dimension* of a tensor in a particular mode is defined as the number of values taken by the index in that mode. If $\mathcal{A}$ is a tensor of order $N$ and dimension $M_i$ in each mode $i \in [N]$, its entries are denoted $\mathcal{A}_{d_1\ldots d_N}$, where the index in each mode takes values $d_i \in [M_i]$.

We will make use of the concept of the *matricization of $\mathcal{A}$ w.r.t. the balanced partition* $(P,Q)$, denoted $[\![\mathcal{A}]\!]_{P,Q} \in \mathbb{R}^{M^{N/2} \times M^{N/2}}$, which is essentially the arrangement of the tensor elements as a matrix whose rows correspond to $P$ and columns to $Q$. Suppose $\mathcal{A} \in \mathbb{R}^{M\times\cdots\times M}$ is a tensor of order $N$, and let $(P,Q)$ be a balanced partition of $[N]$, *i.e.* $P$ and $Q$ are disjoint size $N/2$ subsets of $[N]$ whose union gives $[N]$. The *matricization of $\mathcal{A}$ w.r.t. the partition* $(P,Q)$, denoted $[\![\mathcal{A}]\!]_{P,Q}$, is the $M^{N/2}$-by-$M^{N/2}$ matrix holding the entries of $\mathcal{A}$ such that $\mathcal{A}_{d_1\ldots d_N}$ is placed in row index $1 + \sum_{t=1}^{N/2}(d_{p_t} - 1)M^{N/2-t}$ and column index $1 + \sum_{t=1}^{N/2}(d_{q_t} - 1)M^{N/2-t}$.

We now present the concept of grid tensors, which are a form of function discretization (Hackbusch, 2012). Essentially, the function is evaluated for a set of points on an exponentially large grid in the input space and the outcomes are stored in a tensor. Formally, fixing a set of *template* vectors $\mathbf{x}^{(1)},\ldots,\mathbf{x}^{(Z)} \in [V]$, the points on the grid are the set $\{(\mathbf{x}^{(d_1)},\ldots,\mathbf{x}^{(d_N)})\}_{d_1,\ldots,d_N=1}^{Z}$. Given a function $y(\mathbf{x}^1,\ldots,\mathbf{x}^N)$, the set of its values on the grid arranged in the form of a tensor are called the grid tensor induced by $y$, denoted $\mathcal{A}(y)_{d_1,\ldots,d_N} \equiv y(\mathbf{x}^1 = \mathbf{x}^{(d_1)},\ldots,\mathbf{x}^N = \mathbf{x}^{(d_N)})$.

#### C.1.2  $\varepsilon$-RANK

We will make use of the concept of $\varepsilon$-*rank* Alon et al. (2013) of a matrix $A$ defined for any $\varepsilon > 0$ as the minimum rank over matrices that approximate every entry of $A$ to within an additive $\varepsilon$. We will prove lower bounds on the $\varepsilon$s for which the $\varepsilon$-rank a matrix remain high by the following lemma:

**Lemma 10.** *Let $M \in \mathbb{R}^{n\times n}$ be symmetric matrix and $\varepsilon > 0$, then:*

$$\forall k \leq n \quad \lambda_k\left(M\right) \geq \varepsilon \implies \frac{\varepsilon}{2n}\text{-rank}\left(M\right) \geq k \tag{26}$$

*Proof.* Let $E \in \left[\frac{-\varepsilon}{2n}, \frac{\varepsilon}{2n}\right]^{n \times n}$, we need to prove that $\mathrm{rank}\,(M + E) \geq k$. Since $M$ is symmetric, $M$ is diagonalizable with eigenvalues $\lambda_1 \geq \lambda_2 \geq \cdots \geq \lambda_n$. Denote by $v_1, v_2, \ldots, v_n$ the eigenvectors that are normalized according to the $l_1$ norm, then for any $i \leq k$ we have that:

$$\|(M + E)\, v_i\|_1 \geq \|M v_i\|_1 - \|E v_i\|_1 = \lambda_i - \|E v_i\|_1 \geq \lambda_i - n\frac{\varepsilon}{2n} = \frac{\varepsilon}{2} > 0 \qquad (27)$$

In particular for any $i \leq k$ we have that $(M + E)\, v_i \neq 0$ and since $v_1, v_2, \ldots, v_k$ are linearly independent we conclude that $\mathrm{rank}\,(M + E) \geq k$. $\qquad\square$

Finally, we will use the following lemma for lower bounding the amount of small eigenvalues of symmetric matrices:

**Lemma 11.** *Let $M \in \mathbb{R}^{n \times n}$ be a symmetric matrix with diagonal entries that equal to 1, and denote its eigenvalues by $\lambda_1 \geq \lambda_2, \ldots, \lambda_n$. Then:*

$$r := \max\left\{ k : \lambda_k \geq \frac{1}{n} \right\} \geq \frac{n - 1}{\|M\|_F} \qquad (28)$$

*Proof.* Since the trace of a matrix equals to both the sum of its eigenvalues and its diagonal entries, we get:

$$n = \sum_{i=1}^{n} [M]_{ii} = \mathrm{trace}\,(M) = \sum_{i=1}^{n} \lambda_i \leq r\lambda_1 + \frac{n - r}{n} \leq r\lambda_1 + 1 \qquad (29)$$

Eq. 28 follows from the fact that:

$$\|M\|_F = \sqrt{\lambda_1^2 + \ldots \lambda_n^2} \geq \lambda_1 \qquad (30)$$

$\qquad\square$

### C.1.3  High-Dimensional Spheres

We will use the Lebesgue measure on the sphere for taking expectations on $\mathbb{S}^d$. For any measurable subset $A \subseteq \mathbb{S}^d$ this measure is defined as the $d + 1$ dimensional volume of the "wedge" in the ball $\mathcal{B}^{d+1}$:

$$\mu\,(A) := \frac{\lambda^{d+1}\,(\{tx \,|\, x \in A, t \in [0, 1]\})}{\lambda^{d+1}\,(\mathcal{B}^{d+1})} \qquad (31)$$

Where $\lambda^{d+1}$ denotes the Lebesgue measure on $\mathbb{R}^{d+1}$. We will also use an unnormalized version of this measure:

$$\mu_{\text{unnormalized}}\,(A) := \mu\,(A) \cdot \lambda^{d+1}\,(\mathcal{B}^{d+1}) \qquad (32)$$

Smith & Vamanamurthy (1989) showed that $\mu_{\text{unnormalized}}\,(\mathbb{S}^d)$ is monotonically decreasing for $d > 5$. The following lemma bounds the rate of this decrease:

**Lemma 12.** *For any $d > 5$ the following holds:*

$$\frac{\mu_{\text{unnormalized}}\,(\mathbb{S}^{d-1})}{\mu_{\text{unnormalized}}\,(\mathbb{S}^d)} \leq \frac{d + 1}{2\pi} \qquad (33)$$

*Proof.* Since $d > 5$ we know that $\mu_{\text{unnormalized}}\,(\mathbb{S}^d) > \mu_{\text{unnormalized}}\,(\mathbb{S}^{d+1})$ and therefore:

$$\frac{\mu_{\text{unnormalized}}\,(\mathbb{S}^{d-1})}{\mu_{\text{unnormalized}}\,(\mathbb{S}^d)} \leq \frac{\mu_{\text{unnormalized}}\,(\mathbb{S}^{d-1})}{\mu_{\text{unnormalized}}\,(\mathbb{S}^{d+1})} \qquad (34)$$

Finally, Gipple (2014) showed that $\frac{\mu_{\text{unnormalized}}(\mathbb{S}^{d-1})}{\mu_{\text{unnormalized}}(\mathbb{S}^{d+1})} = \frac{d+1}{2\pi}$, completing the proof. $\qquad\square$

Finally, we will use a well known fact regarding the variation of the sphere volume for different radii (see for example Smith & Vamanamurthy (1989)):

**Fact 1.** *For any $d \in \mathbb{N}$, $R > 0$ the following holds:*

$$\frac{\mu_{\text{unnormalized}}\,(R\mathbb{S}^d)}{\mu_{\text{unnormalized}}\,(\mathbb{S}^d)} = R^{d+1} \qquad (35)$$

## C.2 PROOF OF THE LOWER BOUND

In this subsection, we prove theorem 2 of the main text. We will follow the proofs of Levine et al. (2020); Wies et al. (2021), with important adjustments to the $\varepsilon$-sequential-separation rank definition.

We begin by showing that high $\varepsilon$-*rank* Alon et al. (2013) of the grid tensor matricization implies high $\varepsilon$-*sequential-separation rank* (see section 2.2 of the main text) of the function. Essentially, we apply claim 1 from Levine et al. (2020) to $\varepsilon$-approximations obtained from the $\varepsilon$-*separation-rank* definition. This relation, which holds for all functions, is formulated below for functions realized by the analyzed Transformer network:

**Lemma 13.** *For $y_{\text{in-context}}^{(p,i),L,d_x}$ as defined in theorem 1 of the main text. Let $\varepsilon$-seq-sep $\left( y_{\text{in-context}}^{(p,i),L,d_x} \right)$ denote its $\varepsilon$-sequential-separation rank. Then, for any integer $Z$, any $\varepsilon > 0$, any set of template vectors $\mathbf{x}^{(1)}, \ldots, \mathbf{x}^{(Z)} \in \mathbb{R}^{d_x}$ and any sub-matrix $M$ of $[\![\mathcal{A}(\mathcal{Z}_{y_{\text{in-context}}^{(p,i),L,d_x}})]\!]_{\mathbf{a},\mathbf{b}}$ it holds that:*

$$\varepsilon\text{-seq-}sep\left( y_{\text{in-context}}^{(p,i),L,d_x} \right) \geq \varepsilon\text{-rank}\left( M \right), \tag{36}$$

*where $\mathcal{A}(\mathcal{Z}_{y_{\text{in-context}}^{(p,i),L,d_x}})$ is the grid tensor of $\mathcal{Z}_{y_{\text{in-context}}^{(p,i),L,d_x}}$ with respect to the above template vectors.*

*Proof.* If $\varepsilon$-seq-sep $\left( y_{\text{in-context}}^{(p,i),L,d_x} \right) = \infty$ then the inequality is trivially satisfied. Otherwise, assume that $\varepsilon$-seq-sep $\left( y_{\text{in-context}}^{(p,i),L,d_x} \right) = K \in \mathbb{N}$, and let $\tilde{y}$ be an $\varepsilon$-approximation for $\mathcal{Z}_{y_{\text{in-context}}^{(p,i),L,d_x}}$ with seq-$sep\left( \tilde{y} \right) = K$. By claim 1 of Levine et al. (2020) we have that rank $([\![\mathcal{A}(\tilde{y})]\!]_{\mathbf{a},\mathbf{b}}) \leq K$. Denote by $\tilde{M}$ the sub-matrix of $[\![\mathcal{A}(\tilde{y})]\!]_{\mathbf{a},\mathbf{b}}$ that corresponds to the rows and columns in $M$. Now, since $\tilde{y}$ is an $\varepsilon$-approximation for $\mathcal{Z}_{y^{(p,i),L,d_x}}$ we have that $\left\| M - \tilde{M} \right\|_{\infty} \leq \varepsilon$. Finally, by definition $\varepsilon$-rank $\left( M \right) \leq \text{rank}\left( \tilde{M} \right) \leq \text{rank}([\![\mathcal{A}(\tilde{y})]\!]_{\mathbf{a},\mathbf{b}}) \leq K$. $\qquad\square$

Relying on lemma 13, we will bound the $\varepsilon$-sequential-separation rank from below via the $\varepsilon$-rank of sub-matrices of $[\![\mathcal{A}(\mathcal{Z}_{y_{\text{in-context}}^{(p,i),L,d_x}})]\!]_{\mathbf{a},\mathbf{b}}$. Denote $d := {(d_x - H)}/{2}, \lambda := 3^{L-2}$, lemmas 10, 11 assure us that for $n := \left( \binom{d}{\lambda} \right) = \Omega\left( 2^{L \cdot d_x} \right)$ it is enough to prove that there exists an assignment to the network's weights, as well as choice of templates vectors, for which there exists sub-matrix $M \in \mathbb{R}^{n \times n}$ of $[\![\mathcal{A}(\mathcal{Z}_{y_{\text{in-context}}^{(p,i),L,d_x}})]\!]_{\mathbf{a},\mathbf{b}}$ that is symmetric with diagonal entries that equals to 1 and with $\|M\|_F \leq \left( \sqrt{(d+1)} \right) n^{\frac{3}{4}}$, in order to show that $\varepsilon$-seq-$sep\left( y_{\text{in-context}}^{(p,i),L,d_x} \right) \geq \frac{n^{1/4}}{2\sqrt{d+1}}$ for $\varepsilon \leq \frac{1}{2n^2}$.

Now we will use a corollary that is direct results of the proof in Levine et al. (2020). This corollary shows that if $\mathcal{Z}_{y_{\text{in-context}}^{(p,i),L,d_x}}$ is able to produce vectors that do not change the analysis in Levine et al. (2020), then for any matrix $B \in \mathbb{R}^{n \times d}$ with rows that are $l_2$ normalized, there exists an assignment to the networks weights, as well as choice of templates vectors, for which there exists a sub-matrix of the grid tensor matricization that is equal to[3] $M = \left( BB^T \right)^{\odot \lambda}$. Importantly $M$ is symmetric. In addition, since the rows of $B$ are $l_2$ normalized, the diagonal entries of $M$ equals to 1. Therefore, $M$ upholds the assumptions of lemmas 10, 11, 13 and it is enough to find $B$ for which $\|M\|_F \leq \left( \sqrt{(d+1)} \right) n^{\frac{3}{4}}$.

**Corollary 2.** *Let $d, \lambda > 0$, assume that for any matrix $A \in \mathbb{R}^{\left( \binom{d}{3^{L-2}} \right) \times d}$ with rows that are $l_2$ normalized, there exists a choice of template vectors $\mathbf{x}^{(1)}, \ldots, \mathbf{x}^{(Z)}$, an assignment to the embedding layer and the first self-attention layer key and query weights, such that for any $j_1, j_2 \in \left[ \left( \binom{d}{3^{L-2}} \right) \right]$ the output of the first self-attention layer on $\left( j_1, j_2 + \left( \binom{d}{3^{L-2}} \right) \right)$ is:*

$$\mathbf{y}^{(1,j)} = \left( \sum_{h=1}^{H} W^{O,1,h} W^{V,1,h} \right) \mathbf{u}$$

---

[3] We ignored the constant that appear in eq 28 of Levine et al. (2020) since we can get rid of this constant by dividing the last layer output matrices. Importantly, this constant is larger than 1 and therefore the network weights boundedness assumption is not violated

*for*

$$\forall \alpha \in [d_x] \quad \mathbf{u}_\alpha = \begin{cases} A_{j_1, \phi(\alpha)} & (\alpha - 1) \bmod d_a < \frac{d_a - 1}{2} \wedge \phi(\alpha) \leq d \\ A_{j_2, \phi\left(\alpha - \frac{d_a - 1}{2}\right)} & \frac{d_a - 1}{2} \leq (\alpha - 1) \bmod d_a < d_a - 1 \wedge \phi\left(\alpha - \frac{d_a - 1}{2}\right) \leq d \\ 2N & (\alpha - 1) \bmod d_a = d_a - 1 \\ 0 & \text{Otherwise} \end{cases}$$

*where $\phi(j) \equiv \lfloor j-1/d_a \rfloor \cdot (d_a - 1) + (j - 1 \bmod d_a) + 1$.*

*Then for any matrix $B \in \mathbb{R}^{n \times d}$ with rows that are $l_2$ normalized, there exists an assignment to the networks weights, as well as choice of templates vectors for which there exists sub-matrix of the grid tensor matricization that equal to $M = \left(BB^T\right)^{\odot \lambda}$.*

Now we will shows that indeed $\mathcal{Z}_{y_{in\text{-}context}^{(p,i),L,d_x}}$ is able to produce vectors that do not change the analysis in Levine et al. (2020) and the assumptions of corollary 2 holds.

**Lemma 14.** *Let $A \in \mathbb{R}^{\left(3^{\frac{d}{L-2}}\right) \times d}$ with rows that are $l_2$ normalized, then there exists a choice of template vectors $\mathbf{x}^{(1)}, \dots, \mathbf{x}^{(Z)}$, an assignment to the embedding layer and the first self-attention layer key and query weights, such that for any $j_1, j_2 \in \left[\left(\binom{d}{3^{L-2}}\right)\right]$ the output of the first self-attention layer on $\left(j_1, j_2 + \left(\binom{d}{3^{L-2}}\right)\right)$ is:*

$$\mathbf{y}^{(1,j)} = \left(\sum_{h=1}^{H} W^{O,1,h} W^{V,1,h}\right) \mathbf{u}$$

*for*

$$\forall \alpha \in [d_x] \quad \mathbf{u}_\alpha = \begin{cases} A_{j_1, \phi(\alpha)} & (\alpha - 1) \bmod d_a < \frac{d_a - 1}{2} \wedge \phi(\alpha) \leq d \\ A_{j_2, \phi\left(\alpha - \frac{d_a - 1}{2}\right)} & \frac{d_a - 1}{2} \leq (\alpha - 1) \bmod d_a < d_a - 1 \wedge \phi\left(\alpha - \frac{d_a - 1}{2}\right) \leq d \\ 2N & (\alpha - 1) \bmod d_a = d_a - 1 \\ 0 & \text{Otherwise} \end{cases}$$

*where $\phi(j) \equiv \lfloor j-1/d_a \rfloor \cdot (d_a - 1) + (j - 1 \bmod d_a) + 1$.*

*Proof.* We will ignore $\mathcal{Z}_{y_{in\text{-}context}^{(p,i),L,d_x}}$'s element-wise multiplication with vocabulary embedding matrix by choosing $\forall i, j \; M_{i,j}^V = 1$ (by the terms of corollary 2 it suffices to find any assignment of the learned weights).

For any $i \in \left[2\left(\binom{d}{3^{L-2}}\right) + 1\right]$ our templates vectors will be:

$$x_j^{(i)} = \frac{1}{N} \begin{cases} A_{i, \phi(j)} & \begin{array}{c} i \leq \binom{d}{3^{L-2}} \\ \wedge (j-1) \bmod d_a < \frac{d_a-1}{2} \wedge \phi(\alpha) \leq d \end{array} \\ A_{i - \binom{d}{3^{L-2}} + 1, \phi\left(j - \frac{d_a-1}{2}\right)} & \begin{array}{c} \binom{d}{3^{L-2}} < i \leq 2\binom{d}{3^{L-2}} \\ \wedge \frac{d_a-1}{2} \leq (j-1) \bmod d_a < d_a - 1 \wedge \phi\left(\alpha - \frac{d_a-1}{2}\right) \leq d \end{array} \\ N & (j-1) \bmod d_a = d_a - 1 \\ 0 & \text{Otherwise} \end{cases}$$

We will implement summation of the inputs embedding in the first self-attention layer, we will follow Levine et al. (2020) and set the first layer self-attention key and query weights to:

$$W_{i,j}^{K,1,h} = W_{i,j}^{Q,1,h} = 1_{i=1 \wedge j = d_a}$$

This assignment implements summation of the inputs embedding in the first self-attention layer since:

$$\mathbf{y}^{(1,i)}(\mathbf{x}^{(d_1)}, \ldots, \mathbf{x}^{(d_2)})_\alpha = \sum_{j=1}^{N} \sum_{h=1}^{H} \left\langle W^{Q,1,h} \left( M_V w_1^j \odot \mathbf{x}^{(d_1)} \right), W^{K,1,h} \left( M_V w_1^j \odot \mathbf{x}^{(d_1)} \right) \right\rangle \tag{37}$$

$$W^{O,1,h} W^{V,1,h} \left( M_V w_1^j \odot \mathbf{x}^{(d_1)} \right) \tag{38}$$

$$+ \sum_{j=1}^{N} \sum_{h=1}^{H} \left\langle W^{Q,1,h} \left( M_V w_2^j \odot \mathbf{x}^{(d_2)} \right), W^{K,1,h} \left( M_V w_2^j \odot \mathbf{x}^{(d_2)} \right) \right\rangle \tag{39}$$

$$W^{O,1,h} W^{V,1,h} \left( M_V w_2^j \odot \mathbf{x}^{(d_2)} \right) \tag{40}$$

$$\overset{=1}{=} \sum_{j=1}^{N} \sum_{h=1}^{H} \overbrace{\mathbf{x}_{d_a}^{(d_1)}}^{=1} \cdot \overbrace{\mathbf{x}_{d_a}^{(d_1)}}^{=1} W^{O,1,h} W^{V,1,h} \mathbf{x}^{(d_1)} \tag{41}$$

$$+ \sum_{j=1}^{N} \sum_{h=1}^{H} \overbrace{\mathbf{x}_{d_a}^{(d_2)}}^{=1} \cdot \overbrace{\mathbf{x}_{d_a}^{(d_2)}}^{=1} W^{O,1,h} W^{V,1,h} \mathbf{x}^{(d_2)} \tag{42}$$

$$\overset{2}{=} \left( \sum_{h=1}^{H} W^{O,1,h} W^{V,1,h} \right) N \left( \mathbf{x}^{(d_1)} + \mathbf{x}^{(d_2)} \right) \tag{43}$$

where (1) is because $W^{Q,1,h} = W^{K,1,h}$ are matrices that are zero everywhere except for entry $(1, d_a)$ and that all the entries in the vocabulary embedding matrix equals to 1, and (2) because of linearity. Therefore, for any $j_1, j_2 \in \left[ \left( \binom{d}{3^{L-2}} \right) \right]$ the output of the first self-attention layer on $j_1, j_2$ is:

$$\mathbf{y}^{(1,j)} = \left( \sum_{h=1}^{H} W^{O,1,h} W^{V,1,h} \right) \underbrace{N \left( \mathbf{x}^{(d_1)} + \mathbf{x}^{(d_2)} \right)}_{=:\mathbf{u}} \tag{44}$$

Finally, we need to show that indeed for any $j_1, j_2 \in \left[ \left( \binom{d}{3^{L-2}} \right) \right]$ eq 44 give the desired $\mathbf{u}$:

$$\forall \alpha \in [d_x] \quad \mathbf{u}_\alpha = \begin{cases} A_{j_1, \phi(\alpha)} & (\alpha - 1) \bmod d_a < \frac{d_a - 1}{2} \wedge \phi(\alpha) \leq d \\ A_{j_2, \phi\left(\alpha - \frac{d_a - 1}{2}\right)} & \frac{d_a - 1}{2} \leq (\alpha - 1) \bmod d_a < d_a - 1 \wedge \phi\left(\alpha - \frac{d_a - 1}{2}\right) \leq d \\ 2N & (\alpha - 1) \bmod d_a = d_a - 1 \\ 0 & \text{Otherwise} \end{cases}$$

The third and forth cases are clear from $\mathbf{x}'s$ definition, so it remain to prove the first and second cases. For this we will examine $d_1, d_2$. $d_1 = j_1 \leq \left( \binom{d}{3^{L-2}} \right)$ and therefore:

$$\mathbf{x}_\alpha^{(d_1)} = \begin{cases} A_{j_1, \phi(\alpha)} & (\alpha - 1) \bmod d_a < \frac{d_a - 1}{2} \wedge \phi(\alpha) \leq d \\ 0 & \frac{d_a - 1}{2} \leq (\alpha - 1) \bmod d_a < d_a - 1 \wedge \phi\left(\alpha - \frac{d_a - 1}{2}\right) \leq d \end{cases}$$

$d_2 = j_2 + \left( \binom{d}{3^{L-2}} \right) \in \left[ 1 + \left( \binom{d}{3^{L-2}} \right), 2 \left( \binom{d}{3^{L-2}} \right) \right]$ and therefore:

$$\mathbf{x}_\alpha^{(d_2)} = \begin{cases} 0 & (\alpha - 1) \bmod d_a < \frac{d_a - 1}{2} \wedge \phi(\alpha) \leq d \\ A_{j_2, \phi\left(\alpha - \frac{d_a - 1}{2}\right)} & \frac{d_a - 1}{2} \leq (\alpha - 1) \bmod d_a < d_a - 1 \wedge \phi\left(\alpha - \frac{d_a - 1}{2}\right) \leq d \end{cases}$$

So it clear that also the first and second cases upholds. $\square$

Returning to finding $B$ for which $\|M\|_F \leq \left( \sqrt{(d+1)} \right) n^{\frac{3}{4}}$, we will use the probabilistic method for proving the existence of such $B$, *i.e.* we will show that for random $B$ the expectation of $\|M\|_F \leq \left( \sqrt{(d+1)} \right) n^{\frac{3}{4}}$ and therefore in particular there exists such $B$.

**Lemma 15.** *For any $d, \lambda \in \mathbb{N}$ such that $\lambda \geq d$ the following holds:*

$$\mathbb{E}_{B \sim (\mathbb{S}^d)^n} \left[ \left\| \left( BB^T \right)^{\odot \lambda} \right\|_F \right] \leq \sqrt{(d+1)} \left( \binom{d}{\lambda} \right)^{\frac{3}{4}} \tag{45}$$

*Proof.* We start by bounding the expectation of the squared norm:

$$\mathbb{E}_{B \sim (\mathbb{S}^d)^n} \left[ \left\| \left( BB^T \right)^{\odot \lambda} \right\|_F^2 \right] = \sum_{i,j=1}^n \mathbb{E}_{B \sim (\mathbb{S}^d)^n} \left[ \left[ \left( BB^T \right)^{\odot 2\lambda} \right]_{i,j} \right] \tag{46}$$

$$= \sum_{i,j=1}^n \mathbb{E}_{u,v \sim \mathbb{S}^d} \left[ \langle u, v \rangle^{2\lambda} \right] = n^2 \mathbb{E}_{u,v \sim \mathbb{S}^d} \left[ \langle u, v \rangle^{2\lambda} \right] \tag{47}$$

Now, $\mathbb{E}_{u,v \sim \mathbb{S}^d} \left[ \langle u, v \rangle^{2\lambda} \right] \leq (d+1) \left( \binom{d}{\lambda} \right)^{-\frac{1}{2}}$ by lemma 16 and therefore we got that:

$$\mathbb{E}_{B \sim (\mathbb{S}^d)^n} \left[ \left\| \left( BB^T \right)^{\odot \lambda} \right\|_F^2 \right] \leq (d+1) \left( \binom{d}{\lambda} \right)^{\frac{3}{2}} \tag{48}$$

Finally, by Jensen inequality we have that:

$$\mathbb{E}_{B \sim (\mathbb{S}^d)^n} \left[ \left\| \left( BB^T \right)^{\odot \lambda} \right\|_F \right] \leq \sqrt{\mathbb{E}_{B \sim (\mathbb{S}^d)^n} \left[ \left\| \left( BB^T \right)^{\odot \lambda} \right\|_F^2 \right]} \leq \sqrt{(d+1)} \left( \binom{d}{\lambda} \right)^{\frac{3}{4}} \tag{49}$$

$\square$

### C.3 TECHNICAL LEMMAS

**Lemma 16.** *For any $d, \lambda \in \mathbb{N}$ such that $\lambda \geq d$ the following holds:*

$$\mathbb{E}_{u,v \sim \mathbb{S}^d} \left[ \langle u, v \rangle^{2\lambda} \right] \leq (d+1) \left( \binom{d}{\lambda} \right)^{-\frac{1}{2}} \tag{50}$$

*Proof.* We will use conditional expectation to make reduction for simpler expectation. From rotational invariance of the uniform measure on $\mathbb{S}^d$ we know that for every rotation matrix $R \in SO(d)$ and unit vector $v \in \mathbb{S}^d$ we have that:

$$\mathbb{E}_{u \sim \mathbb{S}^d} \left[ \langle u, v \rangle^{2\lambda} | v \right] = \mathbb{E}_{u \sim \mathbb{S}^d} \left[ \langle Ru, Rv \rangle^{2\lambda} | v \right] = \mathbb{E}_{u \sim \mathbb{S}^d} \left[ \langle u, Rv \rangle^{2\lambda} | v \right] \tag{51}$$

where the first equality holds because $R$ is orthogonal. Therefore, by choosing $R$ such that $Rv = e_1$ we will get that:

$$\mathbb{E}_{u,v \sim \mathbb{S}^d} \left[ \langle u, v \rangle^{2\lambda} \right] = \mathbb{E}_{v \sim \mathbb{S}^d} \left[ \mathbb{E}_{u \sim \mathbb{S}^d} \left[ \langle u, v \rangle^{2\lambda} | v \right] \right] \tag{52}$$

$$= \mathbb{E}_{v \sim \mathbb{S}^d} \left[ \mathbb{E}_{u \sim \mathbb{S}^d} \left[ u_1^{2\lambda} | v \right] \right] = \mathbb{E}_{u \sim \mathbb{S}^d} \left[ u_1^{2\lambda} \right] \tag{53}$$

Now we can calculate the last expectation directly:

$$\mathbb{E}_{u \sim \mathbb{S}^d} \left[ u_1^{2\lambda} \right] = \int_{u \in \mathbb{S}^d} u_1^{2\lambda} d\mu(u) = \int_{-1}^1 \frac{\mu_{\text{unnormalized}} \left( \left( \sqrt{1-x^2} \right) \mathbb{S}^{d-1} \right)}{\mu_{\text{unnormalized}} \left( \mathbb{S}^d \right)} x^{2\lambda} dx \tag{54}$$

Now, by lemma 12 and fact 1 we have that:

$$0 < \frac{\mu_{\text{unnormalized}} \left( \left( \sqrt{1-x^2} \right) \mathbb{S}^{d-1} \right)}{\mu_{\text{unnormalized}} \left( \mathbb{S}^d \right)} \leq \frac{d+1}{2\pi} \left( 1 - x^2 \right)^{\frac{d}{2}} \tag{55}$$

Therefore we have that:

$$\mathbb{E}_{u,v \sim \mathbb{S}^d} \left[ \langle u, v \rangle^{2\lambda} \right] \leq \frac{d+1}{2} \int_{-1}^1 \left( 1 - x^2 \right)^{\frac{d}{2}} x^{2\lambda} dx = (d+1) \int_0^1 \left( 1 - x^2 \right)^{\frac{d}{2}} x^{2\lambda} dx \tag{56}$$

Finally, by lemma 17 each term in the integral is upper bounded by $\left( \binom{d}{\lambda} \right)^{-\frac{1}{2}}$ and thus:

$$\mathbb{E}_{u,v \sim \mathbb{S}^d} \left[ \langle u, v \rangle^{2\lambda} \right] \leq (d+1) \left( \binom{d}{\lambda} \right)^{-\frac{1}{2}} \tag{57}$$

$\square$

**Lemma 17.** *For any $x \in [0, 1]$ and $d, \lambda \in \mathbb{N}$ such that $\lambda \geq d$ the following holds:*

$$x^{2\lambda} \left(1 - x^2\right)^{\frac{d}{2}} \leq \left(\binom{d}{\lambda}\right)^{-\frac{1}{2}} \tag{58}$$

*Proof.* Note that since $x^{2\lambda} \left(1 - x^2\right)^{\frac{d}{2}} = 0$ in the boundaries ($x \in \{0, 1\}$), it is enough to prove the inequality for critical points.

$$0 = \left(x^{2\lambda} \left(1 - x^2\right)^{\frac{d}{2}}\right)' = \left(2\lambda x^{2\lambda-1}\right) \left(1 - x^2\right)^{\frac{d}{2}} - \frac{2d}{2} x^{2\lambda+1} \left(1 - x^2\right)^{\frac{d}{2}-1} \tag{59}$$

$$\iff 0 = 2\lambda \left(1 - x^2\right) - dx^2 \iff x^2 = \frac{2\lambda}{2\lambda + d} \tag{60}$$

Therefore, $x^2 = \frac{2\lambda}{2\lambda+d}$ is the only critical point and:

$$x^{2\lambda} \left(1 - x^2\right)^{\frac{d}{2}} \leq \left(\frac{2\lambda}{2\lambda + d}\right)^{\lambda} \left(1 - \frac{2\lambda}{2\lambda + d}\right)^{\frac{d}{2}} \tag{61}$$

$$= \left(1 - \frac{d}{2\lambda + d}\right)^{\lambda} \left(\frac{d}{2\lambda + d}\right)^{\frac{d}{2}} \tag{62}$$

$$\leq \left(1 - \frac{d}{3\lambda}\right)^{\lambda} \left(\frac{d}{1.5\left(\lambda + d\right)}\right)^{\frac{d}{2}} \tag{63}$$

$$\leq \left(\sqrt[3]{e}\right)^{-d} \left(\sqrt{\frac{e}{1.5}}\right)^{d} \left(\frac{d}{e\left(\lambda + d\right)}\right)^{\frac{d}{2}} \tag{64}$$

$$\leq \left(\frac{d}{e\left(\lambda + d\right)}\right)^{\frac{d}{2}} \leq \left(\binom{d}{\lambda}\right)^{-\frac{1}{2}} \tag{65}$$

where the last inequality follow from the fact that:

$$\left(\binom{\lambda}{d}\right) = \binom{\lambda + d - 1}{d} \leq \left(\frac{e\left(\lambda + d - 1\right)}{d}\right)^{d} \leq \left(\frac{e\left(\lambda + d\right)}{d}\right)^{d} \tag{66}$$

$\square$

# D    EXPERIMENTAL DETAILS

## D.1    KNN-TAPT

We conducted the network training described in section 3.1 of the main text with AdamW optimizer (with the parameters suggested in the original RoBERTa paper: $\beta_1 = 0.9$, $\beta_2 = 0.98$, $\varepsilon = 10^{-6}$ and weight decay of 0.01), with batch sizes of 128 or 256 (depending on model size) and sequences of 256 tokens each. We started with pretrained RoBERTA-base weights from the HuggingFace Transformers repository [4], and continued training them on the MLM task with masking probability of 15%, where each masked token had a probability of 80% of being replaced with the special $\boxed{\text{MASK}}$ token, 10% of being replaced with a random token and 10% of being kept the same. The data used for this phase of training was created using the four different procedures described in section 3.1. After the training was finished, we evaluated the models' performance using the SentEval kit.

## D.2    KNN-PRETRAINING

We conducted the network training described in section 3.2 of the main text with AdamW optimizer (with the parameters suggested in the original GPT-2 paper: $\beta_1 = 0.9$, $\beta_2 = 0.95$, $\varepsilon = 10^{-8}$ and weight decay of 0.1), with batch size of 512 and sequences of 256 tokens each. We pretrained a HuggingFace Transformers implementation of GPT-2 from scratch on Wikipedia with the standard LM objective, and switched to a mixture of the standard data and our generated kNN data in two different points during training. After the training was finished, we evaluated the models' performance on the Natural Questions benchmark.

---
[4] https://huggingface.co/transformers/

## E  kNN-PRETRAINING AT DIFFERENT CHECKPOINTS

The following table includes F1 evaluation scores of zero shoe closed book Natural Questions examples for different model sizes at different training checkpoints. Overall, further pretraining seems to improve the effectiveness of kNN-Pretraining.

| Model Size | Reg.+kNN Steps | NQ F1 |
|---|---|---|
| 110M | 200 / 400+0 | $< 10^{-3}$ |
| 110M | 200+40 | $6.2 \cdot 10^{-3}$ |
| 110M | 400+40 | $7.9 \cdot 10^{-3}$ |
| 345M | 200 / 400+0 | $< 10^{-3}$ |
| 345M | 200+10 | $9.6 \cdot 10^{-3}$ |
| 345M | 400+10 | $1.4 \cdot 10^{-2}$ |

Table 2: Impact of model size and regular pretraining steps on the Natural Questions F1 score of kNN-Pretraining.

## F  kNN-PRETRAINING ON ADDITIONAL BENCHMARKS

The main text describes experiments on the Natural Questions dataset. We test how kNN-Pretraining affects other NLU tasks, by examining several tasks from the GLUE benchmark (Wang et al., 2018) – Multi-Genre Natural Language Inference (MNLI) (Williams et al., 2017), Recognizing Textual Entailment (RTE) (Dagan et al. (2010) and others), and the The Winograd Schema Challenge (WNLI) (Levesque et al., 2012). As in the case of Natural Questions, we evaluate the zero-shot performance of our models since it is a direct probe to the abilities of the model straight after the process of pretraining. In contrast to Natural Questions, the GLUE tasks we examined are classification tasks and not generation tasks, so assessing zero shot performance on them is not straightforward. We therefore follow the template-based method of Gao et al. (2021a) for converting the tasks' data into a format processable by unidirectional language models.

Notably, the examined GLUE classification tasks are not easy for the examined unidirectional models in zero shot. Table 3 includes the zero-shot scores of the 345M parameter model that trained regularly for 200K steps and then continued training for 20K steps of kNN-Pretraining, versus the average of 3 baselines that trained regularly for the same number of overall steps (the same models used in figure 1). Similarly to the results on Natural Questions (figure 1), all examined models score only slightly better than random guess on the examined GLUE tasks. However (and again similarly to the case of Natural Questions), we get a clear signal that kNN-Pretraining significantly moves the needle when applied for just $10\%$ of the regular pretraining time. We conjecture that when using stronger models (that train for longer and over more data), the positive effect of kNN-Pretraining will be enhanced, since as the model improves, it can better understand and utilize the various in-context hints that kNN-Pretraining provides.

| GLUE Task | MNLI | RTE | WNLI |
|---|---|---|---|
| Random guess | 33.3 | 50.0 | 50.0 |
| 3 baselines – Average score | 35.1 | 52.0 | 51.1 |
| 3 baselines – Max score | 35.3 | 52.3 | 54.9 |
| kNN-Pretraining | **35.5** | **53.0** | **56.3** |

Table 3: Zero-shot accuracy scores on several GLUE tasks of a kNN-Pretrained model versus 3 baselines that trained regularly on the same data.

