# OpenReview forum: "The Inductive Bias of In-Context Learning: Rethinking Pretraining Example Design"
_ICLR.cc/2022/Conference — ICLR 2022 Spotlight_

### Official Review · Reviewer_cz9E · 2021-11-01

**Correctness:** 3
**Technical Novelty And Significance:** 3
**Empirical Novelty And Significance:** 3
**Recommendation:** 8
**Confidence:** 3

**Main Review:**

I thought that this paper was very thought-provoking, and I appreciated the attempts to better understand what is going on with pre-trained language models, why they work well, and what might we be able to improve from theses insights.

Strengths:
- Thorough theoretical analysis that reveals the connection between (practically-necessary) small learning rates and inability to use dependencies across text chunks
- Useful framing and discussion of the "in-context bias", where models are more likely to learn dependencies within text chunks seen during pre-training.
- reasonable initial experimental results demonstrating some ways to help models better use cross-text-chunk dependencies (put them into a contiguous text chunk), providing some hope that these results could make models better.

Weaknesses:
- I felt like the empirical validation could have been stronger. The only two tasks examined are sentence similarity tasks (which seem a bit more like a sanity check), and NaturalQuestions. It's particularly surprising to me that this works so well on NQ, and I wish the authors had dug a bit deeper into this, but I also recognize that page limits exist.
- Where else do you think the in-context bias could be useful? It wasn't intuitive for me that it'd be useful for NQ.
- Do you have any initial experiments on the "self-improving" aspect of this technique? This is mentioned several times, but there are no initial results or anything suggesting that it might be a promising direction to pursue.

**Summary Of The Paper:**

This paper formalizes the notion of an in-context bias---the pre-trained neural language models are better at modeling dependencies that appear within one contiguous pre-training text chunk, rather than those that appear in different pre-training text chunks. The authors perform a theoretical analysis of this phenomenon and link the inability to model dependencies between different examples with the models' low-pretraining learning rate. The authors then propose two methods for improving pre-training example design by adding sentences that are similar (KNN), but from different documents. They empirically show that this works better than reasonable baselines on sentence similarity tasks (e.g., STS and SICK), as well as NaturalQuestions.

**Summary Of The Review:**

I thought this paper was interesting and brings a new perspective to practical design decisions used in pre-training language models. I think this paper is relevant to the ICLR audience, and that said audience would be excited to know about it.

---

> ### Author Response · Authors · 2021-11-19
> **Response to reviewer**
>
> Thank you for your supportive feedback!
>
> 1.
>
> With recommendations from you and other reviewers, we quickly evaluated the zero shot performance of our kNN-pretrained model on several additional tasks from the GLUE benchmark (recommended by Reviewer yRBu), and added the results in appendix F.
> The outcome resembles that of Natural Questions -- the zero shot scores are generally low but kNN-Pretraining again significantly moves the needle relative to the baselines after just 10% of the regular pretraining time.
> Given that results are manifested on more NLU tasks and that the signal from NQ is reinforced, we further conjecture that when using stronger models (that train for longer and over more data), the positive effect of kNN-Pretraining will be enhanced -- since as the model improves, it can better understand and utilize the various in-context hints that kNN-Pretraining provides. In the camera ready version, we will follow up pn the experimental signals in the paper body and Appendix F by pretraining stronger models, going beyond the controlled and limited setting of 3 epochs on Wikipedia, in order to show more practical usefulness. We will integrate existing GLUE scores of Appendix F and any new results more naturally in the experiments section of the main body, the message being that kNN pretraining can be useful on a larger variety of tasks.
>
>
> 2. The in-context bias can be useful in many other cases, some examples being --
>
> * Learning about entities, events, or concepts by concatenating all/many of their mentions within the same training example.
> * Including dictionary definitions for rare concepts within the same training example in which they are used.
> * Including documents from wikipedia together in the same training example if their “wikipedia link distance” is small.
> * Generally, any time that a cross corpus signal can be accurately identified as pertinent for a specific task, the in-context bias motivates finding methods to automatically identify this signal and incorporate it in the pretraining example construction protocol. kNN in embedding space is simply our suggestion for doing it automatically, and the benefits we get in NQ, STS, and the newly added GLUE tasks emerge from relevant information to the task being better processed (the information type is a direct result of the quality and traits of the employed embedding space). But this is just an example, we leave it for you and the readers to “crack” further useful signals and applications for the in-context bias :)
>
> 3.
>
> We have indeed emphasized the potential for “self improving” representations that emerges from the kNN pretraining methods, since we truly believe in this avenue of “squeezing the lemon” from the corpus via self-improving example design. However, there is a technical complication for conducting such experiments in our framework: for constructing KNN-Pretraining examples we used the representations of a RoBERTa large model fine tuned by the “SentenceBERT'' scheme of Reimers and Gurevych (2019) to produce good sentence representations. For the evolving example design idea to make sense, the kNN pretrained model must surpass the original Sentence Roberta’s sentence representation in quality so that it can provide a refined kNN search. Since we have kNN pretrained a unidirectional model, which represents sentences worse than bidirectional models, it does not quite reach Sentence RoBERTa’s representation performance yet. So bottom line, despite being conceptually an immediate next step from our current kNN pretraining schemes, it is complicated enough to merit its own separate work. We have accordingly “toned down” the reference to this avenue and mention it in the discussion as a possible route for future work.
>
> Thanks again for your support and thoughtful comments which allowed us to improve our paper!

---

### Official Review · Reviewer_twvb · 2021-11-02

**Correctness:** 4
**Technical Novelty And Significance:** 4
**Empirical Novelty And Significance:** 4
**Recommendation:** 8
**Confidence:** 2

**Main Review:**

Strengths:
 * Clearly written paper
 * The theoretical contributions will have a high impact on training transformer-based models
 * The theoretical analysis is supplemented by experimental analysis

 Weaknesses:
  * No major weaknesses


Some minor aspects:
 * "correct on less than 50 questions out of 20, 000 in the evaluation set" – Separating thousands by commas (American system) is ok. Separating them by space is also ok. Doing both is not.
 * "correct on roughly than 250 questions in the evaluation set" – "than" shouldn't be there
 * Equation 4 would have been easier to read if the right-hand side was expanded to S1 and S2's `{w_2^j}` notation from the previous line.

**Summary Of The Paper:**

The paper presents a theoretical analysis of the strengths of including similar input examples S1 and S2 into the same actual transformer input. The analysis contrasts using examples of concatenated S1;S2 with running the two in separate inputs. The paper then proves empirically that including related examples in the same input (via a method the article calls KNN-pretraining) allows transformers to learn much faster in cases when cross-document dependency is relevant.

**Summary Of The Review:**

Despite its theoretical focus, the paper is easy to read and to follow. The theoretical analysis is supplemented by empirical results.

---

> ### Author Response · Authors · 2021-11-19
> **Response to reviewer**
>
> Thank you for your supportive feedback.
>
> We are happy that you found the paper clear and easy to follow. It is more challenging in a theoretical paper..
>
> We have incorporated your suggestions in the updated paper version (indeed equation 4 better preserves a clear flow from prior equations in the form you suggested).
>
> Thanks again for your support and the suggestions for these improvements!

---

### Official Review · Reviewer_Z6SX · 2021-11-03

**Correctness:** 3
**Technical Novelty And Significance:** 2
**Empirical Novelty And Significance:** 2
**Recommendation:** 8
**Confidence:** 3

**Main Review:**

Strengths: 1. Some experimental results are interesting. Simply adding related sentences in the pre training input context helps end performance.

Weaknesses: 1. I think the presentation of the paper needs to be improved. For the theoretical analysis, it was not clear to me what is the contribution of the current analysis compared to the Levine 2020 paper. I felt similar conclusions can be drawn from the results of that paper as well. It will be good to rewrite highlighting the contributions. I also noticed a lot of repeated text in section 2 from Levine 2020 paper; will be good to modify.

2. The empirical part of the paper shows improved performance of adding similar sentences to the context of LM training. I felt this was quite separate from the theoretical analysis. It does not follow from the theoretical results that adding similar sentences will be a good thing. As a result, having these in the same paper looked incoherent to me.

3. The experimental results are weak. The gains are not super high. I will be more convinced if evaluation is done on a wider range of tasks.

**Summary Of The Paper:**

The authors show empirically and theoretically that transformer based language models cannot effectively exploit dependencies between sentences in different inputs. For the theory part, they introduce a new measure called \eps-sep rank. A function is said to have low \eps-sep rank if it can be approximated up to error \eps by a function with low separation rank. Bounding the measure shows that if learning rate is low, the information from different training instances cannot be well integrated. Empirically, they present experiments where the task specific pre training was altered to include non-task specific similar sentences in the context of the LM. They show that augmenting context of pre training inputs with related similar sentences from Wikipedia helped improve end performance on similarity and zero shot open domain QA tasks.

**Summary Of The Review:**

I was not clear about the impact of the theoretical analysis presented, compared to past work. It will be good to rewrite the section clarifying the contributions and novel conclusions from the analysis. The empirical section sounded disparate from the theoretical part of the paper, and the results are weak. I suggest evaluating on a wider variety of tasks.

---

> ### Author Response · Authors · 2021-11-19
> **Response to reviewer -- part 1**
>
> We thank you for your thoughtful feedback.
>
> 1.
>
> The novelty of our theoretical setup, when compared to Levine et al. 2020 (and all other network expressivity results we are aware of) is in analyzing the sequential representation. Expressivity results traditionally pertain to the question of **how expressive an architecture is with respect to its input** (in our framework this is the power of the “in-context” representation). Our framework is the first to place a completely different expressivity question on the same grounds as the traditional one: **how expressive is the resulting network with respect to examples seen during its training process** (we call this the power of the “sequential” representation).
>
> Specifically when comparing to Levine et al. 2020, they provided bounds on the separation rank of the Transformer architecture with respect to its input (bounding the power of the “in-context” representation). After we introduce the novel sequential setting, and ask a new question about it (can it compare to traditional architecture expressivity), we answer the question, which requires non trivial tools in and of itself. The reason that this answer was non trivial, as we allude to in the new intuitive level proof sketch placed after the presentation of the main theorems, is that when examining expressivity in the lens used by Levine et al. 2020 (the separation rank), there seems to be no difference between the power of the two different representations! Seemingly, in the sequential representation, the new sentence S_2 enters the network and immediately encounters the gradient updated vocabulary matrix weights that depend on S_1. So the two sentences are mixed right off the bat, similarly to the in-context case. Our technical innovation relative to Levine et al. 2020 of introducing the \epsilon separation rank, allowed us to reveal what is really at play -- while the two indeed mix from the start, S_1 is coupled to the learning rate and therefore we are able to show that much of its contribution appears in negligibly small orders of magnitude which do not contribute to the \epsilon separation rank of the function with respect to S_1 and S_2.
>
> At the bottom line, no conclusions regarding the sequential mechanism can be inferred from the work of Levine et al 2020, and therefore the in context bias could only be established given our unique contributions listed above.
>
> Per your suggestions, we removed some redundant text in section 2 and added a summary of our main contributions at the end of the intro section, which we hope helps to convey some of the above notions more clearly.

---

> > ### Author Response · Authors · 2021-11-19
> > **Response to reviewer -- part 2**
> >
> > 2.
> >
> > Regarding the connection of our experimental setup to the theoretical messages established in the main part of our paper. As you note, it indeed does not follow from our theoretical results that adding similar sentences will be a good thing. The logic behind our experiments is quite different -- we take it for granted that in some cases adding similar sentences is a good thing, i.e., will be helpful for solving the task. We chose two cases in which we had strong intuition that similar sentences would be particularly useful (sentence similarity, where neighboring sentences can be useful for learning what is similar, and zero shot question answering from Wikipedia, where better representing the passage where the answer resides by incorporating similar sentences was expected to be useful).
> > Then, we experimented around our basic theoretical findings on the in-context bias, which predict that **if** similar sentences are a useful signal, **then** it would be better to incorporate them in context rather than to train on them but in different examples.
> > We acknowledge that the above motivations were not conveyed very clearly in the submitted version, we added relevant clarifications in the experimental section - thank you!
> >
> > 3.
> >
> > With recommendations from you and other reviewers, we quickly evaluated the zero shot performance of our kNN-pretrained model on several additional tasks from the GLUE benchmark (recommended by Reviewer yRBu), and added the results in appendix F.
> > The outcome resembles that of Natural Questions -- the zero shot scores are generally low but kNN-Pretraining again significantly moves the needle relative to the baselines after just 10% of the regular pretraining time.
> >
> > We acknowledge that even though we think that both proposed methods have potential to be matured into useful training schemes, the breadth of experimentation in our submitted paper was not enough to truly establish this. Note that this is something we were straightforward about in our submitted paper; we consistently refer to the kNN pretraining methods as “controlled experiments” or “controlled setting exemplifications for leveraging the in context bias..” --
> >
> > Given that results are manifested on more NLU tasks and that the signal from NQ is reinforced, we further conjecture that when using stronger models (that train for longer and over more data), the positive effect of kNN-Pretraining will be enhanced -- since as the model improves, it can better understand and utilize the various in-context hints that kNN-Pretraining provides. In the camera ready version, we will follow up pn the experimental signals in the paper body and Appendix F by pretraining stronger models, going beyond the controlled and limited setting of 3 epochs on Wikipedia, in order to show more practical usefulness. We will integrate existing GLUE scores of Appendix F and any new results more naturally in the experiments section of the main body, the message being that kNN pretraining can be useful on a larger variety of tasks.
> >
> > Note that others identify the promise in such methods, e.g., a recent empirical work that shows large gains in a TAPT setting with very basic retrieval functions https://arxiv.org/abs/2111.04130. Our theoretical results can affect works like this by motivating the concatenation of related examples in-context for even larger gains.
> >
> >
> > Thanks again for your thoughtful comments which allowed us to improve our paper!

---

> > > ### Comment · Reviewer_Z6SX · 2021-11-21
> > > **Acknowledgment of response**
> > >
> > > Happy with explanations added, thanks.

---

### Official Review · Reviewer_yRBu · 2021-11-04

**Correctness:** 3
**Technical Novelty And Significance:** 3
**Empirical Novelty And Significance:** 2
**Recommendation:** 8
**Confidence:** 3

**Main Review:**

Strength:
1.	The authors analyze the in-context bias of the self-attention model, which could inspire some research works on designing training examples.
2.	The authors propose to include related texts retrieved by the kNN method in a single training sample, which is proved effective in solving sentence similarity tasks.
3.	A theoretical analysis of the in-context bias.

Weakness:
1.	The introduction of the motivation (the concept of in-context bias) is not easy to understand at the very beginning.  The paper said: “the pretrained NLM can model much stronger dependencies between text segments that appeared in the same training example, than it can between text segments that appeared in different training examples.”  Acutally it seems quite natural for me and I did not realize it is a problem until I saw more explanations in section 1.1.
2.	The theory is a bit complicated and not easy to follow.
3.	The experiments are limited. The authors only conduct the evaluation on sentence similarity tasks and open domain QA tasks. However, there are many other tasks that involve sentence pairs. For example, sentence inference tasks such as MNLI and RTE are common tasks in NLP field. The authors should conduct experiments on more types of sentence pair tasks.


**Summary Of The Paper:**

The paper addresses the bias induced by the inconsistency between the pretraining examples and downstream examples. Specifically, the text is continuous during pretraining while could be non-neighboring in downstream tasks. The authors proposed two methods: kNN-TAPT and kNN-pre-training in Task Adaptive Pre-training (TAPT) step or general Pre-training step respectively, where each training sample is composed of a sentence and its semantically closed neighbors (by kNN search). The method is effective in sentence similarity tasks (in kNN-TAMP scenario) and closed book open domain QA tasks (in kNN-Pre-training scenario).  The paper also gives a theoretical analysis of the so-called in-context bias,  by quantifying the NLM’s ability to model dependencies between two sentences that appear in the same training example (the in-context representation) and in different training examples (the sequential representation).

**Summary Of The Review:**

The paper addresses the inductive bias of in-context learning. The studied problem is meaningful and could be a potential breakthrough for pretrained language model. However, the written of the paper is a bit hard to follow and the experiments is somewhat limited.

---

> ### Author Response · Authors · 2021-11-19
> **Response to reviewer**
>
> We thank you for your thoughtful and supportive feedback.
>
> 1.
>
> Thank you for your comment on the clarity of the motivation at the very beginning. It is indeed very important to have our readers understand why our result is interesting as early as possible. We added a clarifying sentence to the second paragraph of the paper, which would hopefully help other readers in their first glance of the paper.
>
> 2.
>
> Regarding your comment on the complexity of the theory, we added an intuitive level proof sketch after the presentation of the main theorems, which illustrates how we use our presented theoretical framework in order to establish our results (the gap between the in-context and sequential cases). To an interested reader, this proof sketch can provide a missing link, at the intuitive level, between the concepts of separation rank and epsilon separation rank in section 2.2, and the main claims in section 2.3 on the in-availability of information from previously seen examples that were stored during the gradient update mechanism.
>
> 3.
>
> Regarding your comments on our limited experimental setup, and also on the overall view of the paper as given in your Summary Of The Paper section --  our aim was for a slightly different balance between experimental and theoretical contributions than the one mirrored in your text.
>
> Specifically, our paper revolves around establishing a theoretical understanding of the in-context bias related phenomena that affect modern LM training. We prove our statements, but our theoretical framework includes relaxations. These relaxations seem plausible a-priori (and some of them are also reinforced by experiments in earlier works), but we wanted to include experiments in order to reinforce the validity of our framework. Generally, this was the purpose of including the experiments in our paper.
>
> With recommendations from you and other reviewers, we quickly evaluated the zero shot performance of our kNN-pretrained model on several additional tasks from the GLUE benchmark (including MNLI and RTE), and added the results in appendix F.
> The outcome resembles that of Natural Questions -- the zero shot scores are generally low (fitting zero shot on classification tasks with unidirectional models of the tested sizes), but kNN-Pretraining again significantly moves the needle relative to the baselines after just 10% of the regular pretraining time.
>
> We acknowledge that even though we think that both proposed methods have potential to be matured into useful training schemes, the breadth of experimentation in our submitted paper was not enough to truly establish this. Note that this is something we were straightforward about in our submitted paper; we consistently refer to the kNN pretraining methods as “controlled experiments” or “controlled setting exemplifications for leveraging the in context bias..” --
>
> Given that results are manifested on more NLU tasks and that the signal from NQ is reinforced, we further conjecture that when using stronger models (that train for longer and over more data), the positive effect of kNN-Pretraining will be enhanced -- since as the model improves, it can better understand and utilize the various in-context hints that kNN-Pretraining provides.
> In the camera ready version, we will follow up pn the experimental signals in the paper body and Appendix F by pretraining stronger models, going beyond the controlled and limited setting of 3 epochs on Wikipedia, in order to show more practical usefulness. We will integrate existing GLUE scores of Appendix F and any new results more naturally in the experiments section of the main body, the message being that kNN pretraining can be useful on a larger variety of tasks.
>
> Note that others identify the promise in such methods, e.g., a recent empirical work that shows large gains in a TAPT setting with very basic retrieval functions https://arxiv.org/abs/2111.04130. Our theoretical results can affect works like this by motivating the concatenation of related examples in-context for even larger gains.
>
>
> Thanks again for your thoughtful comments which allowed us to improve our paper!

---

### Official Review · Reviewer_ADar · 2021-11-07

**Correctness:** 3
**Technical Novelty And Significance:** 4
**Empirical Novelty And Significance:** 2
**Recommendation:** 8
**Confidence:** 3

**Main Review:**

The main results of this paper are of broad interest. Questions about how the nature of the input to LMs affects learning are important and poorly understood. The claim that the within- and between-example is crucial is novel and thought provoking.

The arguments are highly technical and require a lot of expertise, (including familiarity in particular with Levine, 2020, on which the proof builds substantially) to understand. This is not necessarily a strength or a weakness---there is great value to developing techniques for making this kind of formal argument. However, I find it difficult to follow even the thread of the argument at an intuitive level, which is a shame because it's interesting to me. I suspect I'm not the only potential reader who will have this problem. The paper could reach a much wider audience (which I think would be an improvement) if it provided more intuitive paraphrases of the main results in Section 2.3.

There are a few unaddressed issues with the authors' main suggestion: that we can improve LMs’ abilities to make cross-example connections by constructing examples out of interleaved texts. Assuming a fixed example length, this means portions of text from the same source that otherwise would have been in the same example in the standard chunking scheme would now be separated. However, it seems that, in general, dependencies within a text are more important than dependencies between random semantically related sentences from different texts. Another problem is that a language model trained on this kind of interleaved text would presumably also generate less coherent texts.

I also have some doubts about the empirical results. The results about kNN-task adaptive pretraining seem to have a confound: The method for retrieving data for this intermediate training step involves finding sentences with high similarity, but the task on which this method is shown to lead to improvements is itself a semantic similarity task. The claim is that the method should be helpful for many kinds of tasks, but my intuition is that this "coincidental" alignment of the data collection and the evaluation task makes the result less generalizable.

The empirical results about zero-shot QA don't help much because the task is clearly too difficult. The improvement on the models trained with the new method is from F1 of ~0.001 to ~0.01. Do larger LMs succeed at this task? If so, then it seems that models tested in this paper are just too small or ineffective for the task. Is there an easier task where this method leads to strong performance but ordinary pretraining does not?



**Summary Of The Paper:**

This paper provides an argument for reconsidering how we construct a single training example during LM (pre)training. It gives a formal proof that dependencies between two texts are vanishingly weak when the texts are presented in separate training examples (but not when they are in the same training example). Intuitively, in order to make cross-textual inferences between texts in different examples, a model must be able to access information stored in its weights, but commonly used learning rates and model architectures ensure that the updates to the weights are too small to recover this information.

The paper suggests remedying this problem by changing the way in which we construct training examples. Rather than obtaining examples by separating texts into contiguous chunks, we can construct examples from multiple semantically related chunks found by computing the similarity of text embeddings in the training corpus. Empirical results are presented which show that task-adaptive (i.e. post-pretraining) training on such data leads to greatly improved performance on zero-shot semantic similarity. Further results show that doing pretraining with these kinds of examples improves zero-shot QA performance by a small margin (from an extremely low baseline).

**Summary Of The Review:**

This paper makes an intriguing claim about the effect of the input on LM learning, which is backed up by a formal argument. These kinds of arguments are valuable. However, the argument is difficult to follow for a non-specialist audience. Also, the empirical results the authors report have some confounds due to questionable task selection.

---

> ### Author Response · Authors · 2021-11-19
> **Response to reviewer -- part 1**
>
> We thank you for your thoughtful and supportive feedback.
>
> 1.
>
> We appreciate your interest in an intuitive level description of the theoretical argumentation that leads to our main results. We decided not to include something like this in the original version due to space considerations, but we agree with your comment on its benefits. We have accordingly added an intuitive proof sketch after the statement of the main theorems, which illustrates how we use our presented theoretical framework in order to establish our results (the gap between the in-context and sequential cases). To an interested reader, this proof sketch can provide a missing link, at the intuitive level, between the concepts of separation rank and epsilon separation rank in section 2.2, and the main claims in section 2.3 on the in-availability of information from previously seen examples that were stored during the gradient update mechanism.
>
> 2.
>
> Regarding your concern on the difference between the data seen in-context during kNN pretraining versus regular pretraining, we first draw your attention to the following implementation detail taken from section 3.2, intended to address just that: “During kNN-Pretraining, half of the batch contained regular pretraining examples and half contained the prepared kNN examples”. We see why this sentence could have been overlooked as it is not emphasized or explained in the original version; in the resubmitted version we added the motivation for it. We propose kNN pretraining only after the model trains for a significant amount of pretraining steps (we used 200K, roughly 3 epochs on the Wikipedia pretrianing corpus). Then, during the additional (20K) kNN pretraining steps, we propose training half of the batch in the kNN method which appends related sentences in context, but half of each batch is kept as regular pretraining examples in order to retain the longer context dependencies that was acquired in the first stage, in parallel to better learning the kNN dependencies.
>
> Even with our 50/50 approach, It is true that alongside its benefits, kNN pretraining may suffer from degradation due to missing longer ranged context. We find figure 20(left) in the scaling laws work of  Kaplan et al. (2020) https://arxiv.org/pdf/2001.08361.pdf to demonstrate nicely that degradation of longer ranged context can be also identified in perplexity evaluation. We expect the perplexity of a model trained with 100% kNN pretraining examples, without mixing regular “long-ranged” pretraining examples in the batch, to resemble the one shown in figure 20(left) of Kaplan et al. (2020) for a context size of around 30 tokens, our typical used sentence lengths. Our 50/50 batch integration scheme allows retention of previously learned long ranged connections, so it is expected to attenuate this degradation.

---

> > ### Author Response · Authors · 2021-11-19
> > **Response to reviewer -- part 2**
> >
> > 3.
> >
> > Regarding your concerns about the experimental setup (both about a potential confound in the kNN TAPT experiments and regarding the hardness of zero-shot QA for the model sizes we used) --
> >
> > Overview on the intended role of our experiments:
> > Our paper is based around our theoretical findings which shed light on questions like “my model will train on given texts, does it matter how I construct the specific training examples out of these texts?”. While providing answers, our theoretical framework includes relaxations. These relaxations seem plausible a-priori (and some of them are also reinforced by experiments in earlier works), but we wanted to include experiments in order to reinforce the validity of our framework. Generally, this was our purpose in including the experiments in our paper.
> > To this end, when planning the experiments, we asked whether we can show that when two models train on the **same exact texts**, the strategy for pretraining example construction significantly affects the outcome.
> >
> > Regarding you comment on the kNN TAPT setting:
> > In the kNN TAPT experiments on SentEval, the fact that finding similar sentences from wikipedia helps with better identifying the similarity level between sentences is indeed not surprising, and actually makes this a good setup for testing our hypothesis. What we set out to show and indeed have shown in the kNN TAPT experiment, in direct accordance with our theoretical findings, is that when **fixing** the TAPT data to include the SentEval data *plus* these clearly relevant sentences, adding the related ones *in context* allows the model to better internalize these relevant connections, and the result is better than training with them but *not in context*. We added a clarifying sentence in the relevant experiments section stating that sentence similarity is a good playground for examining the effect of the incontext bias **because** nearest neighbors are expected to be a helpful signal on this task.
> >
> > In the kNN pretraining experiments on NQ, performance is not what we set out to optimize but rather cleanliness of the signal. Larger models indeed get much better results in this task (see, e.g., Roberts et al. 2020 https://arxiv.org/pdf/2002.08910.pdf), and even for the same size one could increase the score further by using an existing pretrained model or training for longer and on more data. Since NQ is an open domain QA setup from Wikipedia (answers can be found in Wikipedia), we thought that if the model had better access to similar sentences when training on the passage in which the answer resides, it will better incorporate that information in its weights and provide these answers more readily at zero shot. This is somewhat similar to the motivation of TAPT if you think about it.
> > So in summary, NQ was chosen due to its close connection to the wikipedia corpus, which allowed us to imagine a clean picture when hoping for significant results that support our in-context bias indications.  We added a clarifying sentence in the relevant experiments section stating why we chose to experiment with Natural Questions.
> >
> >
> > With recommendations from you and other reviewers, we quickly evaluated the zero shot performance of our kNN-pretrained model on several additional tasks from the GLUE benchmark (recommended by Reviewer yRBu), and added the results in appendix F.
> > As in the case of Natural Questions, we evaluate the zero-shot performance of our models, since it is a direct probe to the abilities of the model straight after the process of pretraining. Accordingly, the examined GLUE classification tasks are still not “easy” for the examined unidirectional models in zero shot, and in fact the outcome resembles that of Natural Questions -- the baselines get only slightly higher scores than random guess, but kNN-Pretraining again significantly moves the needle after just 10% of the regular pretraining time.
> > Appendix F shows that even NLU tasks that don’t have a close connection to Wikipedia like NQ does are improved by kNN Pretraining.

---

> > > ### Author Response · Authors · 2021-11-19
> > > **Response to reviewer -- part 3**
> > >
> > > As a final note, we acknowledge that even though we think that both proposed methods have potential to be matured into useful training schemes, the breadth of experimentation in our submitted paper was not enough to truly establish this. Note that this is something we were straightforward about in our submitted paper; we consistently refer to the kNN pretraining methods as “controlled experiments” or “controlled setting exemplifications for leveraging the in context bias..”  --
> > >
> > > Given that results are manifested on more NLU tasks and that the signal from NQ is reinforced, we further conjecture that when using stronger models (that train for longer and over more data), the positive effect of kNN-Pretraining will be enhanced -- since as the model improves, it can better understand and utilize the various in-context hints that kNN-Pretraining provides.
> > > In the camera ready version, we will follow up pn the experimental signals in the paper body and Appendix F by pretraining stronger models, going beyond the controlled and limited setting of 3 epochs on Wikipedia, in order to show more practical usefulness. We will integrate existing GLUE scores of Appendix F and any new results more naturally in the experiments section of the main body, the message being that kNN pretraining can be useful on a larger variety of tasks.
> > >
> > > Note that others identify the promise in such methods, e.g., a recent empirical work that shows large gains in a TAPT setting with very basic retrieval functions https://arxiv.org/abs/2111.04130. Our theoretical results can affect works like this by motivating the concatenation of related examples in-context for even larger gains.
> > >
> > >
> > >
> > > Thanks again for your thoughtful comments which allowed us to improve our paper!

---

> > > > ### Comment · Reviewer_ADar · 2021-11-21
> > > > **Acknowledgment of response**
> > > >
> > > > Thank you to the authors for your detailed response.
> > > >
> > > > I'll start by responding to (3), because I'm not totally convinced. I think it's great that you've replicated the NQ results on GLUE---I'm convinced that this performance bump is not an accident. However, I'm still hesitant to read into small changes in performance so close to chance. The models are clearly barely able to perform the zero-shot QA task, so it's not clear if these improvements would generalize to models that *can* perform the task (which are ultimately more like the models we care about). Some alternatives:
> > > > - Maybe evaluate on LAMA or other LM test sets that are designed to not require task-specific training.
> > > > - Try fine-tuning. Even though you are rightly concerned that fine-tuning tells us less about specific inter-textual dependencies learned *during* pretraining, a significant improvement in fine-tuning performance would suggest general benefits.
> > > >
> > > > As for the SemEval results, the confound I brought up wasn't meant to cast doubt on whether semantic similarity genuinely improved with KNN TAPT. It was meant to cast doubt on whether this approach would be helpful for *any* other task. Let me put it this way: I have no reason to think KNN TAPT would help with solving NLI, or any number of tasks. If you could show meaningful improvements in some setting (fine-tuning, zero-shot, few-shot, etc.) on one or more unrelated tasks, I'd be convinced that this approach might be broadly useful. Otherwise, why would we believe that it's more than a hack to improve SemEval performance?
> > > >
> > > > 1. Thanks for including the proof sketch---this clarify the outline of the proof, though I would still encourage you to include even more intuitive descriptions along the way, if space allows.
> > > >
> > > > 2. Apologies for missing the 50/50 training approach. This seems to be well justified now. If you could demonstrate that KNN pretraining and regular pretraining yield similar PPL on a shared test set, this would be very convincing. Of course, the percent of KNN examples is now a hyperparameter which could be tuned in subsequent work.
> > > >
> > > > Thanks for you work!

---

> > > > > ### Author Response · Authors · 2021-11-23
> > > > > **Final author's response**
> > > > >
> > > > > Thank you very much for your concrete suggestions for further evaluating both kNN pretraining methods. These are good suggestions, and while we didn't have enough time to complete these by the end of the discussion period, we are conducting these evaluations and will include them in the camera ready version.
> > > > >
> > > > > We do expect to show that kNN-TAPT would be useful for other tasks apart from sentence similarity ones. For tasks defined by input x and output y, we think that "elaborating" on x only, via semantically related excerpts from the pretraining corpus, can in many cases cause the model to "better understand" x due to the related excerpts, and therefore in many tasks in which this "better understanding" is relevant to the task, the model will do a better job in predicting y.
> > > > > In other words, semantically related sentences (as captured by SOTA sentence embeddings) which are not necessarily neighboring in the pretraining corpus, can provide non-trivial contextualization and improvement also in other NLU tasks which are not only sentence similarity.
> > > > >
> > > > > The fact that sentence embeddings for sentence similarity was improved is not likely to be "stand alone" as you hint -- many NLU tasks are inter-related, so capturing better representations that improve on SST can be expected to improve the model's performance on other tasks.
> > > > > See for example figure 3 in Vu et al 2021 https://arxiv.org/pdf/2110.07904.pdf who examine task relatedness via prompts, showing that prompts of many NLU tasks have positive correlation, and specifically that prompts for STS-B, for which we've shown that kNN-TAPT yields significant gains, is highly correlated with prompts of several other NLU tasks.
> > > > >
> > > > > With that, we wish reiterate the role of these experiments as we see it within our overall contribution, in a focused manner --
> > > > >
> > > > > We view our theoretical framework and results, given in section 2, as the main contribution of this paper (see overview in our response to reviewer Z6SX). This is a timely theoretical contribution because a wide variety of leading empirical approaches can be tied to it, leveraging this intuitive result in non-trivial manners. Section 1.1 of the paper surveys some of these strong empirical approaches -- their results indicate that the in context bias leads to very meaningful gains in various settings.
> > > > >
> > > > > We elected to supplement to the abovementioned existing empirical signals with more targeted and cleaner experiments, to support the proven phenomenon. This is the role of section 3.
> > > > >
> > > > > Our theoretical result implies that **if** a certain task can be improved by a better joint representation of two signals, **then** it is better to include them in-context.
> > > > > To this end, since related sentences from Wikipedia are a clear signal for sentence similarity, this experiment fulfills the above main purpose of experiments to our overall contribution.
> > > > >
> > > > > The question of improvement of kNN-TAPT on other tasks is only a "bonus";
> > > > > our main contribution, in the form of our theoretical result, extends beyond KNN Pretraining, which is only an example of its usage. Specifically, the theoretical result can inspire further approaches beyond kNN-Pretraining, as we detail in section 2 of our response to Reviewer cz9E (as examples, the in context bias indicates that multiple mentions of the same rare entity, event, or concept can be concatenated within the same training example for better learning. See more examples in that response to the reviewer below).
> > > > >
> > > > > We wish to thank you again for your detailed attention to our paper! Your feedback has truly  helped us improve it.

---

### Author Response · Authors · 2021-11-19
**A summary of the changes in our revised version**

Following the reviewers’ comments, we have uploaded a revised version of our paper, with the following noteworthy modifications:

* We added a clarifying sentence in the second paragraph of the intro.
* We added a summary of our main contributions at the end of the intro section.
* We removed some redundant text in section 2.2.
* We added an intuitive proof sketch after the statement of the main theorems.
* We improved the assumption on the boundedness of the weights and gradients in theorem 1 to an assumption on their absolute values, by using the definition introduced at the bottom of page 12.
* We improved the text in the experiments section such that motivations for the specific chosen datasets are clearer.
* We added a comparison of zero-shot performance of kNN pertained models versus regular baselines on several GLUE tasks in appendix F.

---

### Decision · Program_Chairs · 2022-01-20

**Decision:**

Accept (Spotlight)

**Comment:**

This paper presents a novel framing of what's at stake when selecting/segmenting text for use in language model pretraining. Four reviewers with experience working with these models agreed that the conceptual and theoretical work here is insightful and worth sharing. The empirical work is fairly small-scale and does not yet support broad conclusions, but reviewers did not see such conclusions as necessary for the paper to be valuable.